# Lean and Mean Adaptive Optimization via Subset-Norm and Subspace-Momentum with Convergence Guarantees

**Thien Hang Nguyen** [1 2 *]   **Huy Le Nguyen** [1]

## Abstract

We introduce two complementary techniques for efficient optimization that reduce memory requirements while accelerating training of large-scale neural networks. The first technique, *Subset-Norm* step size, generalizes AdaGrad-Norm and AdaGrad(-Coordinate) through step-size sharing. Subset-Norm (SN) reduces AdaGrad's memory footprint from $O(d)$ to $O(\sqrt{d})$, where $d$ is the model size. For non-convex smooth objectives under coordinate-wise sub-gaussian noise, we show a noise-adapted high-probability convergence guarantee with improved dimensional dependence of SN over existing methods. Our second technique, *Subspace-Momentum*, reduces the momentum state's memory footprint by restricting momentum to a low-dimensional subspace while performing SGD in the orthogonal complement. We prove a high-probability convergence result for Subspace-Momentum under standard assumptions. Empirical evaluation on pre-training and fine-tuning LLMs demonstrates the effectiveness of our methods. For instance, combining Subset-Norm with Subspace-Momentum achieves Adam's validation perplexity for LLaMA 1B in approximately *half* the training tokens (6.8B vs 13.1B) while reducing Adam's optimizer-states memory footprint by more than 80% with minimal additional hyperparameter tuning.

## 1. Introduction

Adaptive optimizers like Adam (Kingma & Ba, 2014), Ada-Grad (Duchi et al., 2011), and RMSProp (Tieleman et al.,

---
[*]Work done at Northeastern University. [1]Khoury College of Computer Sciences, Northeastern University, Boston. [2]Contextual AI. Correspondence to: Thien Nguyen <nguyen.thien@northeastern.edu; thien.nguyen@contextual.ai>, Huy Nguyen <hu.nguyen@northeastern.edu>.

*Proceedings of the $42^{nd}$ International Conference on Machine Learning*, Vancouver, Canada. PMLR 267, 2025. Copyright 2025 by the author(s).

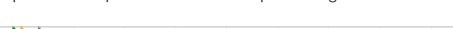

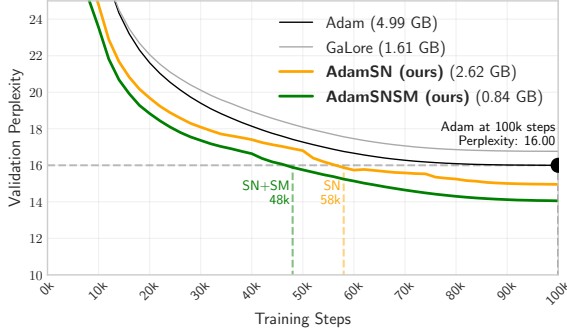

*Figure 1.* Validation perplexity for Adam, GaLore (Zhao et al., 2024), **AdamSN**, and **AdamSNSM** (**ours**) during LLaMA 1B model training for 13.1B tokens (100K steps). Optimizer memory footprint is shown in parentheses. Adam achieves a perplexity of 16.00 at 100,000 steps, while AdamSN and AdamSNSM exhibit lower perplexity earlier in training at 58,000 and 48,000 steps.

2012) are de facto methods for training large-scale deep neural networks. However, the optimizer states for the momentum and second moment (or adaptive step size) terms are memory intensive, consuming as much as twice the size of the model. As deep neural networks continue to grow in the era of large-language models (LLMs), concerns that were previously overlooked, such as the memory consumption of optimizer states, have become an active area of research. Indeed, numerous methods have recently emerged to reduce the memory footprint of optimizer states (e.g. Adam's momentum and second moment terms) with approaches ranging from quantization (Li et al., 2024a; Dettmers et al., 2021; 2024), low-rank decomposition (Hu et al., 2021; Lialin et al., 2023; Zhao et al., 2024; Shazeer & Stern, 2018), sketching-based dimensionality reduction (Muhamed et al., 2024; Hao et al., 2024), etc. Existing methods either lacks theoretical guarantees, requires strong assumptions, trades too much performance, or requires expensive additional tuning for the memory saving, especially in pretraining tasks.

**Our contributions.** We aim to reduce memory consumption while maintaining strong performance and theoretical guarantees. To this end, we introduce two memory-

efficient optimization algorithms for large-scale DNN training: **Subset-Norm** (**SN**) for adaptive step-size memory reduction (Section 3) and **Subspace-Momentum** (**SM**) for momentum compression (Section 4). While existing approaches trade performance for memory savings, our theoretically-grounded methods achieve both a reduced memory footprint and faster training:

- **Subset-Norm (SN)**: A memory-efficient adaptive step-size algorithm with high-probability convergence guarantees for non-convex objectives under coordinate-wise sub-gaussian noise. By unifying AdaGrad-Coordinate's and AdaGrad-Norm's analysis, we show that the SN adaptive step size (Algorithm 2) achieves improved dimensional dependence, while reducing the memory footprint from $O(d)$ to roughly $O(\sqrt{d})$. On LLaMA models' pretraining tasks, SN step sizes achieves better perplexity than coordinate-wise step size across a range of optimizers and model sizes, while using significantly less memory and introducing minimal additional hyperparameters.[1]
- **Subspace-Momentum (SM)**: A momentum compression method that applies momentum in a chosen subspace and SGD in the orthogonal complement with high-probability convergence guarantees under sub-gaussian noise for non-convex smooth objectives. When combined with SN, for some selected dimension $k$ less than $d$,[2] our method (SNSM) reduces the memory footprint of Adam and Ada-Grad+momentum from $2d$ to $k + \sqrt{d}$ (see Table 4) while delivers improved training speed and performance.

Empirical evaluations on LLaMA models from 60M to 1B parameters demonstrate that our algorithms scale effectively and attain better performances than existing optimizers. Our proposed methods are simple to implement, require minimal additional hyperparameter tuning, and are compatible with modern distributed training frameworks like FSDP (Zhao et al., 2023; Rajbhandari et al., 2020). We provide an implementation in PyTorch at https://github.com/timmytonga/sn-sm.

## 2. Preliminaries

### 2.1. Common Optimizers and Memory Footprint

Consider the generic template in Algorithm 1, which captures a broad range of first-order optimizers that leverage either momentum or adaptive step sizes. Many standard optimizers can be represented within this framework by varying choices of momentum and adaptive step-size terms, as shown in Table 1. Generally, optimizers with higher memory requirements, such as Adam, tend to outperform

---

[1]Although the subset size can be tuned (Section 5.3), we provide a heuristic in Section 3.2 that works effectively across model sizes, eliminating the need for additional tuning.

[2]Typically, $k$ is chosen to be around $d/4$.

---

**Algorithm 1** Generic Template for Stochastic Adaptive Optimizers with Momentum

**Input:** Initial point $x_1 \in \mathbb{R}^d$, base step size $\eta > 0$, constant $\epsilon > 0$.
**for** $t = 1$ **to** $T$ **do**
   Obtain stochastic gradient $\widehat{\nabla} f(x_t)$
   $m_t = \text{update\_momentum}(\widehat{\nabla} f(x_t); m_{t-1})$
   $v_t^2 = \text{update\_adaptive\_stepsize}(\widehat{\nabla} f(x_t); v_{t-1}^2)$
   $x_{t+1} = x_t - \eta \cdot \frac{m_t}{v_t + \epsilon}$       {Update step}
**end for**

---

*Table 1.* Update rules for common optimizers in the framework of Algorithm 1. We omit bias correction terms and numerical stabilizer $\epsilon$ for simplicity. Memory for optimizer state is shown for model of size (and memory footprint) $d$.

| Optimizer | Memory | Update Rules |
|---|---|---|
| Adam | $2d$ | $m_t = \beta_1 m_{t-1} + (1 - \beta_1)\widehat{\nabla} f(x_t)$ 
 $v_t^2 = \beta_2 v_{t-1}^2 + (1 - \beta_2) \cdot \widehat{\nabla} f(x_t)^2$ |
| SGDm | $d$ | $m_t = \beta m_{t-1} + (1 - \beta)\widehat{\nabla} f(x_t)$ 
 $v_t^2 = \text{ID}$ |
| AdaGrad-Coord | $d$ | $m_t = \widehat{\nabla} f(x_t)$ 
 $v_t^2 = v_{t-1}^2 + \widehat{\nabla} f(x_t)^2$ |
| RMSProp | $d$ | $m_t = \widehat{\nabla} f(x_t)$ 
 $v_t^2 = \beta_1 v_{t-1}^2 + (1 - \beta_1) \cdot \widehat{\nabla} f(x_t)^2$ |
| AdaGrad-Norm | $1$ | $m_t = \widehat{\nabla} f(x_t)$ 
 $v_t^2 = v_{t-1}^2 + \left\| \widehat{\nabla} f(x_t) \right\|^2$ |
| SGD | $1$ | $m_t = \widehat{\nabla} f(x_t)$ 
 $v_t^2 = \text{ID}$ |

---

more memory-efficient alternatives like SGD and RMSProp. We aim to design principled algorithms that achieve the best of both worlds: strong performance and memory-efficient.

### 2.2. Assumptions and Notations

For our theoretical analysis, we consider the unconstrained non-convex stochastic optimization problem $\min_{x \in \mathbb{R}^d} f(x)$ where $f : \mathbb{R}^d \to \mathbb{R}$ is the objective function. We assume access to an history independent, non-biased gradient estimator $\widehat{\nabla} f(x)$ for any $x \in \mathcal{X}$, that is $\mathbb{E}\left[ \widehat{\nabla} f(x) \mid x \right] = \nabla f(x)$. Furthermore, we assume that $f$ is an $L$-smooth function:

$$\|\nabla f(x) - \nabla f(y)\| \leq L \|x - y\|, \text{ for all } x, y \in \mathbb{R}^d.$$

Smoothness implies the following quadratic upperbound that we will extensively utilize: for all $x, y \in \mathbb{R}^d$ we have $f(y) - f(x) \leq \langle \nabla f(x), y - x \rangle + \frac{L}{2} \|y - x\|^2$.

**Notations.** Let $v_i$ denote the $i$-th coordinate of a vector $v \in \mathbb{R}^d$. If a vector $x_t$ is already indexed as part of a sequence of vectors (where $x_t$ denotes the $t$-th update) then we use $x_{t,i}$ to denote $x_t$'s $i$-th coordinate and $x_{t,\Psi} \in \mathbb{R}^k$ to denote the indexing with respect to an ordered subset $\Psi \subseteq$

$[d]$ of size $k$ where $(x_{t,\Psi})_k = x_{t,\Psi^{(k)}}$ with $\Psi^{(k)}$ denoting the $k$-th element of $\Psi$. For gradients, we let $\nabla_i f(x) := \frac{\partial f}{\partial x_i}$ denote the partial derivative with respect to the $i$-th coordinate. Similarly, for stochastic gradients $\widehat{\nabla} f(x)$, we let $\widehat{\nabla}_i f(x)$ denotes its $i$-th coordinate. If $a, b \in \mathbb{R}^d$, then $ab$ and $a/b$ denotes coordinate-wise multiplication and division, respectively i.e. $(ab)_i = a_i b_i$ and $(a/b)_i = a_i/b_i$.

**Coordinate-wise sub-gaussian noise assumption.** A random variable $X$ is $\sigma$-sub-Gaussian (Vershynin, 2018) if

$$\mathbb{E}\left[\exp\left(\lambda^2 X^2\right)\right] \le \exp\left(\lambda^2 \sigma^2\right) \text{ for all } \lambda \text{ such that } |\lambda| \le \frac{1}{\sigma}.$$

If we denote the stochastic gradient noise as $\xi_t := \widehat{\nabla} f(x_t) - \nabla f(x_t)$ and $\xi_{t,i}$ as the $i$-th coordinate of $\xi_t$, then we assume the noise is per-coordinate subgaussian i.e. there exists $\sigma_i > 0$ for $i \in [d]$ such that $\xi_t$ satisfies

$$\mathbb{E}\left[\exp\left(\lambda^2 \xi_{t,i}^2\right)\right] \le \exp\left(\lambda^2 \sigma_i^2\right), \forall |\lambda| \le \frac{1}{\sigma_i}, \forall i \in [d]. \tag{1}$$

Note that $\|\xi_t\|$ being $\sigma$-subgaussian implies that each $\xi_{t,i}$ is also $\sigma$-subgaussian, so coordinate-wise sub-gaussian is more general than standard scalar sub-gaussian noise assumption. Furthermore, when $\|\cdot\|$ is used without explicitly specifying the norm, we assume it is the $\ell_2$ norm $\|\cdot\|_2$. We also use the 0-indexing convention i.e. $[n] := \{0, 1, \ldots, n-1\}$ for integer $n \in \mathbb{N}$.

# 3. Subset-Norm (SN) Adaptive Step Size

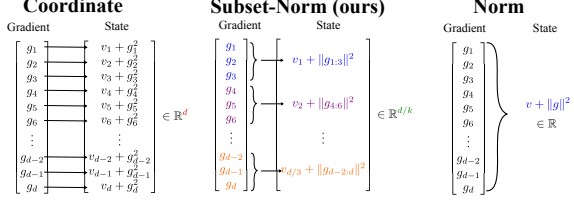

*Figure 2.* AdaGrad variants: Coordinate, Subset-Norm, and Norm. Subset-Norm generalizes Coordinate ($k = 1$) and Norm ($k = d$).

---

**Algorithm 2** SGD with Subset-Norm Adaptive Step Size

---

**Input:** Initial point $x_1 \in \mathbb{R}^d$, base step size $\eta > 0$, partition function $\psi : [d] \to [c]$ that splits the coordinates into $c$ subsets $\Psi_i = \psi^{-1}(i) \subset [d]$, where $\coprod_{i=1}^{c} \Psi_i = [d]$, and $b_{0,i} > 0$ for $i \in [c]$
**for** $t = 1$ **to** $T$ **do**
    Obtain stochastic gradient $\widehat{\nabla} f(x_t)$
    $b_{t,i}^2 = b_{t-1,i}^2 + \left\|\widehat{\nabla}_{\Psi_i} f(x_t)\right\|^2$,    for $i \in [c]$    {Update accumulated gradient norms}
    $x_{t+1,k} = x_{t,k} - \frac{\eta}{b_{t,\psi(k)}} \widehat{\nabla}_k f(x_t)$,    for $k \in [d]$
**end for**

---

We compress the second moment adaptive step size by partitioning parameters into subsets for which they share the same adaptive step size as AdaGrad-Norm (McMahan & Streeter, 2010; Ward et al., 2019). Formally, we need to specify a *partition function* $\psi : [d] \twoheadrightarrow [c]$ that splits the $d$ coordinates into $c$ non-empty subsets $\Psi_i = \psi^{-1}(i) \subset [d]$, where $\coprod_{i=1}^{c} \Psi_i = [d]$. For example, one can pick $\psi(j) = (j/c) \mod k$ to get consecutive equipartitioned subsets $\Psi_i = \{ik, ik+1, \ldots, ik+(k-1)\}$ for some subset-size $k \in \mathbb{N}$ so that $kc = d$.[3]

Given a stochastic gradient $\widehat{\nabla} f(x_t) \in \mathbb{R}^d$ at time $t$ for parameter $x_t$, we denote $\widehat{\nabla}_{\Psi_i} f(x_t) \in \mathbb{R}^k$ to be the subset of the coordinates of the stochastic gradient with respect to the subset $\Psi_i$. For example, given $\psi(j) = (j/c) \mod k$ as above, we have $\left(\widehat{\nabla}_{\Psi_i} f(x_t)\right)_j = \widehat{\nabla}_{ik+j-1} f(x_t)$. Similarly, we can define $\nabla_{\Psi_i} f(x_t) \in \mathbb{R}^{|\Psi_i|}$ to be $\frac{\partial f(x_t)}{\partial x_{\Psi_i}}$.

Now, we define the *subset-norm (SN) adaptive step size* $b_{t,i}$ for subset $\Psi_i$ and the update rule for $x_{t+1}$ (see Figure 2):

$$b_{t,i}^2 = b_{t-1,i}^2 + \left\|\widehat{\nabla}_{\Psi_i} f(x_t)\right\|^2, \text{ for } i \in [c] \tag{2}$$

$$x_{t+1,j} = x_{t,j} - \frac{\eta}{b_{t,\psi(j)}} \widehat{\nabla}_j f(x_t), \text{ for } j \in [d]. \tag{3}$$

Note that choosing $c = d$ and $c = 1$ recovers AdaGrad-Coordinate and AdaGrad-Norm, respectively. We now show a convergence guarantee on arbitrary partitions that will inform us on how to select a good partition strategy.

## 3.1. High-Probability Convergence of Subset-Norm

We show the following high-probability convergence result for the subset-norm adaptive step size:

**Theorem 3.1.** *Suppose that $f : \mathbb{R}^d \to \mathbb{R}$ is $L$-smooth and lower bounded by $f_*$. Given unbiased stochastic gradients $\widehat{\nabla} f(x_t)$ with stochastic gradient noise $\xi_t := \widehat{\nabla} f(x_t) - \nabla f(x_t)$ that is $\sigma_i$-per-coordinate subgaussian for $i \in [d]$. For partitions of the parameters into $c \in \mathbb{N}_+$ disjoint subsets $[d] = \bigcup_{i=0}^{c-1} \Psi_i$ with $\Psi_i \cap \Psi_j = \emptyset$, for $i \ne j$, the iterates $x_t$ given by Algorithm 2 satisfies the following with probability at least $1 - O(c\delta)$ (for failure probability $\delta > 0$)*

$$\frac{1}{T} \sum_{t=1}^{T} \|\nabla f(x_t)\|_2^2 \le G(\delta) \cdot \tilde{O}\left(\frac{\sum_{i=0}^{c-1} \|\sigma_{\Psi_i}\|_2}{\sqrt{T}} + \frac{N(\delta)}{T}\right),$$

*where*

$$G(\delta) := \tilde{O}\left(\sum_{i=0}^{c-1} \|\sigma_{\Psi_i}\|_2^4 + \|\sigma\|_\infty \left(\|\sigma\|_2^2 + c^{3/2}\right) + cL\right),$$

$$N(\delta) := \|\sigma\|_2^2 + \sum_{i=0}^{c-1} \|\sigma_{\Psi_i}\|_2 + Lc.$$

---
[3]We use this strategy in all our implementations for simplicity.

Polylog terms are hidden in Theorem 3.1 for simplicity. The full result, Theorem E.1, and proofs are presented in Appendix E. Theorem 3.1 provides guarantee for all partitions of the parameters into arbitrary disjoint subsets and generalizes AdaGrad-Norm ($c = 1$) and AdaGrad-Coordinate ($c = d$) results. The result is noise-adapted: if $\sum_{i=0}^{c-1} \|\sigma_{\Psi_i}\|_2$ is small enough, the rate becomes the optimal deterministic rate of $O(\frac{1}{T})$ regardless of the base step size $\eta$. The next section explores implications of Theorem 3.1 and strategies for selecting subsets.

### 3.2. Coordinate-Noise Density and Subset-Norm's Improved Dimensional Dependency

Theorem 3.1 presents trade-offs between the partition strategies and the stochastic gradient noise, where we need to balance between the number of subsets $c$ and noise-reduction benefits of parameters-grouping e.g., $\|x\|_2 \le \|x\|_1$.

**Coordinate-noise density** $d^\beta$. To make the intuition above concrete, consider a scenario with various coordinate-noise density rate: fix a rate $\beta \in [0, 1]$, some $d^\beta$ coordinates have noise $\alpha > 0$ while the rest are 0. The rate $\beta$ controls the density of coordinate noise. When $\beta = 0$, only 1 coordinate have noise. When $\beta = 1$, all coordinates have noise. To get a feel for $\beta$'s relationship to the fraction of coordinates containing noise, half the coordinates contain noise when $\beta \approx 0.96$ for $d = 60M$ and $\beta \approx 0.97$ for $d = 10B$ and $\beta \approx 0.98$ for $d = 10^{15}$ (See also Figure 10). See Figure 3 for noise density of LLaMA 60M (details in Appendix C.1). Furthermore, $\alpha$ upper bounds all coordinate noise, i.e. $\|\sigma\|_\infty \le \alpha$, which is common in coordinate-wise analysis (Défossez et al., 2022).

**Derivation of convergence rate given coordinate noise density** $d^\beta$. Given $\beta \in [0, 1]$, we can obtain a concrete expression for the convergence rates of various methods (different subset sizes) from Theorem 3.1. For SGD with Subset-Norm, we consider an *equipartition strategy*, where we divide the coordinates into $c = d^{1-\beta}k$ subsets of size $d^\beta/k$ each with the $d^\beta$ noisy coordinates into just $k$ subsets so that the rest of the $c - k$ subsets have no noisy coordinate. We defer the derivation details to Appendix C.2 and summarize the results in the first row of Table 2.

**Subset Selection.** In Table 2, the equal subset-size partition strategy for Subset-Norm has better dependency on the dimension $d$ when the noise is not completely sparse i.e. $\beta = 0$. Hence, if we expect the actual noise density $\beta$ to be around[4] 0.75 to 0.90, then compressing with a subset size of around $d^{0.45}$ to $d^{0.66}$ is optimal. The dependency on $d$ is

---

[4]Figure 11 shows that overall noise is quite sparse but varies more when limited to a particular layer as in Figure 3. See Section 5.3 for more experiments on subset size selection.

important for modern neural network, since the number of parameters $d$ is typically greater than or on the same order as the total number of iterations $T$.

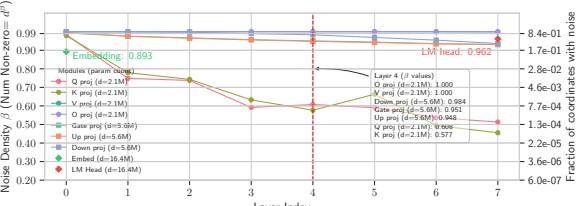

*Figure 3.* Noise density per parameter across layers for LLaMA 60M on pre-training task after 100 steps.

**Subset-size heuristics to avoid additional hyperparameters.** Providing a useful and robust default setting for an algorithm is important to justify claims of reduced costs. We provide a simple partitioning scheme for SN for 2D parameters $p \in \mathbb{R}^{m \times n}$: we simply group along the smaller dimension. For example, if $p$ is of shape $(2048, 1024)$, we group by rows to get a state of size 2048. This is a natural grouping scheme that groups the latent dimension together and aims for the rough $d^{0.45}$ subset size discussed in the previous paragraph. Another simplification is that subset-norm is applied only on *linear* modules, since 2D linear modules makes up the vast majority of parameters in transformers. This means we compress all the attention, MLP, and final LM head weights. This implementation is presented in more details in Appendix A.2. Section 5.3 shows that this heuristic grouping, while simple, is not optimal and can be improved by tuning the subset size, but we opt for simplicity over performance-tuning in our experiments.

**Generic Implementation.** We provide pseudocode for the generic equipartition strategy of Algorithm 2 in Section A.3 that we use for the subset sizes ablations in Section 5.3.

Furthermore, in contrast to methods like AdaFactor or GaLore that are limited to 2D parameters, the generic subset-norm algorithm is coordinate-wise and admits an easy implementation to FSDP (Zhao et al., 2023; Rajbhandari et al., 2020), where parameters are flattened to 1D tensors for efficient communication.

## 4. Subspace-Momentum

Existing algorithmic compression approaches like GaLore (Zhao et al., 2024), GRASS (Muhamed et al., 2024), and FLORA (Hao et al., 2024) project the gradient to a lower dimensional space $\mathbb{R}^k$ for updating the optimizer state via some bounded linear operator $P : \mathbb{R}^d \to \mathbb{R}^k$ such that $P^*P : \mathbb{R}^d \to \mathbb{R}^d$ is a projection, i.e. $(P^*P)^2 = P^*P$, where $P^* : \mathbb{R}^k \to \mathbb{R}^d$ is the adjoint operator of $P$. More

*Table 2.* Algorithms comparison between dimensional dependencies and convergence rates under different coordinate-noise density settings. Given a density rate $\beta$, convergence rates' dimensional dependency are highlighted in red and green to denote the worst and best dependency on the dimension. Note that memory usage of AdaGrad-Coordinate is $O(d)$ while SGD with Subset-Norm (with the partition strategy presented here) is $O(d/k)$, where $k = d^{1.4\beta - 0.6}$ is chosen as an optimal noise dependent subset size.

| Density rate | AdaGrad-Coordinate | AdaGrad-Norm | Subset-Norm (equipartition subsets) |
|---|---|---|---|
| $\beta \in [0, 1]$ | $\tilde{O}\left(d^{1.5+\beta}/\sqrt{T} + d^{2.5}/T\right)$ | $\tilde{O}\left(d^{2.5\beta}/\sqrt{T} + d^{3\beta}/T\right)$ | $\tilde{O}\left(d^{0.3+1.8\beta}/\sqrt{T} + d^{\beta+1}/T\right)$ if $\beta \in [0, 2/3]$ $\tilde{O}\left(d^{0.3+1.8\beta}/\sqrt{T} + d^{1.6\beta+0.6}/T\right)$ if $\beta \in [2/3, 1]$ |
| $\beta = 0$ | $\tilde{O}\left(d^{1.5}/\sqrt{T} + d^{2.5}/T\right)$ | $\tilde{O}\left(1/\sqrt{T} + 1/T\right)$ | $\tilde{O}\left(d^{0.3}/\sqrt{T} + d/T\right)$ |
| $\beta = 0.5$ | $\tilde{O}\left(d^{2}/\sqrt{T} + d^{2.5}/T\right)$ | $\tilde{O}\left(d^{1.25}/\sqrt{T} + d^{1.5}/T\right)$ | $\tilde{O}\left(d^{1.2}/\sqrt{T} + d^{1.5}/T\right)$ |
| $\beta = 0.9$ | $\tilde{O}\left(d^{2.4}/\sqrt{T} + d^{2.5}/T\right)$ | $\tilde{O}\left(d^{2.25}/\sqrt{T} + d^{2.7}/T\right)$ | $\tilde{O}\left(d^{1.92}/\sqrt{T} + d^{2.04}/T\right)$ |
| $\beta = 1$ | $\tilde{O}\left(d^{2.5}/\sqrt{T} + d^{2.5}/T\right)$ | $\tilde{O}\left(d^{2.5}/\sqrt{T} + d^{3}/T\right)$ | $\tilde{O}\left(d^{2.1}/\sqrt{T} + d^{2.2}/T\right)$ |

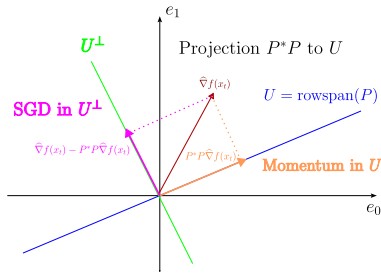

*Figure 4.* Subspace Momentum Illustration.

concretely, given a stochastic gradient $\widehat{\nabla} f(x_t) \in \mathbb{R}^d$ at time $t$, a low-dimensional version $c_t := P\widehat{\nabla} f(x_t) \in \mathbb{R}^k$ is computed that is used to update the states before projecting back to $\mathbb{R}^d$ for update:

$$
\begin{aligned}
m_t &= \beta_1 m_{t-1} + (1 - \beta_1) c_t \\
v_t^2 &= \beta_2 v_{t-1}^2 + (1 - \beta_2) c_t^2 \\
x_{t+1} &= x_t - P^* (m_t/v_t).
\end{aligned}
\tag{4}
$$

This update performs adaptive optimization in the row span $U \subseteq \mathbb{R}^d$ of $P$ when viewed as a linear operator, with $\dim(U) = k$. For example, GaLore (Zhao et al., 2024) utilizes the top $k$ singular vectors of stochastic gradients, and FLORA (Hao et al., 2024) simply projects to a random subspace using dense Gaussian matrices. Due to the optimization operating in a low rank subspace, convergence is not guaranteed unless stronger conditions are assumed.

We propose Subspace Momentum (SM) – presented in Algorithm 3 and illustrated in Figure 4 – where SM guarantees convergence by incorporating the orthogonal complement of $P^*P\widehat{\nabla} f(x_t) \in U$ that lives in the orthogonal complement $U^\perp$ of $U$ (with $U \oplus U^\perp = \mathbb{R}^d$). We can compute the orthogonal complement of $\widehat{\nabla} f(x_t)$ via $\left(\widehat{\nabla} f(x_t) - P^*P\widehat{\nabla} f(x_t)\right) \in U^\perp$.

Subspace Momentum maintains the same memory footprint, $O(k)$, as existing low-rank optimizers. However, SM's update step is full rank: it uses momentum only in

---

**Algorithm 3** SGD with Subspace Momentum (SM)

**Input:** Bounded Linear Operator $P : \mathbb{R}^d \to \mathbb{R}^k$, such that $P^*P$ is a projection, where $P^*$ is $P$'s adjoint.
**for** $t = 1, 2, \ldots, T$ **do**
  Obtain stochastic gradient $\widehat{\nabla} f(x_t)$
  $m_t = \beta_1 m_{t-1} + (1 - \beta_1)P\widehat{\nabla} f(x_t)$    {Momentum in subspace $U = \text{rowspan}(P)$}
  $r_t = \widehat{\nabla} f(x_t) - P^*P\widehat{\nabla} f(x_t)$    {Component in $U^\perp$}
  $x_{t+1} = x_t - \eta(P^*m_t + r_t)$    {Step in both spaces}
**end for**

---

$U := \text{rowspan}(P)$ while performs SGD in $U^\perp$. Unlike joint compression techniques like GaLore (4), SM only affects the momentum term. Hence, SM is modular and fits into the framework of Algorithm 1: there, we can combine it with different adaptive step sizes such as subset-norm.[5]

### 4.1. High-Probability Convergence of Subspace-Momentum

We show that Subspace-Momentum, Algorithm 3, converges with high-probability under the standard assumptions of smoothness and $\sigma$-subgaussian gradient noise:

**Theorem 4.1.** *Suppose that $f : \mathbb{R}^d \to \mathbb{R}$ is L-smooth and lower bounded by $f_*$. Assume unbiased stochastic gradients $\widehat{\nabla} f(x_t)$ with $\sigma$-subgaussian stochastic gradient noise. Then, the iterates $x_t$ given by SGD with Subspace-Momentum (Algorithm 3) with step size $\eta = \min\left\{\frac{1}{2\alpha}; \sqrt{\frac{\Delta_1}{\sigma^2\alpha T}}\right\}$ for $\alpha := \frac{(3-\beta)L}{2(1-\beta)}$ satisfies the following with probability at least $1 - \delta$*

$$
\frac{1}{T}\sum_{i=1}^{T} \|\nabla_t\|_2^2 \leq \frac{8\Delta_1\alpha}{T} + \frac{7\sigma\sqrt{\alpha\Delta_1}}{\sqrt{T}} + \frac{48\sigma^2\log(1/\delta)}{T},
$$

*where $\Delta_1 := f(x_1) - f_*$ is the initial function gap.*

We observe that Theorem 4.1 has a similar rate to vanilla SGD. Unlike adaptive algorithms, we need to know the

---

[5]Section B.9 contains a detailed ablation on different momentum and adaptive step sizes combinations.

problem parameters to get an adaptive convergence rate. The proof is presented in Appendix D.2, where we also provide some intuition for the algorithm.

### 4.2. Subspace Selection, Subspace Switching, and Projection Updates

In our experiments, we use the top-$k$ singular vectors of a stochastic gradient snapshot as our main subspace, similarly to GaLore (Zhao et al., 2024).

Algorithm 3 and the accompanying theory in Section 4.1 are only for a fixed projection. However, from our experiments, we find that performing subspace switching every $G$ steps (as in GaLore) can be beneficial, especially for smaller ranks. Section B.7 contains an ablation studies on this. We incorporate projection updates in our main algorithms by picking a projection update gap and then fully resetting the momentum term to *zero* when we switch (in contrast to GaLore's accumulated statistics when switching subspace).

## 5. Experiments

We evaluate Subset-Norm (SN) and Subspace-Momentum (SM) on LLM pretraining and supervised fine-tuning tasks, where memory is often a bottleneck. We compare against several baselines, with memory estimates given for parameters of size $m \times n$, where we assume WLOG $m \geq n$.

**Baselines.** We consider **AdaGrad** (Duchi et al., 2011), **AdaGradm** where we incorporate momentum 0.9 to AdaGrad, **Adam** (Kingma & Ba, 2014), and **RMSProp** (Tieleman et al., 2012) as standard optimizers. We also consider **GaLore** (Zhao et al., 2024) as a recent memory-efficient method that projects the optimizer states into a low-rank subspace (typically rank $n/4$), using $2(mn/4)$ memory but requiring 6 hyperparameters including subspace rank, projection update frequency, and scaling parameters.

**Our methods.** We incorporate SN and SM to AdaGrad, AdaGradm, Adam and RMSProp. **SN** reduces the adaptive step size (e.g. Adam's second moment term) memory from $mn$ to $m$ for a parameter of size $m \times n$. **SNSM** further compresses the momentum term of momentum methods like Adam and AdaGradm by adding SM with SVD at the cost of additional hyperparameters (See Algorithm 6 for the full implementation used in our experiments). **RMSPropSN** and **AdaGradSN** achieves minimal memory footprint of just $m$ while requiring only 2 hyperparameters.

### 5.1. LLM Pre-Training Experiments

We test our method on the task of pre-training LLaMA models (Dubey et al., 2024; Touvron et al., 2023) on the C4 dataset (Raffel et al., 2023) with a standard setup – details in Appendix A.1. Table 3 presents the main pre-training

results and Table 4 shows the memory footprint[6] of different optimizers across a range of model sizes.

**Additional Baselines.** We provide additional comparisons with FLORA (Hao et al., 2024), LoRA (Hu et al., 2021), and ReLoRA (Lialin et al., 2023) in Table 8. Note that these memory-efficient methods sacrifice performance (over Adam) to save memory while our method, AdamSNSM, achieves the best of both worlds.

#### 5.1.1. DISCUSSIONS

**Subset-Norm (SN) improves upon all existing adaptive methods while reducing memory.** Modifying Adam, AdaGradm, AdaGrad, and RMSProp with the SN adaptive step size not only reduces memory footprint but improves their performance across different scales. Notably, AdaGrad and AdaGradm benefit the most from the SN step size, providing empirical support for the theoretical benefits of SN presented in Section 3.

**Combining Subspace-Momentum (SM) with SN further improves performance while saving additional memory.** Perhaps surprisingly, limiting the use of momentum to a subspace *improves* performance in SN-adaptive step sizes rather than degrading it. Our experiments show that SNSM, combining SN and SM, gives the best performance for the least amount of memory across model sizes. While adding SM introduces additional hyperparameters, Section B.7 suggests that these parameters are not too sensitive.

Furthermore, Section B.8 shows that the choice of the subspace matters i.e. the subspace spanned by a top-$k$ singular vectors of a snapshot of a stochastic gradient seems to be the most beneficial for momentum as opposed to simpler choices like a random subspace. Our current guarantee for SM, presented in Section D, does not yet explain why or when subspace momentum is useful, and theoretical understanding of (EMA style) momentum in stochastic optimization is still limited (Kidambi et al., 2018). We believe this could be related to how momentum is beneficial when noise is low (and harmful when noise is high) and the choice of the subspace could correlate to the amount of gradient noise or optimization landscape that harm or benefit momentum (Wang et al., 2024; Gitman et al., 2019).

**Hyperparameter robustness.** In Table 3, the best learning rate (LR) found via grid search is displayed and is highlighted in red as the best LR changes across scales. This indicates potential sensitivity to tuning for each respective algorithm. We see that Adam requires smaller LR for larger models, but using SN and SNSM does not. AdaGradm seems less sensitive to the base LR overall.

---

[6]The memory footprint is the total parameters in the optimizer states multiplied by 16 bits. See Listing 1 for more details.

*Table 3.* Final perplexity ("Perpl.") along with the number of tokens in parentheses of different optimizers on pretraining LLaMA models task. **Bolded methods** are ours. Columns LR and #TP denote the learning rate and the number of tunable parameters of the corresponding method, respectively. We only tune for the base learning and set other parameters as in previous implementations. The memory column shows the optimizer's states memory consumption given a parameter of shape $m \times n$ with $m \geq n$. Red LR highlights instability.

| Methods | Memory (for $m \times n$) | #TP | **60M** (1.38B) Perpl. | LR | **130M** (2.62B) Perpl. | LR | **350M** (7.86B) Perpl. | LR | **1B** (13.1B) Perpl. | LR |
|---|---|---|---|---|---|---|---|---|---|---|
| Adam | $2mn$ | 3 | 30.46 | 0.005 | 24.60 | 0.005 | 18.67 | 0.001 | 16.00 | 0.0005 |
| **AdamSN** | $mn + m$ | 3 | 29.75 | 0.05 | 22.90 | 0.05 | 17.49 | 0.05 | 14.96 | 0.05 |
| **AdamSNSM** | $rn + m$ | 5 | 29.74 | 0.05 | **22.43** | 0.05 | **16.91** | 0.05 | 14.05 | 0.05 |
| AdaGradm | $2mn$ | 2 | 30.40 | 0.10 | 24.86 | 0.10 | 18.30 | 0.10 | 17.42 | 0.10 |
| **AdaGradmSN** | $mn + m$ | 2 | **29.73** | 2.00 | 22.58 | 2.00 | 17.14 | 2.00 | 14.48 | 2.00 |
| **AdaGradSNSM** | $rn + m$ | 4 | 29.81 | 1.00 | **22.43** | 1.00 | 16.99 | 1.00 | **13.96** | 1.00 |
| AdaGrad | $mn$ | 1 | 37.12 | 0.05 | 25.76 | 0.05 | 18.14 | 0.05 | 15.25 | 0.01 |
| **AdaGradSN** | $m$ | 1 | 29.85 | 2.00 | 24.19 | 1.00 | 17.72 | 1.00 | 14.82 | 1.00 |
| RMSProp | $mn$ | 2 | 35.51 | 0.001 | 25.94 | 0.001 | 20.01 | 0.001 | 17.03 | 0.001 |
| **RMSPropSN** | $m$ | 2 | 34.57 | 0.01 | 25.67 | 0.01 | 18.72 | 0.01 | 15.97 | 0.001 |
| GaLore (Zhao et al., 2024) | $2rn$ | 6 | 34.73 | 0.01 | 25.31 | 0.01 | 18.95 | 0.01 | 16.76 | 0.001 |
| **Rank $r$ / Dimension $m$** | | | 128/512 | | 256/768 | | 256/1024 | | 512/2048 | |

*Table 4.* Optimizer states memory footprint (in GB for BF16 dtype) for different LLaMA models. Our methods, AdamSN, AdamSNSM, and RMSPropSN (RMSPSN), are modifications of Adam and RMSProp (RMSP) to utilize Subset-Norm (SN) and Subspace-Momentum (SM). For GaLore and AdamSNSM, the subspace is of dimension[2] $d/r$, where the memory accounts for additional space for storing the projection matrices.

| Opt. | AdamW | AdamSN | RMSP | GaLore | AdamSNSM | RMSPSN |
|---|---|---|---|---|---|---|
| **Mem.** | $2d$ | $d + \sqrt{d}$ | $d$ | $4d/r$ | $2d/r + \sqrt{d}$ | $\sqrt{d}$ |
| 60M | 0.22 | 0.14 | 0.11 | 0.15 | 0.08 | 0.03 |
| 130M | 0.50 | 0.30 | 0.25 | 0.29 | 0.16 | 0.05 |
| 350M | 1.37 | 0.75 | 0.69 | 0.53 | 0.28 | 0.06 |
| 1B | 4.99 | 2.62 | 2.49 | 1.61 | 0.84 | 0.12 |
| 3B | 10.01 | 5.16 | 5.00 | 2.96 | 1.52 | 0.15 |
| 7B | 25.10 | 13.04 | 12.55 | 7.01 | 2.73 | 0.49 |

*Table 5.* Last and minimum validation perplexity for SFT of LLaMA 7B on the UltraFeedback dataset between Adam, LoRA, and AdamSNSM for 2 different ranks. We also show the wall-clock time and peak memory for batchsize 1 for these optimizers.

| | Adam | LoRA (r=64) | AdamSNSM (r=64) | SNSM (r=32) |
|---|---|---|---|---|
| Last | 2.622 | 2.632 | 2.584 | **2.580** |
| Min. | 2.401 | 2.410 | 2.392 | **2.390** |
| Time (min.) | 266 | 249 | 303 | 301 |
| Memory (GB) | 77.11 | 20.75 | 42.89 | 42.89 |

**Closing the theory-practice gap.** While there is a non-trivial performance gap between Adam and AdaGrad(m) for larger models, using the SN step size closes this gap across scales. This shows that AdaGrad style algorithms can be competitive to Adam when using the SN step size. Interestingly, vanilla AdaGrad seems to perform well as model size increases. This is important because AdaGrad enjoys stronger theoretical understanding than Adam and has one fewer parameter – $\beta_2$ – to tune.

## 5.2. LLMs Supervised Fine-Tuning (SFT) Experiments

We further evaluate on a supervised-fine-tuning task, where we fine-tune a pre-trained LLaMA 7B model on the UltraFeedback dataset (Cui et al., 2024) using the chosen responses with max sequence length of 1024. We train for 1 epoch with linear decay and gradient clipping of 1. Table 5 contains the result with the time and memory of one training epoch on a single A100-80GB GPU. Note SNSM's $r$ denotes the dimension of SM but the optimization is full-rank.

**Discussion.** We observe similar improvement over Adam as in pre-training tasks. Surprisingly, the smaller rank (for momentum) is more beneficial than the larger rank. In contrast to LoRA, since we report peak-memory here, due to the full parameter training of SNSM, the primary memory bottlenecks are gradients and activations. Furthermore, we note that the primary contributor to SNSM's slower wall clock time is the SVD computation on large dimension. We try larger projection update gaps in Table 6 which reduce this cost while maintaining good performance for our methods. Furthermore, we discuss potential more efficient alternatives in Section 5.3 and leave further exploration to future works.

**GLUE Fine-tuning.** Additional results on fine-tuning on GLUE tasks with BERT models are in Appendix B.4.

## 5.3. Ablation Studies

In this section, we present ablation studies on various parameters of SN and SM.

**Subset-Norm's subset size ablation.** While we use a simple scheme to compress the adaptive step size of linear

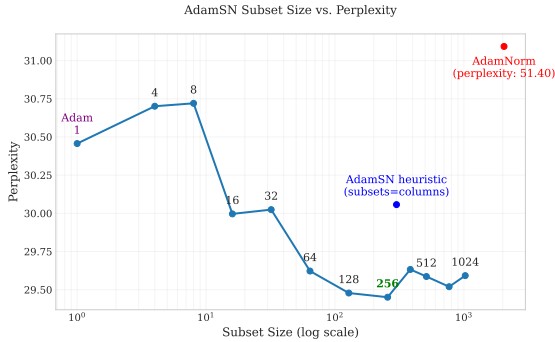

*Figure 5.* Subset size ablation for AdamSN on LLaMA 60M trained for 1.38B tokens (batch size of 512 of max length 256 for 10,000 steps). The higher the subset size, the smaller the memory footprint of the second moment optimizer state.

modules in the previous experiments, Table 2 suggests that there is an optimal subset size that depends on the noise. Figure 5 shows performance for various subset-size selection. Since the step size scales with the subset size, the optimal base LR should be decreased as we decrease the subset size closer towards Adam. We include additional results for 130M model in Figure 8.

While one can use the heuristics discussed on models where linear modules make up the vast majority, for arbitrary models with weights of $d$ elements, we found that a subset size of $\sqrt{d}/2$ is probably a reasonable choice. If more resources are available, the subset size can also be tuned.

**Subspace selection.** While the top-$k$ singular vectors of stochastic gradients gives a subspace with strong performance, performing SVD can be expensive for larger models and storing the dense projection consumes non-negligible memory for large ranks. Gradient-independent projections like random gaussian as in FLORA (Hao et al., 2024) avoids SVD and can save memory by storing the pseudorandom seed (at the cost of recomputating the projection at every step). One can further speed up the random projection by using a fast subspace embedding like the Subsampled-Randomized Hadamard Transform (SRHT) used in the Fast-JL transform (Ailon & Chazelle, 2009). Random projections like SRHT can also be used to approximate SVD (Appx-SVD) computation (Halko et al., 2011) that can be much faster than full SVD. Finally, the cheapest projection is a subspace of random standard bases. Recently, GRASS (Muhamed et al., 2024) explores this idea and tests sampling random bases with large gradient norms. We examine different choices for the subspace and compare their time, space, and performance in Appendix B.8 (Table 13).

**Larger projection update gaps.** Frequently updating the projection map using SVD can be expensive, especially for larger models. Furthermore, updating the projection

*Table 6.* Effects of less frequent subspace update schedule (gap). Compared to Table 3 where the gap is fixed to 200 across all scales.

| Model Size | 60M | 130M | 350M | 1B |
|---|---|---|---|---|
| Gap/Steps (5%) | 200/10K | 1K/20K | 3K/60K | 5K/100K |
| AdamSNSM | 29.84 | 22.71 | 18.43 | 15.28 |
| AdaGradSNSM | 30.28 | 22.76 | 17.02 | 13.90 |
| GaLore | 36.69 | 29.37 | 21.27 | 19.14 |
| *Fixed* Subspace | 10K/10K | 20K/20K | 60K/60K | 100K/100K |
| AdamSNSM | 30.65 | 23.65 | 18.94 | 15.16 |
| AdaGradSNSM | 31.43 | 24.85 | 18.04 | 14.62 |
| GaLore | 37.95 | 26.63 | 21.49 | 27.11 |

every 200 steps can be arbitrary. In Table 6, we examine more structured schedules: (1) updating every 5% of the total training steps (corresponding to 200/10K steps for the 60M model) and (2) only using a fixed subspace at the start. Compared to Table, 3 where a fixed gap of 200 is used across scales, we see SNSM's performance stay relatively similar when we increase the update gap to 5% of the total training steps, whereas GaLore's performance suffers more.

*Table 7.* Fixed Subspace Choices on LLaMA 60M. We examine GaLore and SNSM with top-$k$ singular vectors projections (SVD) and random subspaces (Random) using dense gaussian projections.

| | GaloreSVD | GaloreRandom | SNSM+SVD | SNSM+Random |
|---|---|---|---|---|
| Perplexity | 37.95 | 38.23 | 30.65 | 40.15 |

Interestingly, for fixed subspace (100% gap), GaLore still achieves decent performance even though the optimization only happens in a small subspace up until the 1B model, where the training stops improving after 50K/100K steps. In Table 7, we see that a random subspace seems to work decently well too. This suggests that a majority of progress can be made in a small subspace in smaller models. In contrast, this is not the same for restricting momentum to a subspace. Furthermore, we notice that there are training loss spikes at the times when we switch subspace for Ga-Lore that impacts training with 5% gap, most likely due to incompatible optimizers' statistics between subspaces. This could explain why GaLore's 100% gap performs similarly or even better than 5% gap for certain run. Finally, we note that AdaGradSNSM performs the best here with the larger gaps as the dimension increases.

**Additional experiments and ablations.** We provide additional experiments and detailed ablations on wall-clock time speedup, peak-memory savings, the effect of clipping, batch sizes, random seeds, combinations between adaptive step sizes and momentum, and more in Appendix B.

## 6. Related Works

As model sizes grow, memory-efficient training techniques have become crucial. Following up on AdaFactor (Shazeer & Stern, 2018), low-rank methods like Galore (Zhao et al., 2024), LoRA (Hao et al., 2024), and ReLORA (Lialin et al., 2023) reduce memory usage by approximating large weight matrices with low-rank representations. Projection-based approaches, such as GRASS (Muhamed et al., 2024) and FLORA (Hao et al., 2024), compress gradients or combine low-rank ideas with projections to reduce memory requirements. Recently, AdaMeM (Vyas et al., 2024a) proposes to incorporate the orthogonal subspace to the AdaFactor optimizer; this is related to but different from our simpler SM algorithms, where we use subspace decompositions to decouple the momentum and SGD. BAdam (Luo et al., 2024), a block coordinate descent method that utilizes Adam as an inner solver, has been proposed for fine-tuning large language models. In contrast to our proposed methods, these methods are largely heuristic-driven and often lack convergence guarantees under standard assumptions. On the other hand, methods like SM3 (Anil et al., 2019), which uses subset (cover) statistics to show convergence in online learning, and MicroAdam (Modoranu et al., 2024), which provides convergence guarantees for a gradient compression scheme with error correction, offer theoretical guarantees.

Additional approaches to reducing memory during training include optimizer quantization (Li et al., 2024a; Dettmers et al., 2021; 2024), attention computation compression/optimization (Wu et al., 2022; Dao et al., 2022; Dao, 2023; Shah et al., 2024), activation checkpointing (Chen et al., 2016), and distributed training (Rajbhandari et al., 2020). For inference, compression techniques are also actively being explored (Sakr & Khailany, 2024; Dettmers et al., 2022; Xiao et al., 2024; Lin et al., 2024; Frantar et al., 2023). These are orthogonal directions to our work and can be combined. Another orthogonal direction is approximated second-order optimization, where one aims to approximate the Hessian preconditioner using only first-order information in order to achieve faster convergence. Some works in this area include (Gupta et al., 2018; Liu et al., 2023a; Vyas et al., 2024b). These methods typically demonstrate faster training but at the cost of super-linear memory and additional computational overhead.

Convergence analysis of non-convex optimization methods has seen significant progress, with recent works providing convergence proofs for adaptive algorithms like Adam (Li et al., 2024b; Défossez et al., 2022). Numerous studies have explored convergence properties of various adaptive and stochastic gradient methods (Chen et al., 2018; Défossez et al., 2022; Ene & Nguyen, 2021; Liu et al., 2023c;b; Ward et al., 2019; Zou et al., 2019; Reddi et al., 2018; Nesterov, 1983), while lower bound analyses (Arjevani et al., 2023)

have highlighted fundamental limits in non-convex optimization. Here, obtaining convergence results for EMA updates (Adam style) for subset-norm and under further relaxed assumptions like affine smoothness (Wang et al., 2023; Attia & Koren, 2023), affine noise (Hong & Lin, 2024; Faw et al., 2022), heavy-tailed noise (Zhang et al., 2019; 2020; Nguyen et al., 2023a;b) are of great interest.

**Comparison with Adam-mini.** Very recently, Adam-mini (Zhang et al., 2024) also uses shared step sizes as Subset-Norm; however, the partition strategy is quite different from ours. While Adam-mini also employs a grouping strategy for the adaptive step size, it is primarily motivated empirically and lacks a general grouping strategy for general parameters. In contrast, our theory results show that grouping by noise magnitude leads to improvement. In experiments, our AdamSNSM uses less memory than Adam-mini, due to the fact that Adam-mini uses full momentum while we use momentum only in a subspace (which outperforms full momentum in many cases given a good choice of subspace). Furthermore, in terms of perplexity, Adam-mini performs very closely to AdamW while our methods outperform Adam (which performs similarly to AdamW) on a range of language tasks and model sizes.

## 7. Conclusion and Future Works

In this paper, we introduce two principled optimizer states' memory reduction methods —Subset-Norm (SN) and Subspace-Momentum (SM)—designed to address the high memory costs associated with adaptive optimizers in large-scale deep learning. SN and SM achieve memory savings without compromising performance and admit high-probability convergence guarantees under relaxed assumptions. Extensive experiments pre-training and fine-tuning LLMs validate our methods' effectiveness and efficiency.

**Future works.** Promising directions include exploring SN and SM on additional domains like Reinforcement Learning, where high memory and high noise are also bottlenecks when scaling up to large models. Fully generalizing our methods involves developing general projections (beyond matrix decomposition) for SM on higher order tensors for use on additional architectures like CNNs. A more in-depth investigation to more optimal subset partition strategies for SN is also an interesting open question, since our analysis in Section 3.2 only applies to equipartition subsets. Further theoretical understanding for SM's subspace dependency for improving the subspace selection for SM is desirable. Furthermore, the convergence of SNSM is still unknown. Finally, the benefits of momentum in stochastic optimization in general is still a mystery; using SM to study effects of momentum on particular subspaces could open doors to obtain a more fine-grained understanding for why or when momentum helps.

## Acknowledgement

This work was supported by NSF CCF 2311649. We thank Alina Ene, Themistoklis Haris, and Duy Nguyen for insightful discussions and Hieu Nguyen for assistance with the experiments. This work was completed in part using the Discovery cluster, supported by Northeastern University's Research Computing team.

## Impact Statement

This paper presents work whose goal is to advance the field of Machine Learning. There are many potential societal consequences of our work, none which we feel must be specifically highlighted here.

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

---

**Algorithm 4** Adam-Subset-Norm with a Simple Partitioning Scheme

---

**Input:** Learning rate $\eta$, EMA parameters $\beta_1$ and $\beta_2$, $\epsilon > 0$, optional weight decay $wd \geq 0$

**for** each $p \in \mathbb{R}^{m \times n}$ in params **do**

  grad $\leftarrow p$.grad

  $r \leftarrow 0$ if $m \geq n$ else $1$

  $k \leftarrow p$.shape$[r]$             {Set $k = m$ if $r = 0$, else $k = n$}

  gradN $\leftarrow$ grad.norm(dim=$1 - r$) $\in \mathbb{R}^k$       {Subset norm}

  $m \leftarrow \beta_1 m + (1 - \beta_1) \cdot$ grad $\in \mathbb{R}^{m \times n}$

  $v \leftarrow \beta_2 v + (1 - \beta_2) \cdot$ gradN$^2 \in \mathbb{R}^k$      {Omitting bias correction terms}

  $p \leftarrow p + \eta \frac{m}{\sqrt{v} + \epsilon}$            {Broadcast division}

  $p \leftarrow p - \eta \cdot wd$             {Weight decay}

**end for**

---

## A. Experimental Details

In this section, we provide hyperparameters details, implementation details (pseudocode), and other practical considerations.

### A.1. Experimental details

All of our pre-training experiments are conducted on NVIDIA RTX4090/3090 GPUs. Unless specified otherwise, we run all experiments on BF16 format, weight decay of 0, gradient clipping of 1.0, cosine learning rate decay to 10% of the max learning rate with 10% linear warmup steps, and batch size of 512 (similarly to (Zhao et al., 2024) and (Touvron et al., 2023; Dubey et al., 2024)).[7] For all our experiments, we use the default $(\beta_1, \beta_2) = (0.9, 0.999)$ and only tune for the base learning rate within a grid of $\{0.5, 0.1, 0.05, 0.01, 0.005, 0.001\}$.[8] We train for 1.38B, 2.62B, 7.86B, and 13.1B tokens for models of sizes 60M, 130M, 350M, and 1B parameters, respectively, following (Zhao et al., 2024) and matches roughly the scaling laws in (Hoffmann et al., 2022).

For GaLore, we use the same hyperparameters as in (Zhao et al., 2024), where we use rank 128/512, 256/768, 256/1024, and 512/2048 for the 60M, 130M, 350M, and 1B models, respectively (Table 2 of (Zhao et al., 2024)).[9] For AdamSNSM, we use the same ranks and projection update gap (of 200) as GaLore for all models.[10] However, we do not tune for an additional scaling parameter unlike GaLore, and we compresses the LM head (final linear layer) with SN and SM also.[11]

### A.2. Adam-Subset-Norm Implementation

Algorithm 4 presents the pseudocode for Adam-Subset-Norm as mentioned in Section 3.2 where we partition the coordinates (for each parameter) into subsets of equal sizes.

### A.3. Generic Subset-Norm Adaptive Step Size Implementation

The heuristic implementation in Section A.2 is simple and does not require any tuning. However, to modify existing algorithms to work with arbitrary subsets, one could utilize reshape as in Algorithm 5 as an example.

### A.4. AdamSNSM Implementation Details

Algorithm 6 provides the pseudocode and implementation details for the version of AdamSNSM with SVD subspace momentum and heuristics subset-norm (as described in Section 3.2) used in our experiments.

---

[7]Note that these addition improve the performance for all baselines. See Appendix B.10.

[8]Except AdaGradSNm where we find higher learning rates in $\{0.5, 1, 2, 5\}$ to be better. We tune the lr on the 60M model and use the same learning rate for the larger model, where the base learning rate is only reduced if the method fails to converge.

[9]Note that our reproduced results for GaLore and baselines are similar to (Zhao et al., 2024).

[10]Note that a smaller gap is more expensive than a larger gap. Our experiments below show that we can increase the projection update gap without much performance loss. If data is not limited, one could use a larger gap to speed up training. However, if data is limited, then a smaller gap to converge in fewer tokens is potentially more desirable.

[11]Existing methods typically do *not* compress the embedding layer and final LM head, while our methods seem robust to this choice. Compressing these layers save additional memory.

---

**Algorithm 5** Generic Subset-Norm Adaptive Step Size Update Rule (PyTorch-y Notation)

---

**Input:** Parameter $P \in \mathbb{R}^d$, step size $\eta > 0$, $\beta, \epsilon > 0$, and partition size $k$ such that $k$ divides $d$

$R \leftarrow (\nabla P).\text{reshape}(d/k, k)$                 {Reshape gradient into shape $\frac{d}{k} \times k$}

$V \leftarrow \beta V + (1 - \beta) \cdot ((R\text{**}2).\text{sum}(\text{dim=1})) \in \mathbb{R}^{d/k}$     {Update state $V$ via subset norm reduction on dim 1}

$U \leftarrow \frac{R}{\sqrt{V}+\epsilon} \in \mathbb{R}^{d/k \times k}$            {Broadcast addition and division for update step}

$P \leftarrow P - \eta \cdot U.\text{view}(d)$          {Reshape $U$ back to $\mathbb{R}^d$ and update $P$}

---

**Algorithm 6** AdamSNSM with Subspace Momentum via Top-$k$ Singular Vectors from SVD

---

**Input:** Learning rate $\eta$, rank $k$, update gap $G$, momentum parameters $\beta_1, \beta_2 \in (0, 1)$, and stability parameter $\epsilon$

**for** $t = 1, \ldots, T$ **do**

     Obtain stochastic gradient $g_t \in \mathbb{R}^{m \times n}$       {WLOG, assume $m \geq n$}

     **if** $t \mod G = 0$ **then**

         $U, S, V = \text{SVD}(g_t)$       {Compute singular value decomposition}

         $P = U[:, : k] \in \mathbb{R}^{m \times k}$       {Extract top $k$ singular vectors}

     **end if**

     $m = \beta_1 m + (1 - \beta_1) P^T g_t \in \mathbb{R}^{k \times n}$       {Update subspace momentum}

     $r = g_t - PP^T g_t$       {Compute orthogonal SGD component}

     $s = \text{sum}(g_t, \text{dim} = 1) \in \mathbb{R}^n$       {Sum all columns for subset-norm heuristic}

     $v = \beta_2 v + (1 - \beta_2) s^2 \in \mathbb{R}^n$       {EMA of subset-norm}

     $x_t = x_{t-1} + \eta \frac{Pm+r}{\sqrt{v}+\epsilon}$       {Update with subspace momentum and subset-norm step size}

**end for**

---

## A.5. Measuring Memory Footprint of Optimizers

In PyTorch, we can obtain the number of parameters in optimizer states using the code in Listing 1.

## A.6. Peak memory measurement during training for different optimizers

We measure peak memory consumption directly via running nvidia-smi in Figure 6 while training as oppose to controlled measurement as in Table 4. Note that these peak measurements incur additional memory from gradient computation and algorithms' overhead.

```python
def get_optimizer_state_size(optimizer) -> Tuple[int, Dict[str, int]]:
    total_state_size = 0
    state_size_breakdown = {}
    for group in optimizer.param_groups:
        for p in group['params']:
            state = optimizer.state[p]
            for state_key, state_value in state.items():
                if torch.is_tensor(state_value):
                    if state_value.numel() == 1:
                        # we do not count singleton
                        continue
                    total_state_size += state_value.numel()
                    if state_key not in state_size_breakdown:
                        state_size_breakdown[state_key] = 0
                    state_size_breakdown[state_key] += state_value.numel()
    return total_state_size, state_size_breakdown
```

*Listing 1.* PyTorch function to calculate optimizer state size

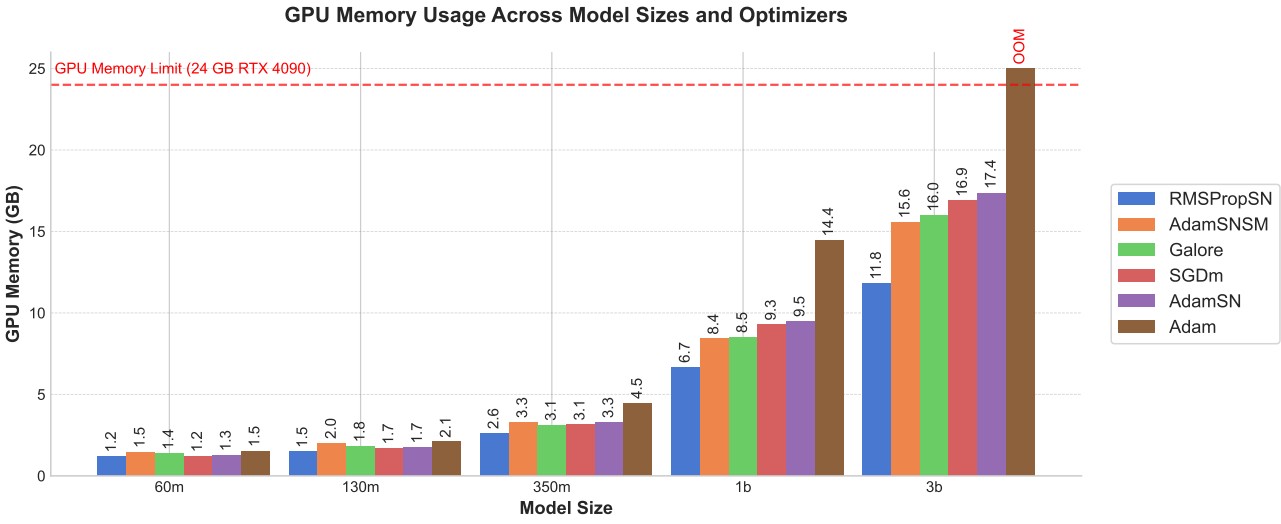

*Figure 6.* Peak GPU Memory Usage (Gb) for various model sizes, obtained with batch size 1 and activation checkpointing to measure the optimizer state footprint.

## B. Additional Experiments and Ablation Studies

### B.1. Additional Comparisons with Memory Efficient Optimizers for Pre-Training LLaMA Models

Table 8 extends Table 3 to compare against other recent methods.

*Table 8.* Additional comparisons with other memory efficient optimizers

| Method | LLaMA 60M | LLaMA 130M | LLaMA 350M | LLaMA 1B |
|---|---|---|---|---|
| AdamW | 30.46 | 24.60 | 18.67 | 16.00 |
| AdamSNSM (**ours**) | 29.74 | 22.43 | 16.91 | 13.96 |
| GaLore (Zhao et al., 2024) | 34.73 | 25.31 | 20.51 | 16.76 |
| FLORA (Hao et al., 2024) | 32.52 | – | 23.69 | – |
| LoRA (Hu et al., 2021) | 34.99 | 33.92 | 25.58 | 19.21 |
| ReLoRA (Lialin et al., 2023) | 37.04 | 29.37 | 29.08 | 18.33 |

### B.2. Wall-clock speedup and peak memory

We provide the per iteration time, peak memory (via nvidia-smi), and time to Adam's val perplexity after 100K steps for the 1B model for each method on a 2x4090 machine with the same setup as in Table 3 (seq length 256, total batch size 512, micro batchsize 16) in Table 9.

*Table 9.* Per iteration time, peak memory (via nvidia-smi), and time to Adam's val perplexity after 100K steps for the 1B model for each method on a 2x4090 machine with the same setup as in Table 3 (seq length 256, total batch size 512, micro batchsize 16) in Table 9

| | Adam | AdamSNSM (Gap=5000) | AdamSNSM (Gap=200) | AdamSN | GaLore (Gap=200) |
|---|---|---|---|---|---|
| Time for 1K iters | 7426 s | 7465 s | 7624 s | 7399 s | 7827 s |
| Time per iteration | 7.43 s/it | 7.47 s/it | 7.62 s/it | 7.39 s/it | 7.83 s/it |
| Time to perplexity $< 16$ | $\sim$206.4 hrs (100K iters) | $\sim$136.9 hrs ($<$66K iters) | $\sim$101.6 hrs ($<$48K iters) | $\sim$118.9 hrs ($<$58K iters) | $>$217 hrs ($>$100K iters) |
| Peak mem | 21.554 GB/GPU | 16.642 GB/GPU | 16.642 GB/GPU | 19.193 GB/GPU | 18.187 GB/GPU |

### B.3. Vision Tasks

**Diffusion Transformers.** While our main focus is on large models that are more typical to language models where memory is often a bottleneck, vision models are also increasing in size. Hence, we conduct further evaluations using the DiT-L/2 model (458M)[12] on a setup with batch size 2048, image size 64, and 8×A6000 GPUs. We compared our method (SNSM) with Adam. As shown in Table 10, SNSM outperforms Adam in FID similarly to LLM tasks.

*Table 10.* FID scores over training iterations for the DiT-L/2 model (458M parameters) on $64 \times 64$ images.

| FID / Iter | 200k | 300k | 400k | 500k | 700k |
|---|---|---|---|---|---|
| Adam | 56.69 | 56.63 | 40.69 | 41.15 | 39.61 |
| AdamSNSM | 66.76 | 66.31 | 34.05 | 32.31 | 32.26 |

**CIFAR10 and CIFAR100.** We further evaluate Adam, AdamSN, and AdamSNSM (rank 64 and no update gap) by training `vit_base_patch16_224`[13] (Dosovitskiy et al., 2020) (around 85M params) from the `timm` library[14] on the CIFAR10 and CIFAR100 (Krizhevsky, 2009) datasets for 10 epochs with a batch size of 64 and weight decay 0 on a 2x4090 machine. We tune the lr across {1e-3, 5e-3, 1e-4, 5e-4, 5e-5, 1e-5} grid for all methods. The results are shown in Table 11.

*Table 11.* Performance Comparison of Optimizers on Vision Transformers for CIFAR10 and CIFAR100

| Best val accuracy (10 epochs) | Adam | AdamSN | AdamSNSM (r=64, g=1000) |
|---|---|---|---|
| CIFAR100 | 43.30% | 45.20% | 45.60% |
| CIFAR10 | 69.02% | 69.18% | 71.21% |
| **Peak Mem (bs 64)** | 9.288GB | 8.886GB | 8.878GB |

These preliminary experiments show promising results for the application of our methods to vision tasks where models are becoming larger.

### B.4. Fine-tuning on GLUE Tasks

Table 12 presents results for fine-tuning on GLUE dataset for various methods. The SN step size maintains good performance while reducing the memory footprint.

### B.5. AdaGrad, AdaGrad-Norm, and AdaGrad-Subset-Norm

We examine the subset-norm step size for AdaGrad in Figure 7. We again see that subset-norm is slightly better than the full coordinate version while using a lot less memory. This is consistent with our observations for Adam and RMSProp when we replace the standard coordinate-wise step size with the subset-norm adaptive step size.

---

[12] https://github.com/facebookresearch/DiT
[13] https://huggingface.co/timm/vit_base_patch16_224.augreg2_in21k_ft_in1k
[14] https://timm.fast.ai/

*Table 12.* Performance metrics across GLUE tasks. QQP, RTE, SST-2, MRPC, STSB, QNLI, and MNLI use accuracy as the metric, while CoLA uses the Matthews correlation coefficient. The **best** and runner-up results for each task and the average score are highlighted.

| Method | QQP | RTE | SST2 | MRPC | STSB | QNLI | MNLI | COLA | Avg |
|---|---|---|---|---|---|---|---|---|---|
| Adam | **92.0** | 77.9 | 94.9 | 89.2 | 90.5 | 93.0 | **87.6** | **65.4** | 86.3 |
| GaLore ($r = 4$) | 90.9 | 79.4 | **95.2** | 88.7 | **90.8** | 92.4 | 86.9 | 61.9 | 85.8 |
| RMSProp | 91.9 | 79.4 | **95.2** | **91.4** | 90.3 | 92.8 | **87.6** | 65.1 | **86.7** |
| RMSPropSN | 91.9 | **80.1** | 95.1 | 90.0 | 90.7 | **93.1** | 87.5 | 63.8 | 86.5 |
| AdamSN | 91.2 | 74.4 | 94.5 | 89.5 | 90.4 | 92.0 | 86.7 | 64.4 | 85.4 |

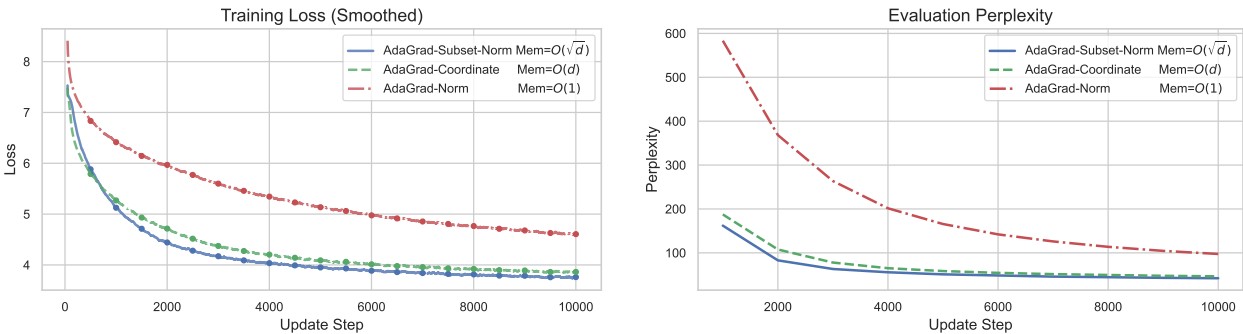

*Figure 7.* Pretraining LLaMA 60M on the C4 dataset for AdaGrad variants. Memory consumption estimate as a function of parameter count $d$ is shown in the legend.

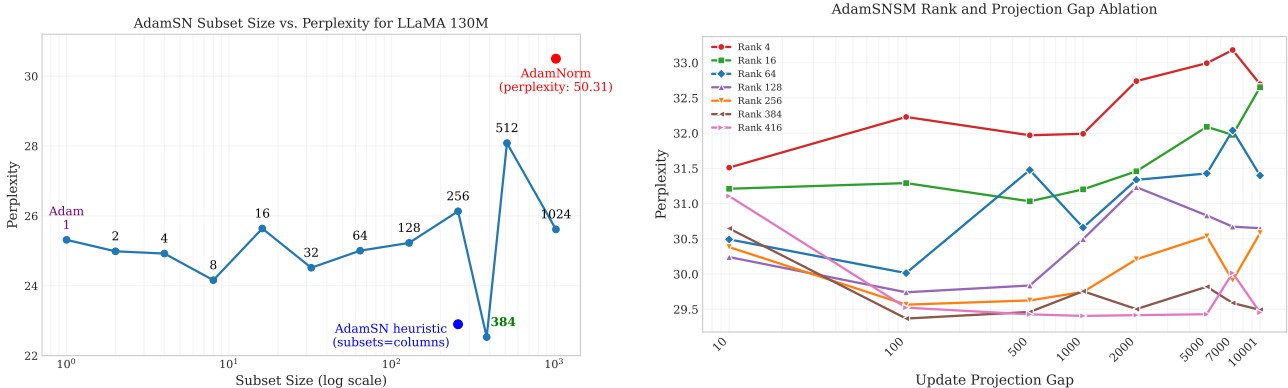

*Figure 8.* Subset size ablation for AdamSN on LLaMA 130M trained for 2.62B tokens (batch size of 512 of max length 256 for 20,000 steps). The higher the subset size, the smaller the memory footprint of the second moment optimizer state.

*Figure 9.* Rank and gap ablation for AdamSNSM on LLaMA 60M for 10,000 steps. The lower the rank, the less memory consumption used by the momentum state. The higher the projection gap, the less SVD computation is performed which is cheaper.

### B.6. Additional Subset-Size Experiments for 130M model

We provide additional subset-size experiments similar to the ones in Section 5.3 for LLaMA 130M in Figure 8.

### B.7. Subspace-Momentum Rank and Gap Ablations

**Rank and gap ablations.** We examine the impact of varying rank and update gap of subspace momentum, similarly to (Zhao et al., 2024), in Figure 9. There, we see that the higher the rank, the better the results. For the update gap, it seems like there is an optimal choice. However, due to the SVD computation, a larger gap will be cheaper than a more frequent gap.

### B.8. Subspace-Momentum Projection Choice Ablations

**Projection types.** Table 13 tests different choices for projection in SM discussed in Section 5.3. Note that for memory storage, SVD, Random Projection via dense Gaussian projection (Gaussian), and Approximated-SVD (Appx-SVD) need to store the $r \times n$ projection matrix (unless we recompute at every step). The remaining methods only need to store the indices for sampling and/or the random seed to regenerate any random choices.

Note that the choice of the projection is important as some projections are more computationally and memory expensive than other, although trading other qualities for given the cost. Simple projections like selecting a subset of coordinates for momentum (Subset-Momentum) are not only faster but enables simple distributed training like FSDP unlike more complex subspace selection mechanism that requires additional priors about the parameters (shape, low-rank, etc.) that might not

*Table 13.* Different projections selection for Subspace-Momentum and validation perplexity. All methods are evaluated on LLaMA 60M with rank 128/512 and a projection update gap of 200. Time and space rows denote time and space to compute and store the projection.

| AdamSNSM's projection | SVD | Approx-SVD | Gaussian | SRHT | Top-$k$ | Random-rows | OPCA | Oja |
|---|---|---|---|---|---|---|---|---|
| Time (for $m \times n$) | $O(mn^2)$ | $O(mn \log k + kn^2)$ | $O(kn)$ | $O(\max(m,n))$ | $O(mn)$ | $O(k)$ | $O(kn)$ | $O(kn)$ |
| Space (for rank $k$) | $O(kn)$ | $O(kn)$ | $O(kn)$ | $O(k)$ | $O(k)$ | $O(k)$ | $O(kn)$ | $O(kn)$ |
| Validation Perplexity | 29.74 | 31.51 | 42.48 | 33.33 | 31.42 | 33.17 | 29.63 | 30.69 |

always satisfied.

**Online $k$-PCA and Streaming $k$-PCA for Up-to-date Subspace.** Computing subspace from stochastic gradient snapshots can be noisy. Recently, (Liang et al., 2024) proposes a formulation of online-PCA to handle the problem of staled top-$k$ components as the stochastic gradients evolve. We test this algorithm in the OPCA column. Another natural algorithm to ensure the top-$k$ components stay up-to-date is Oja's algorithm for streaming $k$-PCA (Huang et al., 2021). We also test this algorithm in Table 13. While we can maintain up to date projection using these schemes, more frequent updates suffer from the same issue of transferring optimization statistics from one subspace to another. We only test for not resetting the statistics in this setting and leave additional investigation for future works. Furthermore, these schemes are more expensive computationally due to additional computation requirement at every step. OPCA further uses Adam for inner optimization which incurs additional memory.

### B.9. Step Sizes and Momentum Choices Full Ablations

We investigate various combinations of momentum and adaptive step size approaches in Table 14. For adaptive methods, we compare EMA, which uses exponential moving average to accumulate the second moment ($v_t^2 = \beta v_{t-1}^2 + (1-\beta)g_t^2$), with AdaGrad's cumulative accumulation approach ($b_t^2 = b_{t-1}^2 + g_t^2$). Methods with the SN suffix utilize subset norm for parameter grouping, contrasting with per-coordinate approaches that are standard. While EMA momentum follows the standard momentum implementation, subspace momentum employs a reduced rank approximation with rank 128 for this model size.

*Table 14.* Different combinations of momentum (columns) and adaptive step-size (rows) and the effect of the learning rate schedule on each combination (cosine learning rate decay schedule with warmup "coslr" or constant learning rate "lr."). Memory footprint for each adaptive step size and/or momentum are shown. Green and red highlight runs with perplexity below 30 and above 50 respectively.

| Final eval perplexity (lr) LLaMA 60M for 1.31B tokens | No momentum Mem = 0 | EMA momentum Mem = $m \cdot n$ | Subspace momentum Mem = $\max(m, n) \cdot$ rank |
|---|---|---|---|
| **No Adaptive Step-size** Mem = 0 | SGD 86.60 (coslr=1e-3) 100.04 (lr=1.0) | SGDm 55.76 (coslr=1e-3) 56.07 (lr=1.0) | SGD+SM 89.97 (coslr=1e-3) 213.21 (lr=5e-4) |
| **EMA Coordinate** Mem = $m \cdot n$ | RMSProp 35.01 (coslr=1e-3) 36.46 (lr=5e-4) | Adam 30.46 (coslr=5e-3) 33.47 (lr=1e-2) | AdamSM 32.34 (coslr=1e-3) 32.25 (lr=5e-4) |
| **EMA Subset-Norm** Mem = $\max(m, n)$ | RMSPropSN 34.86 (coslr=1e-2) 34.57 (lr=1e-2) | AdamSN 29.75 (coslr=5e-2) 33.69 (lr=1e-2) | AdamSNSM 29.74 (coslr=5e-2) 32.49 (lr=1e-2) |
| **AdaGrad Coordinate** Mem = $m \cdot n$ | AdaGrad 37.12 (coslr=5e-3) 46.47 (lr=5e-4) | AdaGradm 31.48 (coslr=5e-2) 43.99 (lr=1e-2) | AdaGradSM 30.99 (coslr=5e-2) 41.32 (lr=5e-4) |
| **AdaGrad Subset-Norm** Mem = $\max(m, n)$ | AdaGradSN 33.19 (coslr=5e-3) 41.23 (lr=0.1) | AdaGradSNm 29.73 (coslr=5e-3) 44.98 (lr=0.1) | AdaGradSNSM 29.81 (coslr=5e-3) 40.11 (lr=0.1) |

| Method | 60M (no clipping) | 60M (with clipping) | 130M (no clipping) | 130M (with clipping) |
|---|---|---|---|---|
| Adam | 30.58 | 30.46 | 25.07 | 25.07 |
| AdamSN | **30.06** | **29.75** | 23.54 | 22.89 |
| GaLore | 34.91 | 34.73 | 25.43 | 25.31 |

*Table 15.* Pre-training LLMs ablation experiments for gradient clipping. We compare validation perplexity between LLaMA 60M and 130M with and without clipping. We use the same hyperparameters as in Section A.1 but just add clipping. .

*Table 16.* Batch size ablation for various optimizers along with optimal learning rate.

| Batch size | Adam | | GaLore | | AdamSN | | AdamSNSM | |
|---|---|---|---|---|---|---|---|---|
| | Perpl. | LR | Perpl. | LR | Perpl. | LR | Perpl. | LR |
| 1024 | 27.94 | 0.005 | 32.75 | 0.01 | **27.68** | 0.05 | 28.02 | 0.05 |
| 512 | 30.46 | 0.005 | 34.73 | 0.01 | 29.75 | 0.05 | **29.74** | 0.05 |
| 256 | 36.65 | 0.001 | 44.71 | 0.001 | 37.03 | 0.001 | **32.82** | 0.05 |
| 128 | 41.72 | 0.001 | 49.75 | 0.001 | 42.04 | 0.001 | **36.82** | 0.05 |

**Discussions.** From Table 14, Subset norm (SN) step sizes consistently outperform coordinate-wise implementations while requiring less memory. Adaptivity proves crucial for optimization effectiveness, where the first row without adaptivity perform consistently poorly. The addition of momentum is beneficial in all configurations while SM is more beneficial for adaptive step sizes. The impact of learning rate scheduling is also evident across configurations, with cosine decay consistently outperforming constant learning rates. Notably, we observe varying degrees of learning rate sensitivity: adaptive methods demonstrate greater robustness to learning rate selection, while non-adaptive methods require more precise tuning.

### B.10. Gradient Clipping

Gradient clipping is standard in training LLMs for many open source models like LLaMA, DeepSeek, OPT, etc. (DeepSeek-AI et al., 2024; Touvron et al., 2023; Workshop et al., 2022; Zhang et al., 2022; Chowdhery et al., 2023; Ding et al., 2023). Clipping has a strong connection to stochastic gradient noise being *heavy-tailed* (Zhang et al., 2019) and many theoretical results have been shown to suggest some form of clipping is beneficial when the noise could follow a heavy-tail distribution(Cutkosky & Mehta, 2021; Gorbunov et al., 2020; Li & Liu, 2022; Nguyen et al., 2023b;a). We present the results with clipping equal to 1.0 for each method in Table 15.

In Table 15, we see that gradient clipping indeed helps most of the methods achieve slightly better perplexity. In our experiments, we notice that adding some form of gradient clipping produces more stable training.

### B.11. Batch Sizes and Random Seeds

**Fixed number of steps.** We measure the impact of different batch sizes on pre-training LLaMA 60M for 10,000 steps in Table 16.[15] We use the same configuration as in other experiments. Typically, smaller batch sizes require smaller learning rates, but curiously, AdamSNSM seems to be stable with the choice of learning rates. Even more interestingly, AdamSNSM's final performance seems to be affected less by the smaller batch size as opposed to other methods, especially GaLore.

**Fixed data quantity.** In the previous section, we compare the performances on different batch sizes fixing the same number of steps. In this section, we fix the amount of data to 1.3B tokens for pre-training LLaMA 60M. Hence, adjusting the batch size would also adjust the number of steps. Table 17 contains the result where SNSM shows consistently better performance than Adam across different batch sizes.

**Random seeds.** Throughout our experiments, we fix the random seed for all runs within a same table. In Table 17, we investigate the effects of random seeds by running each batch size on 3 random seeds and report the mean and standard deviation. We see that SNSM has better variance than Adam for many batch sizes overall. We also examine the random

---

[15]This reduces the amount of total tokens trained. However, we only compare optimizers against one another. To compare the same optimizer against different batch sizes, one should train for the same amount of tokens.

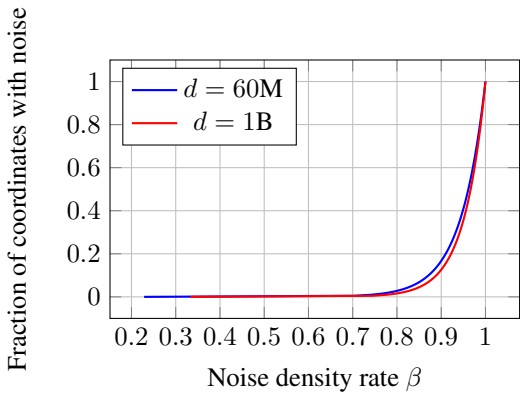

*Figure 10.* Fraction of coordinates with noise over noise density rate.

variation on the 130M model in Table 18.

*Table 17.* Mean and standard deviation (in parentheses) evaluation perplexities of Adam and AdamSNSM optimizers when pretraining LLaMA 60M for 1.3B tokens over 3 random seeds. SNSM rank = 128 and gap = 200. Learning rates were tuned over a grid for each batch size.

| Batch size | 1024 | 512 | 256 | 128 | 64 | 32 | 16 | 8 | 4 |
|---|---|---|---|---|---|---|---|---|---|
| Adam | 31.80 (1.87) | 30.46 (0.29) | 32.11 (1.32) | 34.57 (0.16) | 36.34 (0.16) | 38.91 (0.12) | 43.12 (0.26) | 48.88 (0.17) | 57.28 (0.80) |
| AdamwSN | 30.11 (0.15) | 29.81 (0.12) | 30.32 (0.07) | 31.30 (0.02) | 32.72 (0.11) | 35.38 (0.11) | 40.46 (0.97) | 45.81 (0.11) | 51.01 (0.25) |
| AdamSNSM | 31.39 (0.17) | 29.93 (0.07) | 30.08 (0.19) | 30.57 (0.08) | 32.35 (0.14) | 34.51 (0.14) | 37.05 (0.20) | 39.39 (0.02) | 44.27 (0.10) |

| | Adam | AdamSN | Adagrad | AdaGradSN |
|---|---|---|---|---|
| Mean | 24.69 | 22.98 | 25.95 | 24.57 |
| Stdev | 0.07 | 0.07 | 0.16 | 0.37 |

*Table 18.* Mean and standard deviation across 3 runs for different optimizers on pretraining LLaMA 130M task.

## C. Coordinate-Noise Density

This section further examine the coordinate-noise density model by providing additional empirical results across the train progress. We also provide the full derivation for the convergence of AdaGrad algorithms under various noise density rate.

### C.1. Empirical Validation

**Coordinate-noise density experiments.** To validate the coordinate-noise density model, we sample stochastic gradients repeatedly (via different mini batches) to obtain a sample variance estimate for the true sub-gaussian parameter $\sigma_i$ for each coordinate: if $g_1, \ldots, g_n \in \mathbb{R}^d$ are independent stochastic gradient samples, we can calculate the sample variance $S^2$ as an estimator for $\sigma^2$ as $S^2 = \frac{1}{n-1} \sum_{i=1}^n (g_i - \bar{g})^2$, where $\bar{g} = \frac{1}{n} \sum_{i=1}^n g_i$ is the sample mean. We pick $n = 200$ samples (with batch size equals 128) for estimating coordinate-noise on LLaMA 60M across various steps during the training process. Figure 11 shows the aggregated noise distribution across *all* parameters for LLaMA 60M after 100 training steps. There, the noise is quite low for the vast majority of coordinates except for some outliers. While the noise seems sparse in aggregate, a more fine-grained analysis, presented in Figure 3, shows that noises are dense per parameter, except for the $Q$ and $K$ attention projections in the deeper layers. Figures 12 to 17 in Appendix C.1 present more noise density rates across various parameters throughout different points of the training progress.

Figure 12 to 16 show the normalized noise density ratio for different parameters of LLaMA 60M as described in Section 3.2. The noise patterns show a clear layer-dependent structure, where early layers (like layer 0) maintain consistently high density

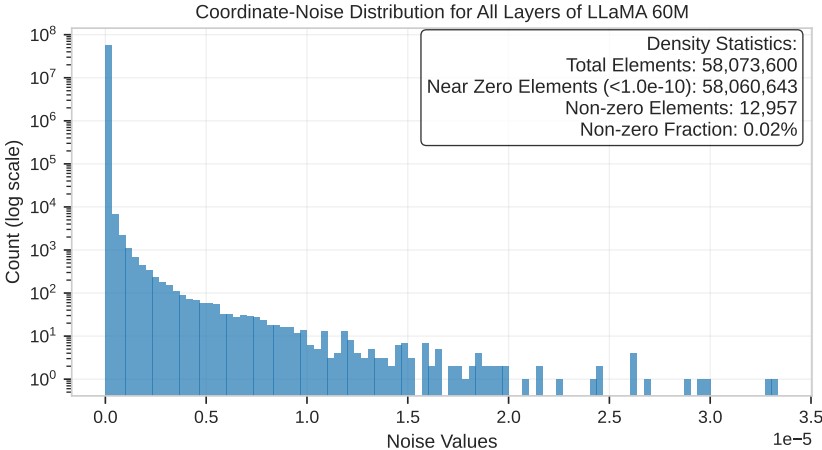

*Figure 11.* Aggregated noise distribution across *all* parameters after 100 steps of training.

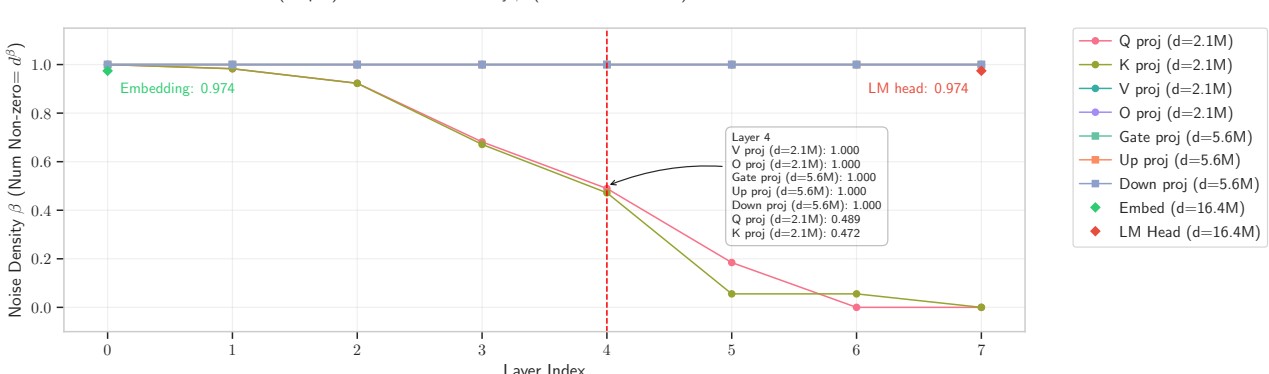

*Figure 12.* Noise density for different parameters of LLaMA 60M at Step 0.

(close to 1.0) throughout training, while deeper layers start very sparse and gradually become denser as training progresses. Notably, the embedding layer shows an opposite trend, starting relatively dense and becoming increasingly sparse by step 5000, suggesting different dynamics for embedding updates compared to attention layers. The middle layers show an interesting transition pattern, starting sparse but rapidly becoming dense after about 1000 steps, indicating a potential critical phase in training where these layers become more actively involved in learning.

### C.2. Convergence Rate Derivation

We derive the dimensional dependency of convergence rates for different AdaGrad variants below.

**AdaGrad-Coordinate.** For $c = d$ (AdaGrad-Coordinate), we get $\sum_{i=0}^{c-1} \|\sigma_{\Psi_i}\| = \alpha d^\beta$, $\|\sigma\|_2^2 = \alpha^2 d^\beta$, and $\sum_{i=0}^{c-1} \|\sigma_{\Psi_i}\|^4 = \alpha^4 d^\beta$, so the bound from Theorem 3.1 becomes

$$\frac{1}{T} \sum_{t=1}^{T} \|\nabla_t\|_2^2 \leq \tilde{O}\left(\alpha^4 d^\beta + \alpha^3 d^\beta + dL + d^{1.5}\alpha\right) \cdot \tilde{O}\left(\frac{\alpha d^\beta}{\sqrt{T}} + \frac{\alpha^2 d^\beta + \alpha d^\beta + Ld}{T}\right).$$

The dependency on $d$ for the slow term $O(1/\sqrt{T})$ is $d^{1.5}d^\beta = d^{1.5+\beta}$. The dependency on $d$ for the fast term $O(1/T)$ is $d^{1.5}d = d^{2.5}$. Note that there is an inherent $d^{1.5}$ dependency for the slow term that does not reduce as the coordinate-noise density decrease.

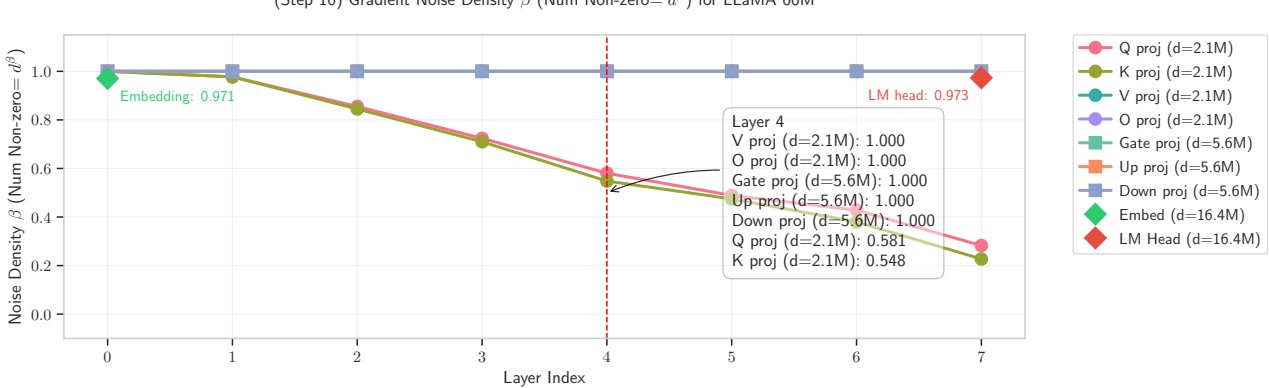

*Figure 13.* Noise density for different parameters of LLaMA 60M at Step 10.

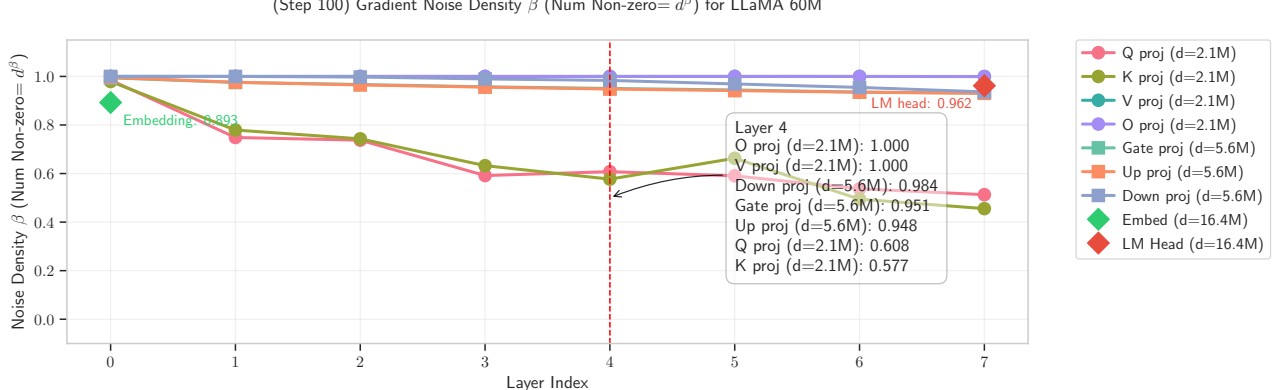

*Figure 14.* Noise density for different parameters of LLaMA 60M at Step 100.

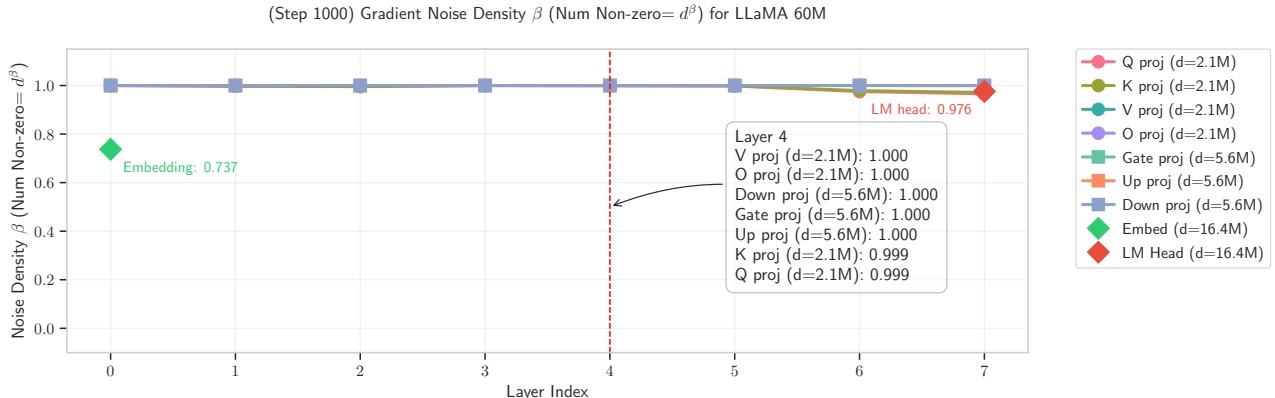

*Figure 15.* Noise density for different parameters of LLaMA 60M at Step 1000.

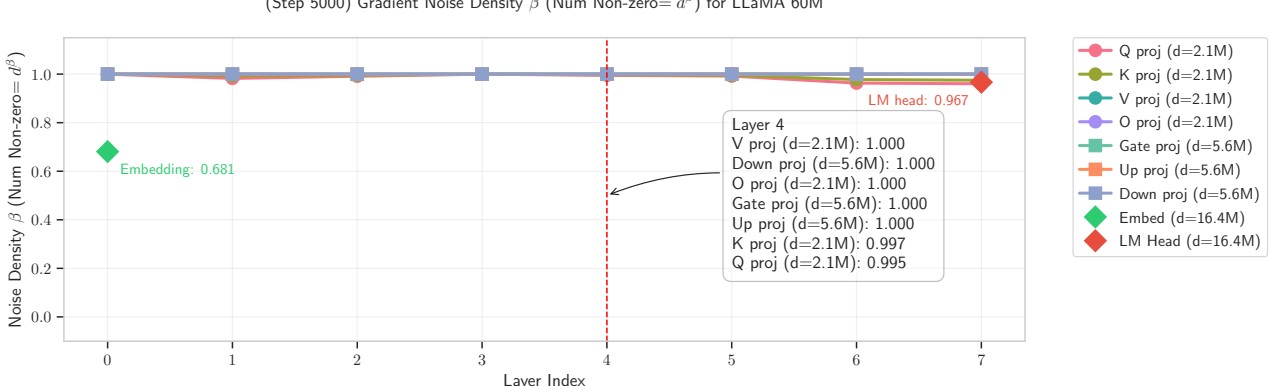

*Figure 16.* Noise density for different parameters of LLaMA 60M at Step 5000.

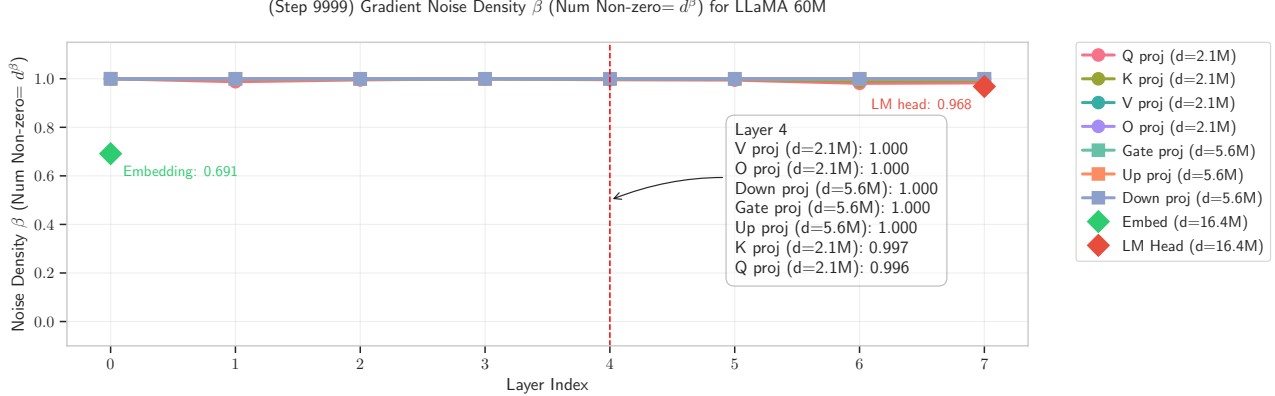

*Figure 17.* Noise density for different parameters of LLaMA 60M at Step 9999.

**AdaGrad-Norm** For $c = 1$ (AdaGrad-Norm), we get $\|\sigma\|_2^2 = \sum_{i=0}^{d} \|\sigma_i\|^2 = \alpha^2 d^\beta$, $\|\sigma\|_2 = \alpha d^{\beta/2}$, and $\|\sigma\|^4 = \alpha^4 d^{2\beta}$. This means that our bound from Theorem 3.1 becomes

$$\frac{1}{T} \sum_{t=1}^{T} \|\nabla_t\|_2^2 \leq \tilde{O}\left(\alpha^4 d^{2\beta} + \alpha^3 d^\beta + L + \alpha\right) \cdot \tilde{O}\left(\frac{\alpha d^{\beta/2}}{\sqrt{T}} + \frac{\alpha^2 d^\beta + \alpha d^{\beta/2} + L}{T}\right).$$

The dependency on $d$ for the slow term $O(1/\sqrt{T})$ is $d^{2\beta} \cdot d^{\beta/2} = d^{2.5\beta}$. The dependency on $d$ for the fast term $O(1/T)$ is $d^{2\beta} \cdot d^\beta = d^{3\beta}$. Note that when $\beta = 0$, or when all the noise is on a single coordinate, we recover the dimension-free results of previous works.

**AdaGrad-Subset-Norm.** Now, consider the following partition strategy, where we divide the coordinates into $c = d^{1-\beta} k$ subsets of size $d^\beta/k$ each with the $d^\beta$ noisy coordinates into just $k$ subsets so that the rest of the $c - k$ subsets do not contain any noisy coordinate. This is a reasonable choice due to the empirical validation from Section C.1: The noisy parameters seem to cluster in groups corresponding to the architecture.

With this strategy, we have $\left\|\sigma_{\Psi_j}\right\|_2^2 = \alpha^2 d^\beta/k \implies \left\|\sigma_{\Psi_j}\right\|_2 = \alpha d^{\beta/2}/k^{0.5}$ if $j$ is a noisy subset. We can compute $\sum_{i=0}^{c-1} \|\sigma_{\Psi_i}\| = \alpha d^{\beta/2} k^{0.5}$, $\|\sigma\|_2^2 = \sum_{i=0}^{c-1} \|\sigma_{\Psi_i}\|_2^2 = \alpha^2 d^\beta$, and $\sum_{i=0}^{c-1} \|\sigma_{\Psi_i}\|^4 = \alpha^4 d^{2\beta}/k$. From Theorem 3.1, we get a bound of

$$\frac{1}{T} \sum_{t=1}^{T} \|\nabla_t\|_2^2 \leq \tilde{O}\left(\alpha^4 d^{2\beta}/k + \alpha^3 d^\beta + d^{1-\beta} k L + \left(d^{1-\beta} k\right)^{3/2} \alpha\right) \cdot$$

$$\tilde{O}\left(\frac{\alpha d^{\beta/2} k^{0.5}}{\sqrt{T}} + \frac{\alpha^2 d^\beta + \alpha d^{\beta/2} k^{0.5} + L d^{1-\beta} k}{T}\right).$$

Set $k = d^{7\beta/5 - 3/5}$ so that $\left(d^{1-\beta} k\right)^{3/2} = d^{2\beta}/k = d^{3\beta/5 + 3/5}$. Then we can simplify

$$\frac{1}{T} \sum_{t=1}^{T} \|\nabla_t\|_2^2 \leq \tilde{O}\left(\alpha^4 d^{3(\beta+1)/5} + \alpha^3 d^\beta + d^{2(\beta+1)/5} L + d^{3(\beta+1)/5} \alpha\right) \cdot$$

$$\tilde{O}\left(\frac{\alpha d^{(12\beta-3)/10}}{\sqrt{T}} + \frac{\alpha^2 d^\beta + \alpha d^{(12\beta-3)/10} + L d^{2(\beta+1)/5}}{T}\right).$$

The dependency on $d$ for the slow term $O(1/\sqrt{T})$ is $d^{3(\beta+1)/5} \cdot d^{(12\beta-3)/10} = d^{3(1+6\beta)/10} = d^{0.3+1.8\beta}$. The dependency on $d$ for the fast term $O(1/T)$ is a bit more complicated: For $\beta \in [0, \frac{2}{3}]$, we have the dependency on $d$ is $d^{3(\beta+1)/5} \cdot d^{2(\beta+1)/5} = d^{\beta+1}$. For $\beta \in [\frac{2}{3}, 1]$, we have the dependency on $d$ is $d^{3(\beta+1)/5} \cdot d^\beta = d^{3(\beta+1)/5+\beta} = d^{1.6\beta+0.6}$. Note that this is only a possible partition strategy where the subset sizes are of equal size (which is probably the most natural and easiest to implement). There, the optimal subset size is $k = d^{1.4\beta-0.6}$, for which if we plug in $\beta \in [0, 1]$ we get a range from 1 to $d^{0.8}$.

### C.3. From Theory to Practice

Our theory provides an optimal grouping strategy that depends on the noise density. However, in practice, we must trade off the cost to figure out a good grouping and the performance gain from it. The key from the theory improvement is to group the coordinates with similar noise magnitudes together. However, any expensive method to figure out these groups (e.g. the Hessian in Adam-mini) would have detrimental effects on the wall clock time and memory. Instead, our heuristic as in Section 3.2 is meant to be a simple method to capture most of these groups.

Intuitively, coordinates in the same row/column either act on the same input or are used to compute the same output. The noise and normalization on each input and output would affect coordinates in the same row/columns in a correlated way. To provide some evidence, we perform the experiments in Section C.1 again in Figure 18, but with the noise grouped by the corresponding dimension according to the heuristics. There we see most groups have very low noise (very close to 0, namely less than $10^{-12}$) while a small number of groups (top 1 percentile in the annotation) have much larger noises. Overall, our heuristics aim to capture the similar noise coming from the same inputs and outputs. Our experiments suggest that this is a major part of the gain. There might be other simple sources of correlation in the noise magnitudes, which we leave for future work.

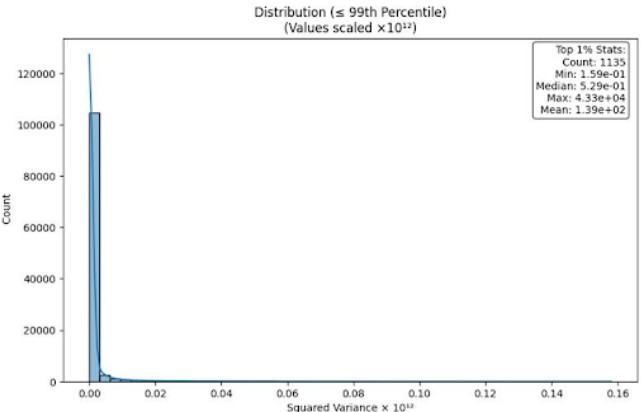

*Figure 18.* Coordinate noise grouped by the corresponding dimension according to the heuristics

# D. Subspace-Momentum Convergence Proof

In this section, we provide a high-probability convergence proof for SGD with Subspace-Momentum for non-convex smooth objective under sub-gaussian gradient noise.

## D.1. Setup and intuition

**Notations.** Given a linear operator $P : \mathbb{R}^d \to \mathbb{R}^k$, we have $P^* : \mathbb{R}^k \to \mathbb{R}^d$ is $P$'s adjoint[16], and we consider $P^*P : \mathbb{R}^d \to \mathbb{R}^d$ is a projection operator i.e. $P^*P$ is a bounded linear operator such that $(P^*P)^2 = P^*P$. Given a space $V \subseteq \mathbb{R}^d$, we denote its orthogonal subspace by $V^\perp := \left\{ v \in \mathbb{R}^d : \langle v, u \rangle = 0, \ \forall u \in V \right\}$.

Let $U = \text{row}(P) \subseteq \mathbb{R}^d$ be the row span of $P$. Let $\Psi : \mathbb{R}^d \to U$ be $\Psi(x) = P^*Px$ and $\Psi^\perp : \mathbb{R}^d \to U^\perp$ be $\Psi^\perp(x) = x - P^*Px$. Then for any vector $x$ in $\mathbb{R}^d$, have the orthogonal decomposition

$$x = \Psi(x) + \Psi^\perp(x).$$

**SGD with Subspace Momentum.** Let $g_t := \widehat{\nabla} f(x_t)$ denotes the stochastic gradient at time $t$. Let $\hat{c}_t = Pg_t$, $g_t^U = \Psi g_t = P^*Pg_t \in U$, and $g_t^\perp = g_t - g_t^U \in U^\perp$. Let $\nabla_t := \nabla f(x_t)$ be a short hand for the gradient at time $t$ and let $\nabla_t^U := \Psi(\nabla f(x_t)) \in U$ and $\nabla_t^\perp := \Psi^\perp(\nabla f(x_t)) \in U^\perp$ be the orthogonal decomposition of $\nabla f(x_t)$ with respect to $U$ and $U^\perp$, so that $\nabla_t = \nabla_t^U + \nabla_t^\perp$. Note that the superscript of a variable tries to suggest the space that it lives in (either $U$ or $U^\perp$). We have the following update rule for subspace momentum:

$$
\begin{aligned}
\hat{m}_t &= \beta \hat{m}_{t-1} + (1-\beta)Pg_t \\
g_t^\perp &= g_t - P^*Pg_t \\
m_t &= P^*\hat{m}_t \\
x_{t+1} &= x_t - \eta \left( m_t + g_t^\perp \right).
\end{aligned}
$$

Note that

$$
\begin{aligned}
m_t &= \beta P^*\hat{m}_{t-1} + (1-\beta)P^*Pg_t \\
&= \beta m_{t-1} + (1-\beta)g_t^U.
\end{aligned}
$$

---

[16]In $\mathbb{R}^d$, the adjoint $P^*$ of a linear operator $P$ is the linear operator given by the *transpose* of the matrix representation of $P$. We can also generalize Subspace-Momentum to general Hilbert spaces.

Expanding the terms, we see that this is just momentum in $U$

$$
\begin{aligned}
m_t &= \beta P^* \hat{m}_{t-1} + (1-\beta) P^* P g_t \\
&= \beta P^* \hat{m}_{t-1} + (1-\beta) g_t^U \\
&= \beta^2 P^* \hat{m}_{t-2} + (1-\beta)\beta g_{t-1}^U + (1-\beta) g_t^U \\
&= (1-\beta) \sum_{i=0}^{t} \beta^i g_{t-i}^U .
\end{aligned} \tag{5}
$$

Hence, we can think of the update of SGD-SM as performing two separate algorithms in the orthogonal subspaces: momentum in the subspace $U$ and SGD in the subspace $U^\perp$ (see also Figure 4) i.e. if we decompose $x_t$ into its orthogonal components $x_t = x_t^U + x_t^\perp$, then

$$
\begin{aligned}
x_{t+1}^U &= x_t^U - \eta m_t \\
&= x_t^U - \eta(1-\beta) \sum_{i=0}^{t} \beta^i g_{t-i}^U \\
x_{t+1}^\perp &= x_t^\perp - \eta g_t^\perp .
\end{aligned}
$$

For our analysis, let $\xi_t := g_t - \nabla_t$ denote the stochastic gradient error at time $t$. We can further decompose the error into its subspace components:

$$
\begin{aligned}
\xi_t &= \xi_t^U + \xi_t^\perp \\
&= \left( g_t^U - \nabla_t^U \right) + \left( g_t^\perp - \nabla_t^\perp \right) .
\end{aligned}
$$

**Basic facts.** We establish some facts for subspace momentum.

1. Pythagorean: $\|g_t\|^2 = \|g_t^U\|^2 + \|g_t^\perp\|^2$ and $\|\nabla_t\|^2 = \|\nabla_t^U\|^2 + \|\nabla_t^\perp\|^2$ and so on for these decompositions.

2. Subspace smoothness: If $f$ is smooth $\|\nabla f(x) - \nabla f(y)\| \le L \|x - y\|$, then due to contraction property of the projection operator, we have that the projected gradients of $f$ are also $L$-Lipschitz:

$$
\begin{aligned}
\|P^* P \nabla f(x) - P^* P \nabla f(y)\|^2 &= \|\nabla f(x) - \nabla f(y)\| \tag{6} \\
&\le L \|x - y\| .
\end{aligned}
$$

3. Subspace non-bias:

$$
\begin{aligned}
\mathbb{E}\left[ g_t^U - \nabla_t^U \right] &= \mathbb{E}\left[ \xi_t^U \right] \\
&= \mathbb{E}\left[ P^* P \left( g_t - \nabla_t \right) \right] \\
&= 0,
\end{aligned}
$$

and similarly for the orthogonal subspace

$$
\begin{aligned}
\mathbb{E}\left[ g_t^\perp - \nabla_t^\perp \right] &= \mathbb{E}\left[ \xi_t^\perp \right] \\
&= \mathbb{E}\left[ \xi_t - \xi_t^U \right] \\
&= 0.
\end{aligned}
$$

4. Subspace bounded variance: if the stochastic gradient's variance is bounded, then its subspace components are also bounded $\mathbb{E}\left[ \|\xi_t\|^2 \right]$:

$$
\begin{aligned}
\mathbb{E}\left[ \|g_t^U - \nabla_t^U\|^2 \right] &= \mathbb{E}\left[ \|\xi_t^U\|^2 \right] \\
&= \mathbb{E}\left[ \|\xi_t\|^2 - \|\xi_t^\perp\|^2 \right] \\
&\le \sigma^2 - \mathbb{E}\left[ \|\xi_t^\perp\|^2 \right],
\end{aligned}
$$

and similarly,

$$
\mathbb{E}\left[ \|\xi_t^\perp\|^2 \right] \le \sigma^2 - \mathbb{E}\left[ \|\xi_t^U\|^2 \right] .
$$

## D.2. Subspace-Momentum convergence proof

Suppose that $f : \mathbb{R}^d \to \mathbb{R}$ is $L$-smooth and stochastic gradients $\widehat{\nabla} f(x_t) = g_t$ is unbiased, i.e. $\mathbb{E}[g_t] = \nabla f(x_t)$, and has $\sigma$-sub-gaussian noise, i.e. $\mathbb{E}[\exp(\lambda^2 \|g_t - \nabla f(x_t)\|^2)] \leq \exp(\lambda^2 \sigma^2)$ for all $\lambda$ s.t. $|\lambda| \leq 1/\sigma$. First, we will show an error bound that is a starting point for the high-probability convergence results.

**Lemma D.1.** *If $f$ is L-smooth, then SGD with Subspace-Momentum (Algorithm 3) yields*

$$f(x_{T+1}) - f(x_1) \leq -\eta \sum_{t=1}^{T} \|\nabla_t\|^2 - \eta \sum_{t=1}^{T} \langle \nabla_t, \xi_t \rangle + \frac{(3-\beta)L\eta^2}{2(1-\beta)} \sum_{t=1}^{T} \|g_t\|^2 .$$

*Remark* D.2. Lemma D.1 shows that the optimization error of SGD-SM is quite similar to SGD-M.

*Proof.* Note that $m_t \in U$ and $r_t \in U^\perp$. Starting with smoothness, we have

$$
\begin{aligned}
f(x_{t+1}) &\leq f(x_t) + \langle \nabla f(x_t), x_{t+1} - x_t \rangle + \frac{L}{2} \|x_{t+1} - x_t\|^2 \\
&= f(x_t) - \eta \langle \nabla f(x_t), m_t + g_t^\perp \rangle + \frac{\eta^2 L}{2} \|m_t + g_t^\perp\|^2 \\
&= f(x_t) - \eta \langle \nabla f(x_t), m_t \rangle - \eta \langle \nabla f(x_t), g_t^\perp \rangle + \frac{\eta^2 L}{2} \|m_t + g_t^\perp\|^2 .
\end{aligned}
$$

We have

$$
\begin{aligned}
f(x_{t+1}) - f(x_t) &\leq -\eta \langle \nabla_t^U, m_t \rangle - \eta \langle \nabla_t^\perp, g_t^\perp \rangle + \frac{\eta^2 L}{2} \|m_t + g_t^\perp\|^2 \\
&= -\eta \langle \nabla_t^U, m_t \rangle - \eta \langle \nabla_t^\perp, g_t^\perp \rangle + \frac{\eta^2 L}{2} \|m_t\|^2 + \frac{\eta^2 L}{2} \|g_t^\perp\|^2 . \qquad \text{(Pythagorean)}
\end{aligned}
$$

Summing it up, we get

$$f(x_{T+1}) - f(x_1) \leq \underbrace{-\eta \sum_{t=1}^{T} \langle \nabla_t^U, m_t \rangle + \frac{\eta^2 L}{2} \sum_{t=1}^{T} \|m_t\|^2}_{\text{SGD with momentum error in } U} \underbrace{-\eta \sum_{t=1}^{T} \langle \nabla_t^\perp, g_t^\perp \rangle + \frac{\eta^2 L}{2} \sum_{t=1}^{T} \|g_t^\perp\|^2}_{\text{vanilla SGD error in } U^\perp} . \qquad (7)$$

We analyze $-\eta \langle \nabla_t^U, m_t \rangle + \frac{\eta^2 L}{2} \|m_t\|^2$ and $-\eta \langle \nabla_t^\perp, g_t^\perp \rangle + \frac{\eta^2 L}{2} \|g_t^\perp\|^2$ separately. Intuitively, the error within each subspace is controlled by their respective algorithm. Investigating the momentum term, we have

$$
\begin{aligned}
-\langle \nabla_t^U, m_t \rangle &= -\langle \nabla_t^U, \beta m_{t-1} + (1-\beta) g_t^U \rangle \\
&= -\beta \langle \nabla_t^U, m_{t-1} \rangle - (1-\beta) \langle \nabla_t^U, g_t^U \rangle \\
&= -\beta \langle \nabla_t^U - \nabla_{t-1}^U, m_{t-1} \rangle - \beta \langle \nabla_{t-1}^U, m_{t-1} \rangle - (1-\beta) \langle \nabla_t^U, g_t^U \rangle .
\end{aligned}
$$

We examine $-\langle \nabla_t^U - \nabla_{t-1}^U, m_{t-1} \rangle$:

$$
\begin{aligned}
-\langle \nabla_t^U - \nabla_{t-1}^U, m_{t-1} \rangle &= -\langle \nabla_t - \nabla_{t-1}, m_{t-1} \rangle + \langle \nabla_t^\perp - \nabla_{t-1}^\perp, m_{t-1} \rangle \\
&= -\langle \nabla_t - \nabla_{t-1}, m_{t-1} \rangle \\
&\leq \|\nabla_t - \nabla_{t-1}\| \|m_{t-1}\| \\
&\leq L \|x_t - x_{t-1}\| \|m_{t-1}\| \\
&= \eta L \|m_{t-1} + g_{t-1}^\perp\| \|m_{t-1}\| \\
&\leq \eta L \|m_{t-1} + g_{t-1}^\perp\|^2 .
\end{aligned}
$$

Now we have

$$-\left\langle \nabla_t^U, m_t \right\rangle \le \eta L\beta \left\| m_{t-1} + g_{t-1}^\perp \right\|^2 - \beta \left\langle \nabla_{t-1}^U, m_{t-1} \right\rangle - (1-\beta)\left\langle \nabla_t^U, g_t^U \right\rangle$$

$$\le \eta L \sum_{i=1}^{t-1} \beta^{t-i} \left\| m_i + g_i^\perp \right\|^2 - (1-\beta)\sum_{i=1}^{t} \beta^{t-i} \left\langle \nabla_i^U, g_i^U \right\rangle.$$

Summing over $t$, we have

$$-\eta \sum_{t=1}^{T} \left\langle \nabla_t^U, m_t \right\rangle \le \eta^2 L \sum_{t=1}^{T} \sum_{i=1}^{t-1} \beta^{t-i} \left\| m_i + g_i^\perp \right\|^2 - (1-\beta)\eta \sum_{t=1}^{T} \sum_{i=1}^{t} \beta^{t-i} \left\langle \nabla_i^U, g_i^U \right\rangle$$

$$= L\eta^2 \sum_{i=1}^{T} \sum_{t=i}^{T} \beta^{t-i} \left\| m_i + g_i^\perp \right\|^2 - (1-\beta)\eta \sum_{i=1}^{T} \sum_{t=i}^{T} \beta^{t-i} \left\langle \nabla_i^U, g_i^U \right\rangle \qquad \text{(swap the sum)}$$

$$\le L\eta^2 \sum_{i=1}^{T} \left\| m_i + g_i^\perp \right\|^2 \sum_{t=i}^{T} \beta^t - (1-\beta)\eta \sum_{i=1}^{T} \left\langle \nabla_i^U, g_i^U \right\rangle \sum_{t=i}^{T} \beta^t$$

$$\le \frac{L\eta^2}{1-\beta} \sum_{i=1}^{T} \left\| m_i + g_i^\perp \right\|^2 - \eta \sum_{i=1}^{T} \left\langle \nabla_i^U, g_i^U \right\rangle$$

$$= \frac{L\eta^2}{1-\beta} \sum_{i=1}^{T} \left( \|m_i\|^2 + \left\| g_i^\perp \right\|^2 \right) - \eta \sum_{i=1}^{T} \left\langle \nabla_i^U, \xi_i^U \right\rangle - \eta \sum_{i=1}^{T} \left\| \nabla_i^U \right\|^2.$$

Now, we look at $\sum_{i=1}^{T} \|m_i\|^2$:

$$\sum_{t=1}^{T} \|m_t\|^2 = \sum_{t=1}^{T} \left\| \beta m_{t-1} + (1-\beta)g_t^U \right\|^2$$

$$\le \sum_{t=1}^{T} \beta \|m_{t-1}\|^2 + (1-\beta)\left\| g_t^U \right\|^2 \qquad \text{(convexity of } \|\cdot\|^2\text{)}$$

$$\le \sum_{t=1}^{T} \beta \|m_t\|^2 + (1-\beta)\sum_{t=1}^{T} \left\| g_t^U \right\|^2$$

$$\implies \sum_{t=1}^{T} \|m_t\|^2 \le \sum_{t=1}^{T} \left\| g_t^U \right\|^2.$$

Examining the momentum error terms, we get

$$-\eta \sum_{t=1}^{T} \left\langle \nabla_t^U, m_t \right\rangle + \frac{\eta^2 L}{2} \sum_{t=1}^{T} \|m_t\|^2$$

$$\le \left( \frac{L\eta^2}{1-\beta} + \frac{\eta^2 L}{2} \right) \sum_{i=1}^{T} \|m_i\|^2 + \frac{L\eta^2}{1-\beta} \sum_{i=1}^{T} \left\| g_i^\perp \right\|^2 - \eta \sum_{i=1}^{T} \left\langle \nabla_i^U, \xi_i^U \right\rangle - \eta \sum_{i=1}^{T} \left\| \nabla_i^U \right\|^2$$

$$\le \left( \frac{(3-\beta)L\eta^2}{2(1-\beta)} \right) \sum_{i=1}^{T} \left\| g_i^U \right\|^2 + \frac{L\eta^2}{1-\beta} \sum_{i=1}^{T} \left\| g_i^\perp \right\|^2 - \eta \sum_{i=1}^{T} \left\langle \nabla_i^U, \xi_i^U \right\rangle - \eta \sum_{i=1}^{T} \left\| \nabla_i^U \right\|^2. \qquad (8)$$

We consider the SGD error terms in the orthogonal subspace:

$$-\eta \left\langle \nabla_t^\perp, g_t^\perp \right\rangle + \frac{\eta^2 L}{2} \left\| g_t^\perp \right\|^2 = -\eta \left\langle \nabla_t^\perp, \nabla_t^\perp - g_t^\perp \right\rangle - \eta \left\| \nabla_t^\perp \right\|^2 + \frac{\eta^2 L}{2} \left\| g_t^\perp \right\|^2$$

$$= -\eta \left\langle \nabla_t^\perp, \xi_t^\perp \right\rangle - \eta \left\| \nabla_t^\perp \right\|^2 + \frac{\eta^2 L}{2} \left\| g_t^\perp \right\|^2. \qquad (9)$$

Now we are ready to combine (8) and (9). First note the common terms $\left\| g_i^\perp \right\|^2$ in both equations combine to a sum similarly to $\left\| g_t^U \right\|^2$:

$$\underbrace{\frac{L\eta^2}{1-\beta} \sum_{t=1}^{T} \left\| g_t^\perp \right\|^2}_{\text{momentum}} + \underbrace{\frac{\eta^2 L}{2} \sum_{t=1}^{T} \left\| g_t^\perp \right\|^2}_{\text{SGD}} = \left( \frac{(3-\beta)L\eta^2}{2(1-\beta)} \right) \sum_{t=1}^{T} \left\| g_t^\perp \right\|^2.$$

Combining both terms, we see that the terms are combined from both subspaces (red from (8) and blue from (9)):

$$-\eta \sum_{t=1}^{T} \left\| \nabla_t^U \right\|^2 - \eta \sum_{t=1}^{T} \left\| \nabla_t^\perp \right\|^2 = -\eta \sum_{t=1}^{T} \left\| \nabla_t \right\|^2$$

$$-\eta \sum_{t=1}^{T} \left\langle \nabla_t^U, \xi_t^U \right\rangle - \eta \sum_{t=1}^{T} \left\langle \nabla_t^\perp, \xi_t^\perp \right\rangle = -\eta \sum_{t=1}^{T} \left\langle \nabla_t, \xi_t \right\rangle$$

$$\frac{(3-\beta)L\eta^2}{2(1-\beta)} \sum_{t=1}^{T} \left\| g_t^U \right\|^2 + \left( \frac{L\eta^2}{1-\beta} + \frac{\eta^2 L}{2} \right) \sum_{t=1}^{T} \left\| g_t^\perp \right\|^2 = \frac{(3-\beta)L\eta^2}{2(1-\beta)} \sum_{t=1}^{T} \left\| g_t \right\|^2.$$

Plugging everything back into (7), we have

$$f(x_{T+1}) - f(x_1) \leq -\eta \sum_{t=1}^{T} \left\| \nabla_t \right\|^2 - \eta \sum_{t=1}^{T} \left\langle \nabla_t, \xi_t \right\rangle + \frac{(3-\beta)L\eta^2}{2(1-\beta)} \sum_{t=1}^{T} \left\| g_t \right\|^2. \tag{10}$$

$\square$

### D.2.1. PROOF OF THEOREM 4.1.

Our proof uses the technical tools from (Liu et al., 2023c), although the strategy here has been simplified. We use Lemma A.1 from (Liu et al., 2023c):

**Lemma D.3** (Lemma A.1 from (Liu et al., 2023c)). *For any $a \geq 0$, $0 \leq b \leq \frac{1}{2\sigma}$ and a nonnegative $\sigma$-subgaussian random variable $X$, we have*

$$\mathbb{E}\left[ 1 + b^2 X^2 + \sum_{i=2}^{\infty} \frac{1}{i!} \left( aX + b^2 X^2 \right)^i \right] \leq \exp\left( 3\sigma^2 \left( a^2 + b^2 \right) \right).$$

We present some useful tools:

**Corollary D.4.** *Suppose that $X$ is a mean zero random vector in $\mathbb{R}^d$, where $\|X\|$ is $\sigma$-subgaussian. For $0 \leq a \leq \frac{1}{4\sigma^2}$ and $B \in \mathbb{R}^d$ then*

$$\mathbb{E}\left[ \exp\left( a \|X\|^2 + \langle B, X \rangle \right) \right] \leq \exp\left( 3\sigma^2 (a + \|B\|^2) \right).$$

*Proof.* We have

$$\mathbb{E}\left[ \exp\left( a \|X\|^2 + \langle B, X \rangle \right) \right] = \mathbb{E}\left[ 1 + a^2 \|X\|^2 + \langle B, X \rangle + \sum_{k=2}^{\infty} \frac{1}{k!} \left( a \|X\|^2 + \langle B, X \rangle \right)^k \right]$$

$$= \mathbb{E}\left[ 1 + a \|X\|^2 + \sum_{k=2}^{\infty} \frac{1}{k!} \left( a \|X\|^2 + \langle B, X \rangle \right)^k \right]$$

$$\leq \mathbb{E}\left[ 1 + a \|X\|^2 + \sum_{k=2}^{\infty} \frac{1}{k!} \left( a \|X\|^2 + \|B\| \|X\| \right)^k \right]$$

$$\leq \exp\left( 3\sigma^2 (a + \|B\|^2) \right).$$

$\square$

We can now control martingale via:

**Lemma D.5.** *If we have a sequence of random variable $X_t$ with $\mathcal{F}_t = \sigma(X_1, X_2, \ldots, X_{t-1})$ for $t = 1, 2, \ldots, T$. If we can bound $\mathbb{E}\left[\exp\left(X_t\right) \mid \mathcal{F}_t\right] \le \exp(Y_t)$, where $Y_t$ is $\mathcal{F}_t$-measurable, then*

$$\sum_{t=1}^{T} X_t \le \sum_{t=1}^{T} Y_t + \log\left(1/\delta\right)$$

*holds with probability at least $1 - \delta$.*

*Proof.* Define the $Z_t = X_t - Y_t$ and $S_t = \sum_{i=t}^{T} Z_i$. Then

$$\begin{aligned}
\mathbb{E}\left[\exp\left(Z_t\right) \mid \mathcal{F}_t\right] &= \mathbb{E}\left[\exp\left(X_t - Y_t\right) \mid \mathcal{F}_t\right] \\
&= \exp\left(-Y_t\right)\mathbb{E}\left[\exp\left(X_t\right) \mid \mathcal{F}_t\right] &&(Y_t \text{ is } \mathcal{F}_t\text{-measurable}) \\
&\le \exp(-Y_t)\exp(Y_t) = \exp(0) = 1.
\end{aligned}$$

Then we show $\mathbb{E}\left[\exp\left(S_1\right)\right] \le 1$ via an induction: we have $\mathbb{E}\left[\exp\left(S_T\right) \mid \mathcal{F}_T\right] = \mathbb{E}\left[\exp\left(Z_T\right) \mid \mathcal{F}_T\right] \le 1$. Suppose that $\mathbb{E}\left[\exp\left(S_{t+1}\right) \mid \mathcal{F}_{t+1}\right]$

$$\begin{aligned}
\mathbb{E}\left[\exp(S_t) \mid \mathcal{F}_t\right] &= \mathbb{E}\left[\exp(Z_t)\exp(S_{t+1}) \mid \mathcal{F}_t\right] \\
&= \mathbb{E}\left[\exp\left(Z_t\right)\mathbb{E}\left[\exp\left(S_{t+1}\right) \mid \mathcal{F}_{t+1}\right] \mid \mathcal{F}_t\right] \\
&\le \mathbb{E}\left[\exp(Z_t) \mid \mathcal{F}_t\right] \le 1.
\end{aligned}$$

Hence, this implies that $\mathbb{E}\left[\exp(S_1)\right] \le 1$. By Markov's inequality, this means that $S_1 \le \log(\frac{1}{\delta})$ with probability at least $1 - \delta$:

$$\begin{aligned}
S_1 &= \sum_{t=1}^{T} Z_t = \sum_{t=1}^{T} X_t - Y_t \le \log\left(1/\delta\right) \\
&\implies \sum_{t=1}^{T} X_t \le \sum_{t=1}^{T} Y_t + \log\left(1/\delta\right).
\end{aligned}$$

$\square$

We can now prove Theorem 4.1:

*Proof of Theorem 4.1.* Starting from Lemma D.1 and letting $\alpha = \frac{(3-\beta)L}{2(1-\beta)}$ and $\Delta_1 := f(x_1) - f_*$ for simplicity, we have

$$\begin{aligned}
\Delta_{T+1} &- \Delta_1 \\
&\le -\eta \sum_{i=1}^{T} \|\nabla_t\|^2 - \eta \sum_{i=1}^{T} \langle \nabla_t, \xi_t \rangle + \alpha\eta^2 \sum_{t=1}^{T} \|g_t\|^2 \\
&= -\eta \sum_{i=1}^{T} \|\nabla_t\|^2 - \eta \sum_{i=1}^{T} \langle \nabla_t, \xi_t \rangle + \alpha\eta^2 \sum_{t=1}^{T} \|\xi_t + \nabla_t\|^2 \\
&= \eta\left(\alpha\eta - 1\right) \sum_{i=1}^{T} \|\nabla_t\|^2 + \eta\left(\alpha\eta - 1\right) \sum_{i=1}^{T} \langle \nabla_t, \xi_t \rangle + \alpha\eta^2 \sum_{t=1}^{T} \|\xi_t\|^2.
\end{aligned}$$

Rearranging and defining some weight $w > 0$, we have

$$w\left(\Delta_{T+1} - \Delta_1\right) + \eta w\left(1 - \alpha\eta\right) \sum_{i=1}^{T} \|\nabla_t\|^2 \le \eta w\left(\alpha\eta - 1\right) \sum_{i=1}^{T} \langle \nabla_t, \xi_t \rangle + \alpha w\eta^2 \sum_{t=1}^{T} \|\xi_t\|^2.$$

Let $\mathcal{F}_t := \sigma\left(\xi_1, \ldots, \xi_{t-1}\right)$ denote the natural filtration. Now, since $\mathbb{E}\left[\sum_{t=1}^{T} \langle \nabla_t, \xi_t \rangle\right] = 0$ and $\xi_t$ is $\sigma$-sub-gaussian, we have that $\nabla_t \in \mathcal{F}_t$ and so if $0 \leq w\alpha\eta^2 \leq \frac{1}{4\sigma^2}$, Corollary D.4 implies

$$\mathbb{E}\left[\exp\left(w\eta\left(\alpha\eta - 1\right)\langle\nabla_t, \xi_t\rangle + w\alpha\eta^2 \|\xi_t\|^2\right) \mid \mathcal{F}_t\right] \leq \exp\left(3\sigma^2\left(w\alpha\eta^2 + w^2\eta^2\left(\alpha\eta - 1\right)^2 \|\nabla_t\|^2\right)\right),$$

Then Lemma D.5 implies that with probability at least $1 - \delta$, we have

$$w\eta\left(\alpha\eta - 1\right)\sum_{t=1}^{T}\langle\nabla_t, \xi_t\rangle + w\alpha\eta^2 \sum_{t=1}^{T}\|\xi_t\|^2 \leq 3\sigma^2 \sum_{t=1}^{T}\left(w\alpha\eta^2 + w^2\eta^2\left(\alpha\eta - 1\right)^2\|\nabla_t\|^2\right) + \log\left(1/\delta\right)$$

$$= 3\sigma^2 w\eta^2\alpha T + 3\sigma^2 w^2\eta^2\left(\alpha\eta - 1\right)^2 \sum_{t=1}^{T}\|\nabla_t\|^2 + \log\left(1/\delta\right).$$

Then with probability at least $1 - \delta$, we have

$$w\left(\Delta_{T+1} - \Delta_1\right) + \eta w\left(1 - \alpha\eta\right)\sum_{i=1}^{T}\|\nabla_t\|^2 \leq \eta w\left(\alpha\eta - 1\right)\sum_{i=1}^{T}\langle\nabla_t, \xi_t\rangle + \alpha w\eta^2 \sum_{t=1}^{T}\|\xi_t\|^2$$

$$\leq 3\sigma^2 w\eta^2\alpha T + 3\sigma^2 w^2\eta^2\left(\alpha\eta - 1\right)^2 \sum_{t=1}^{T}\|\nabla_t\|^2 + \log\left(1/\delta\right)$$

$$\implies \eta w\left(1 - \alpha\eta\right)\sum_{i=1}^{T}\|\nabla_t\|^2 \leq w\Delta_1 + 3\sigma^2 w\eta^2\alpha T + 3\sigma^2 w^2\eta^2\left(\alpha\eta - 1\right)^2 \sum_{t=1}^{T}\|\nabla_t\|^2 + \log\left(1/\delta\right).$$

Combining the $\|\nabla_t\|^2$ terms, we get

$$\left(\eta w\left(1 - \alpha\eta\right) - 3\sigma^2 w^2\eta^2\left(\alpha\eta - 1\right)^2\right)\sum_{i=1}^{T}\|\nabla_t\|^2 \leq w\Delta_1 + 3\sigma^2 w\eta^2\alpha T + \log\left(1/\delta\right). \tag{11}$$

Setting $w = \frac{1}{12\sigma^2\eta}$, then

$$\eta w\left(1 - \alpha\eta\right) - 3\sigma^2 w^2\eta^2\left(\alpha\eta - 1\right)^2 = \eta w\left(1 - \alpha\eta - 3\sigma^2 w\eta\left(\alpha\eta - 1\right)^2\right)$$

$$= \eta w\left(1 - \alpha\eta - \frac{1}{4}\left(\alpha\eta - 1\right)^2\right)$$

$$\geq \eta w\frac{1}{4}.$$

if $1 - \alpha\eta \geq 1/2$. Furthermore, we have that $w\alpha\eta^2 = \frac{\alpha\eta}{12\sigma^2} \leq \frac{1}{4\sigma^2}$ if $\eta \leq \frac{3}{\alpha}$, as required for Corr D.4. Hence, if $\eta \leq \frac{1}{2\alpha}$ then both requirements are satisfied. Consider the LHS of 11, we can bound

$$\left(\eta w\left(1 - \alpha\eta\right) - 3\sigma^2 w^2\eta^2\left(\alpha\eta - 1\right)^2\right)\sum_{i=1}^{T}\|\nabla_t\|^2 \geq \eta w\frac{1}{4}\sum_{i=1}^{T}\|\nabla_t\|^2$$

Finally, we have

$$\frac{\eta w}{4}\sum_{i=1}^{T}\|\nabla_t\|^2 \leq w\Delta_1 + 3\sigma^2 w\eta^2\alpha T + \log\left(1/\delta\right)$$

$$\sum_{i=1}^{T}\|\nabla_t\|^2 \leq \frac{4}{\eta}\Delta_1 + 3\sigma^2\eta\alpha T + 48\sigma^2 \log\left(1/\delta\right).$$

Setting $\eta = \min\left\{\frac{1}{2\alpha}; \sqrt{\frac{\Delta_1}{\sigma^2 \alpha T}}\right\}$, we have that with probability at least $1 - \delta$

$$\sum_{i=1}^{T} \|\nabla_t\|^2 \leq \frac{4}{\eta}\Delta_1 + 3\sigma^2 \eta \alpha T + 48\sigma^2 \log(1/\delta)$$

$$= \frac{4}{\min\left\{\frac{1}{2\alpha}; \sqrt{\frac{\Delta_1}{\sigma^2 \alpha T}}\right\}}\Delta_1 + 3\sigma^2 \min\left\{\frac{1}{2\alpha}; \sqrt{\frac{\Delta_1}{\sigma^2 \alpha T}}\right\}\alpha T + 48\sigma^2 \log(1/\delta)$$

$$\leq 4\left(2\alpha + \sqrt{\frac{\sigma^2 \alpha T}{\Delta_1}}\right)\Delta_1 + 3\sigma^2 \sqrt{\frac{\Delta_1}{\sigma^2 \alpha T}}\alpha T + 48\sigma^2 \log(1/\delta)$$

$$= 8\Delta_1 \alpha + 4\sigma\sqrt{\alpha T \Delta_1} + 3\sigma\sqrt{\Delta_1 \alpha T} + 48\sigma^2 \log(1/\delta)$$

$$= 8\Delta_1 \alpha + 7\sigma\sqrt{\alpha T \Delta_1} + 48\sigma^2 \log(1/\delta)$$

$$\implies \frac{1}{T}\sum_{i=1}^{T} \|\nabla_t\|^2 \leq \frac{8\Delta_1 \alpha}{T} + \frac{7\sigma\sqrt{\alpha\Delta_1}}{\sqrt{T}} + \frac{48\sigma^2 \log(1/\delta)}{T}.$$

We are done. $\qquad\square$

# E. Subset-Norm adaptive step size full theorem and proof

We show the full result in Theorem E.1 with all the polylog terms omitted from Theorem 3.1.

**Theorem E.1.** *Suppose that $f : \mathbb{R}^d \to \mathbb{R}$ is $L$-smooth and lower bounded by $f_*$. Given unbiased stochastic gradients $\widehat{\nabla} f(x_t)$ with stochastic gradient noise $\xi_t := \widehat{\nabla} f(x_t) - \nabla f(x_t)$ being $\sigma_i$-per-coordinate subgaussian for $i \in [d]$. For partitions of the parameters into disjoint subsets $[d] = \bigcup_{i=0}^{c-1} \Psi_i$ with $\Psi_i \cap \Psi_j = \emptyset$, if $i \neq j$, the iterates $x_t$ given by (3) satisfies the following inequality with probability at least $1 - 6c\delta$ (for failure probability $\delta > 0$):*

$$\frac{1}{T}\sum_{t=1}^{T} \|\nabla_t\|_2^2 \leq G(\delta) \cdot \left(\frac{4\sum_{i=0}^{c-1} \|\sigma_{\Psi_i}\|}{\sqrt{T}} + \frac{I(\delta)}{T}\right), \text{ where } G(\delta) \text{ and } I(\delta) \text{ are polylog terms:}$$

$$G(\delta) := \frac{\Delta_1}{\eta} + H(\delta) + \left(\ln T/\delta \|\sigma\|_2^2 + c\eta L + 4c^{3/2}\sigma_{\max}\sqrt{\log\frac{1}{\delta}}\right)\log\left(\frac{4\sqrt{T}\sum_{i=0}^{c-1} \|\sigma_{\Psi_i}\| + I(\delta)}{b_{0,\min}}\right)$$

$$I(\delta) := \|b_0\|_1 + \frac{2\Delta_1}{\eta} + \frac{8\log\frac{1}{\delta}}{b_{0,\min}}\|\sigma\|_2^2 + \sqrt{\log\frac{1}{\delta}}\sum_{i=0}^{c-1} \|\sigma_{\Psi_i}\| + 8\eta Lc\log\frac{4\eta L}{b_{0,\min}}$$

$$H(\delta) := \sum_{i=0}^{c-1}\left(\ln(T/\delta)\|\sigma_{\Psi_i}\|^2 + 2\alpha\right)\left(\frac{8\|\sigma_{\Psi_i}\|^2 \log\frac{1}{\delta}}{b_{0,i}^2} + 2\log\left(1 + \|\sigma_{\Psi_i}\|^2 T + \|\sigma_{\Psi_i}\|^2 \log\frac{1}{\delta}\right)\right).$$

*where $\|\sigma\|_2^2 = \sum_{i=1}^{d} \sigma_i^2$, $\|\sigma_{\Psi_i}\|^2 = \sum_{j \in \Psi_i} \sigma_j^2$, $\sigma_{\max} = \max_{i \in [d]} \sigma_i$, $\Delta_1 = f(x_1) - f_*$, $b_{0,\min} = \min_{i \in [d]} b_{0,i} > 0$.*

## E.1. Proof of Theorem E.1

For simplicity, in our analysis, we will use $\widehat{\nabla}_{t,i} := \widehat{\nabla}_i f(x_t)$ and $\nabla_{t,i} := \nabla_i f(x_t)$ to denote the $i$-th coordinate of the stochastic gradients and gradients at iterate $t$, respectively. The proof utilizes techniques and follows the strategies (Liu et al., 2023c), where the main effort is to adapt the techniques for handling subsets from the AdaGrad-Norm and AdaGrad-Coordinate proofs in (Liu et al., 2023c).

*Proof.* We write $\frac{\widehat{\nabla}_t}{b_t}$ to denote $\left(\frac{\widehat{\nabla}_t}{b_t}\right)_k = \frac{\widehat{\nabla}_k f(x_t)}{b_{t,i}}$ for $k \in \Psi_i$ (we will use this notation briefly to show some steps and will not be crucial in the main analysis). We start with the smoothness of $f$ and $\Delta_t := f(x_t) - f_*$.

$$\Delta_{t+1} - \Delta_t \le \langle \nabla f(x_t), x_{t+1} - x_t \rangle + \frac{L}{2} \|x_{t+1} - x_t\|^2$$

$$= -\eta \left\langle \nabla_t, \frac{\widehat{\nabla}_t}{b_t} \right\rangle + \frac{\eta^2 L}{2} \left\| \frac{\widehat{\nabla}_t}{b_t} \right\|^2 \tag{12}$$

$$= -\eta \sum_{i=0}^{c-1} \sum_{j \in \Psi_i} \frac{\nabla_{t,j} \widehat{\nabla}_{t,j}}{b_{t,i}} + \frac{\eta^2 L}{2} \sum_{i=0}^{c-1} \sum_{j \in \Psi_i} \frac{\widehat{\nabla}_{t,j}^2}{b_{t,i}^2}$$

$$= -\eta \sum_{i=0}^{c-1} \sum_{j \in \Psi_i} \frac{\nabla_{t,j} (\xi_{t,j} + \nabla_{t,j})}{b_{t,i}} + \frac{\eta^2 L}{2} \sum_{i=0}^{c-1} \sum_{j \in \Psi_i} \frac{\widehat{\nabla}_{t,j}^2}{b_{t,i}^2} \qquad (\xi_{t,i} = \widehat{\nabla}_{t,i} - \nabla_{t,i})$$

$$= -\eta \sum_{i=0}^{c-1} \sum_{j \in \Psi_i} \frac{\nabla_{t,j}^2}{b_{t,i}} - \eta \sum_{i=0}^{c-1} \sum_{j \in \Psi_i} \frac{\nabla_{t,j} \xi_{t,j}}{b_{t,i}} + \frac{\eta^2 L}{2} \sum_{i=0}^{c-1} \sum_{j \in \Psi_i} \frac{\widehat{\nabla}_{t,j}^2}{b_{t,i}^2}$$

$$= -\eta \sum_{i=0}^{c-1} \sum_{j \in \Psi_i} \frac{\nabla_{t,j}^2}{b_{t,i}} - \eta \sum_{i=0}^{c-1} \sum_{j \in \Psi_i} \frac{\nabla_{t,j} \xi_{t,j}}{a_{t,i}} + \eta \sum_{i=0}^{c-1} \sum_{j \in \Psi_i} \left( \frac{1}{a_{t,i}} - \frac{1}{b_{t,i}} \right) \nabla_{t,j} \xi_{t,j} + \frac{\eta^2 L}{2} \sum_{i=0}^{c-1} \sum_{j \in \Psi_i} \frac{\widehat{\nabla}_{t,j}^2}{b_{t,i}^2}. \tag{13}$$

Now, we analyze $\frac{1}{a_{t,i}} - \frac{1}{b_{t,i}}$ for $i = 0, 1, \ldots, c-1$:

$$\left| \frac{1}{a_{t,i}} - \frac{1}{b_{t,i}} \right| = \left| \frac{b_{t,i} - a_{t,i}}{a_{t,i} b_{t,i}} \right|$$

$$= \left| \frac{b_{t,i}^2 - a_{t,i}^2}{a_{t,i} b_{t,i} (b_{t,i} + a_{t,i})} \right|$$

$$= \left| \frac{b_{t-1,i}^2 + \left\| \widehat{\nabla}_{\Psi_i} f(x_t) \right\|^2 - b_{t-1,i}^2 - \|\nabla_{\Psi_i} f(x_t)\|^2}{a_{t,i} b_{t,i} (b_{t,i} + a_{t,i})} \right|$$

$$= \left| \frac{\left\| \widehat{\nabla}_{\Psi_i} f(x_t) \right\|^2 - \|\nabla_{\Psi_i} f(x_t)\|^2}{a_{t,i} b_{t,i} (b_{t,i} + a_{t,i})} \right|$$

$$= \left| \frac{\left( \left\| \widehat{\nabla}_{\Psi_i} f(x_t) \right\| - \|\nabla_{\Psi_i} f(x_t)\| \right) \left( \left\| \widehat{\nabla}_{\Psi_i} f(x_t) \right\| + \|\nabla_{\Psi_i} f(x_t)\| \right)}{a_{t,i} b_{t,i} (b_{t,i} + a_{t,i})} \right|.$$

Since $b_{t,i} = \sqrt{b_{t-1,i}^2 + \left\| \widehat{\nabla}_{\Psi_i} f(x_t) \right\|^2} \ge \left\| \widehat{\nabla}_{\Psi_i} f(x_t) \right\|$ and $a_{t,i} = \sqrt{b_{t-1,i}^2 + \|\nabla_{\Psi_i} f(x_t)\|^2} \ge \|\nabla_{\Psi_i} f(x_t)\|$, we have

$$\left| \frac{1}{a_{t,i}} - \frac{1}{b_{t,i}} \right| \le \left| \frac{\left( \left\| \widehat{\nabla}_{\Psi_i} f(x_t) \right\| - \|\nabla_{\Psi_i} f(x_t)\| \right) \left( \left\| \widehat{\nabla}_{\Psi_i} f(x_t) \right\| + \|\nabla_{\Psi_i} f(x_t)\| \right)}{a_{t,i} b_{t,i} \left( \left\| \widehat{\nabla}_{\Psi_i} f(x_t) \right\| + \|\nabla_{\Psi_i} f(x_t)\| \right)} \right|$$

$$\le \left| \frac{\left\| \widehat{\nabla}_{\Psi_i} f(x_t) \right\| - \|\nabla_{\Psi_i} f(x_t)\|}{a_{t,i} b_{t,i}} \right|$$

$$\le \frac{\left\| \widehat{\nabla}_{\Psi_i} f(x_t) - \nabla_{\Psi_i} f(x_t) \right\|}{a_{t,i} b_{t,i}}$$

$$= \frac{\|\xi_{t,\Psi_i}\|}{a_{t,i} b_{t,i}}.$$

Hence, we have

$$\left| \frac{1}{a_{t,i}} - \frac{1}{b_{t,i}} \right| \leq \frac{\|\xi_{t,\Psi_i}\|}{a_{t,i} b_{t,i}}.$$

Then from 13, taking the absolute value of $\sum_{i=0}^{c-1} \sum_{j\in\Psi_i} \left( \frac{1}{a_{t,i}} - \frac{1}{b_{t,i}} \right) \nabla_{t,j} \xi_{t,j}$, we can bound:

$$
\begin{aligned}
\Delta_{t+1} - \Delta_t &\leq -\eta \sum_{i=0}^{c-1} \sum_{j\in\Psi_i} \frac{\nabla_{t,j}^2}{b_{t,i}} - \eta \sum_{i=0}^{c-1} \sum_{j\in\Psi_i} \frac{\nabla_{t,j} \xi_{t,j}}{a_{t,i}} + \eta \sum_{i=0}^{c-1} \sum_{j\in\Psi_i} \left| \frac{1}{a_{t,i}} - \frac{1}{b_{t,i}} \right| |\nabla_{t,j} \xi_{t,j}| + \frac{\eta^2 L}{2} \sum_{i=0}^{c-1} \sum_{j\in\Psi_i} \frac{\widehat{\nabla}_{t,j}^2}{b_{t,i}^2} \\
&\leq -\eta \sum_{i=0}^{c-1} \sum_{j\in\Psi_i} \frac{\nabla_{t,j}^2}{b_{t,i}} - \eta \sum_{i=0}^{c-1} \sum_{j\in\Psi_i} \frac{\nabla_{t,j} \xi_{t,j}}{a_{t,i}} + \eta \sum_{i=0}^{c-1} \frac{\|\xi_{t,\Psi_i}\|}{a_{t,i} b_{t,i}} \sum_{j\in\Psi_i} |\nabla_{t,j} \xi_{t,j}| + \frac{\eta^2 L}{2} \sum_{i=0}^{c-1} \sum_{j\in\Psi_i} \frac{\widehat{\nabla}_{t,j}^2}{b_{t,i}^2} \\
&\overset{(1)}{\leq} -\eta \sum_{i=0}^{c-1} \sum_{j\in\Psi_i} \frac{\nabla_{t,j}^2}{b_{t,i}} - \eta \sum_{i=0}^{c-1} \sum_{j\in\Psi_i} \frac{\nabla_{t,j} \xi_{t,j}}{a_{t,i}} + \eta \sum_{i=0}^{c-1} \frac{\|\xi_{t,\Psi_i}\|}{a_{t,i} b_{t,i}} \|\nabla_{t,\Psi_i}\| \|\xi_{t,\Psi_i}\| + \frac{\eta^2 L}{2} \sum_{i=0}^{c-1} \sum_{j\in\Psi_i} \frac{\widehat{\nabla}_{t,j}^2}{b_{t,i}^2} \\
&\leq -\eta \sum_{i=0}^{c-1} \sum_{j\in\Psi_i} \frac{\nabla_{t,j}^2}{b_{t,i}} - \eta \sum_{i=0}^{c-1} \sum_{j\in\Psi_i} \frac{\nabla_{t,j} \xi_{t,j}}{a_{t,i}} \\
&\quad + \eta \sum_{i=0}^{c-1} \|\xi_{t,\Psi_i}\| \left( \frac{\|\xi_{t,\Psi_i}\|^2}{2 b_{t,i}^2} + \frac{\|\nabla_{t,\Psi_i}\|^2}{2 a_{t,i}^2} \right) + \frac{\eta^2 L}{2} \sum_{i=0}^{c-1} \sum_{j\in\Psi_i} \frac{\widehat{\nabla}_{t,j}^2}{b_{t,i}^2},
\end{aligned}
$$

where (1) is due to $\sum_{j\in\Psi_i} |\nabla_{t,j} \xi_{t,j}| = \langle |\nabla_{t,\Psi_i}|, |\xi_{t,\Psi_i}| \rangle \leq \|\nabla_{t,\Psi_i}\| \|\xi_{t,\Psi_i}\|$ and $|\cdot|$ denotes coordinate-wise absolute value when we apply to vectors. The last inequality is due to $2ab \leq a^2 + b^2$. Now, we can sum both sides for $t = 1, \ldots, T$ to telescope the LHS:

$$
\begin{aligned}
\Delta_{T+1} - \Delta_1 \leq \sum_{t=1}^{T} \Big( &-\eta \sum_{i=0}^{c-1} \sum_{j\in\Psi_i} \frac{\nabla_{t,j}^2}{b_{t,i}} - \eta \sum_{i=0}^{c-1} \sum_{j\in\Psi_i} \frac{\nabla_{t,j} \xi_{t,j}}{a_{t,i}} \\
&+ \eta \sum_{i=0}^{c-1} \|\xi_{t,\Psi_i}\| \left( \frac{\|\xi_{t,\Psi_i}\|^2}{2 b_{t,i}^2} + \frac{\|\nabla_{t,\Psi_i}\|^2}{2 a_{t,i}^2} \right) + \frac{\eta^2 L}{2} \sum_{i=0}^{c-1} \sum_{j\in\Psi_i} \frac{\widehat{\nabla}_{t,j}^2}{b_{t,i}^2} \Big).
\end{aligned}
$$

Rearranging gives

$$
\begin{aligned}
\sum_{t=1}^{T} \sum_{i=0}^{c-1} \sum_{j\in\Psi_i} \frac{\nabla_{t,j}^2}{b_{t,i}} \leq \frac{\Delta_1 - \Delta_{T+1}}{\eta} &- \underbrace{\sum_{t=1}^{T} \sum_{i=0}^{c-1} \sum_{j\in\Psi_i} \frac{\nabla_{t,j} \xi_{t,j}}{a_{t,i}}}_{A} \\
&+ \underbrace{\sum_{t=1}^{T} \sum_{i=0}^{c-1} \|\xi_{t,\Psi_i}\| \left( \frac{\|\xi_{t,\Psi_i}\|^2}{2 b_{t,i}^2} + \frac{\|\nabla_{t,\Psi_i}\|^2}{2 a_{t,i}^2} \right)}_{B} + \frac{\eta L}{2} \underbrace{\sum_{t=1}^{T} \sum_{i=0}^{c-1} \sum_{j\in\Psi_i} \frac{\widehat{\nabla}_{t,j}^2}{b_{t,i}^2}}_{C}.
\end{aligned}
$$

On the LHS, we note that

$$\sum_{t=1}^{T} \sum_{i=0}^{c-1} \sum_{j\in\Psi_i} \frac{\nabla_{t,j}^2}{b_{t,i}} = \sum_{t=1}^{T} \sum_{i=0}^{c-1} \frac{\|\nabla_{t,\Psi_i}\|^2}{b_{t,i}}.$$

We now bound each term separately. It's easiest to bound $C$: $\sum_{t=1}^{T} \sum_{i=0}^{c-1} \sum_{j \in \Psi_i} \frac{\widehat{\nabla}_{t,j}^2}{b_{t,i}^2}$:

$$
\sum_{t=1}^{T} \sum_{i=0}^{c-1} \sum_{j \in \Psi_i} \frac{\widehat{\nabla}_{t,j}^2}{b_{t,i}^2} = \sum_{i=0}^{c-1} \sum_{t=1}^{T} \sum_{j \in \Psi_i} \frac{\widehat{\nabla}_{t,j}^2}{b_{t,i}^2} = \sum_{i=1}^{d} \sum_{t=1}^{T} \frac{b_{t,i}^2 - b_{t-1,i}^2}{b_{t,i}^2} \leq \sum_{i=1}^{d} 2 \log \frac{b_{T,i}}{b_{0,i}}.
$$

$$
= \sum_{i=0}^{c-1} \sum_{t=1}^{T} \frac{\left\| \widehat{\nabla}_{t,\Psi_i} \right\|^2}{b_{t,i}^2}
$$

$$
= \sum_{i=0}^{c-1} \sum_{t=1}^{T} \frac{b_{t,i}^2 - b_{t-1,i}^2}{b_{t,i}^2}
$$

$$
= \sum_{i=0}^{c-1} \sum_{t=1}^{T} 1 - \frac{b_{t-1,i}^2}{b_{t,i}^2}
$$

$$
\leq \sum_{i=0}^{c-1} \sum_{t=1}^{T} \log \frac{b_{t,i}^2}{b_{t-1,i}^2}
$$

$$
= 2 \sum_{i=0}^{c-1} \log \prod_{t=1}^{T} \frac{b_{t,i}}{b_{t-1,i}}
$$

$$
= 2 \sum_{i=0}^{c-1} \log \frac{b_{T,i}}{b_{0,i}}.
$$

We now have a useful inequality

$$
\sum_{t=1}^{T} \frac{\left\| \widehat{\nabla}_{t,\Psi_i} \right\|^2}{b_{t,i}^2} \leq 2 \log \frac{b_{T,i}}{b_{0,i}}, \ \forall i = 0, \ldots, c-1. \tag{14}
$$

Next, we deal with $-\sum_{t=1}^{T} \sum_{i=0}^{c-1} \sum_{j \in \Psi_i} \frac{\nabla_{t,j} \xi_{t,j}}{a_{t,i}}$ via a martingale argument. Let $\mathcal{F}_t := \sigma\left(\xi_1, \ldots, \xi_{t-1}\right)$ denote the natural filtration. Note that $x_t$ is $\mathcal{F}_t$-measurable. For any $w > 0$, we have for each $i \in [c]$:

$$
\mathbb{E}\left[ \exp\left( -w \sum_{j \in \Psi_i} \frac{\nabla_{t,j} \xi_{t,j}}{a_{t,i}} - 2w^2 \sum_{j \in \Psi_i} \frac{\sigma_j^2 \nabla_{t,j}^2}{a_{t,i}^2} \right) \mid \mathcal{F}_t \right]
$$

$$
= \exp\left( -2w^2 \sum_{j \in \Psi_i} \frac{\sigma_j^2 \nabla_{t,j}^2}{a_{t,i}^2} \right) \mathbb{E}\left[ \exp\left( -w \sum_{j \in \Psi_i} \frac{\nabla_{t,j} \xi_{t,j}}{a_{t,i}} \right) \mid \mathcal{F}_t \right]
$$

$$
\leq 1.
$$

Then a simple inductive argument and using Markov's inequality gives with probability at least $1 - \delta$:

$$
-w \sum_{t=1}^{T} \sum_{j \in \Psi_i} \frac{\nabla_{t,j} \xi_{t,j}}{a_{t,i}} \leq 2w^2 \sum_{t=1}^{T} \sum_{j \in \Psi_i} \frac{\sigma_j^2 \nabla_{t,j}^2}{a_{t,i}^2} + \log \frac{1}{\delta}.
$$

By a union bound across all $c$ subsets, we have w.p. at least $1 - c\delta$:

$$
-\sum_{t=1}^{T} \sum_{i=0}^{c-1} \sum_{j \in \Psi_i} \frac{\nabla_{t,j} \xi_{t,j}}{a_{t,i}} \leq \sum_{t=1}^{T} \sum_{i=0}^{c-1} \sum_{j \in \Psi_i} \frac{w \sigma_j^2 \nabla_{t,j}^2}{a_{t,i}^2} + \frac{c}{w} \log \frac{1}{\delta}. \tag{15}
$$

Let's call the event that (15) happens $E_1$. Now, consider $\sum_{t=1}^{T} \sum_{i=0}^{c-1} \sum_{j \in \Psi_i} \frac{\nabla_{t,j}^2}{a_{t,i}^2}$. We have

$$\sum_{j \in \Psi_i} \frac{\nabla_{t,j}^2}{a_{t,i}^2} = \frac{\|\nabla_{t,\Psi_i}\|^2}{a_{t,i}^2} = \frac{\|\nabla_{t,\Psi_i}\|^2}{b_{t-1,i}^2 + \|\nabla_{t,\Psi_i}\|^2}$$

$$\overset{(*)}{\leq} \frac{2\left\|\widehat{\nabla}_{t,\Psi_i}\right\|^2 + 2\|\xi_{t,\Psi_i}\|^2}{b_{t-1,i}^2 + 2\left\|\widehat{\nabla}_{t,\Psi_i}\right\|^2 + 2\|\xi_{t,\Psi_i}\|^2}$$

$$\frac{\|\nabla_{t,\Psi_i}\|^2}{a_{t,i}^2} \leq 2\frac{\left\|\widehat{\nabla}_{t,\Psi_i}\right\|^2}{b_{t,i}^2} + 2\frac{\|\xi_{t,\Psi_i}\|^2}{b_{t,i}^2}.$$

For $(*)$ we use the fact that $\frac{x}{c+x}$ is an increasing function and $\|\nabla_{t,\Psi_i}\|^2 = \left\|\widehat{\nabla}_{t,\Psi_i} + \xi_{t,\Psi_i}\right\|^2 \leq 2\left\|\widehat{\nabla}_{t,\Psi_i}\right\|^2 + 2\|\xi_{t,\Psi_i}\|^2$.
Let $\sigma_{\max} := \max_{i \in [d]} \sigma_i$, then under event $E_1$, we have with probability at least $1 - c\delta$:

$$-\sum_{t=1}^{T} \sum_{i=0}^{c-1} \sum_{j \in \Psi_i} \frac{\nabla_{t,j} \xi_{t,j}}{a_{t,i}} \leq \sum_{t=1}^{T} \sum_{i=0}^{c-1} \sum_{j \in \Psi_i} \frac{w \sigma_j^2 \nabla_{t,j}^2}{a_{t,i}^2} + \frac{c}{w} \log \frac{1}{\delta}$$

$$\leq w \sigma_{\max}^2 \sum_{t=1}^{T} \sum_{i=0}^{c-1} \sum_{j \in \Psi_i} \frac{\nabla_{t,j}^2}{a_{t,i}^2} + \frac{c}{w} \log \frac{1}{\delta}$$

$$\leq w \sigma_{\max}^2 \sum_{t=1}^{T} \sum_{i=0}^{c-1} \left( 2\frac{\left\|\widehat{\nabla}_{t,\Psi_i}\right\|^2}{b_{t,i}^2} + 2\frac{\|\xi_{t,\Psi_i}\|^2}{b_{t,i}^2} \right) + \frac{c}{w} \log \frac{1}{\delta}$$

$$= \underbrace{\sigma_{\max} \sqrt{c \log \frac{1}{\delta}}}_{=:\alpha} \sum_{t=1}^{T} \sum_{i=0}^{c-1} \left( 2\frac{\left\|\widehat{\nabla}_{t,\Psi_i}\right\|^2}{b_{t,i}^2} + 2\frac{\|\xi_{t,\Psi_i}\|^2}{b_{t,i}^2} \right) + \sigma_{\max} \sqrt{c \log \frac{1}{\delta}} \quad (\text{set } w := \frac{\sqrt{c \log \frac{1}{\delta}}}{\sigma_{\max}})$$

$$= 2\alpha \sum_{t=1}^{T} \sum_{i=0}^{c-1} \left( \frac{\left\|\widehat{\nabla}_{t,\Psi_i}\right\|^2}{b_{t,i}^2} + \frac{\|\xi_{t,\Psi_i}\|^2}{b_{t,i}^2} \right) + \alpha.$$

where the second to last equality is due to choosing $w = \frac{\sqrt{c \log \frac{1}{\delta}}}{\sigma_{\max}}$ and the last equality is letting $\alpha := \sigma_{\max} \sqrt{c \log \frac{1}{\delta}}$ for readability.

Let $M_{T,i} = \max_{t \leq T} |\xi_{t,i}|$. Using our notation, we can define $M_{T,\Psi_i} := \max_{t \leq T} \|\xi_{t,\Psi_i}\|$. Under event $E_1$ (and our new

bound for $C$), we have that with probability at least $1 - c\delta$:

$$
\sum_{t=1}^{T}\sum_{i=0}^{c-1}\frac{\|\nabla_{t,\Psi_i}\|^2}{b_{t,i}} \overset{(C)}{\leq} \frac{\Delta_1}{\eta} - \sum_{t=1}^{T}\sum_{i=0}^{c-1}\sum_{j\in\Psi_i}\frac{\nabla_{t,j}\xi_{t,j}}{a_{t,i}} + \sum_{t=1}^{T}\sum_{i=0}^{c-1}\|\xi_{t,\Psi_i}\|\left(\frac{\|\xi_{t,\Psi_i}\|^2}{2b_{t,i}^2} + \frac{\|\nabla_{t,\Psi_i}\|^2}{2a_{t,i}^2}\right) + \eta L\sum_{i=0}^{c-1}\log\frac{b_{T,i}}{b_{0,i}}
$$

$$
\leq \frac{\Delta_1}{\eta} - \sum_{t=1}^{T}\sum_{i=0}^{c-1}\sum_{j\in\Psi_i}\frac{\nabla_{t,j}\xi_{t,j}}{a_{t,i}} \tag{16}
$$

$$
+ \sum_{t=1}^{T}\sum_{i=0}^{c-1}M_{T,\Psi_i}\left(\frac{\|\xi_{t,\Psi_i}\|^2}{2b_{t,i}^2} + \frac{\|\nabla_{t,\Psi_i}\|^2}{2a_{t,i}^2}\right) + \eta L\sum_{i=0}^{c-1}\log\frac{b_{T,i}}{b_{0,i}} \qquad (\text{def of } M_{T,\Psi_i})
$$

$$
\overset{(E_1)}{\leq} \frac{\Delta_1}{\eta} + 2\alpha\sum_{t=1}^{T}\sum_{i=0}^{c-1}\left(\underbrace{\frac{\left\|\widehat{\nabla}_{t,\Psi_i}\right\|^2}{b_{t,i}^2}}_{\text{bound with (C)}} + \frac{\|\xi_{t,\Psi_i}\|^2}{b_{t,i}^2}\right) + \alpha +
$$

$$
\sum_{t=1}^{T}\sum_{i=0}^{c-1}M_{T,\Psi_i}\left(\frac{\|\xi_{t,\Psi_i}\|^2}{2b_{t,i}^2} + \frac{\|\nabla_{t,\Psi_i}\|^2}{2a_{t,i}^2}\right) + \eta L\sum_{i=0}^{c-1}\log\frac{b_{T,i}}{b_{0,i}} \tag{17}
$$

$$
\overset{(C)}{\leq} \frac{\Delta_1}{\eta} + 2\alpha\sum_{t=1}^{T}\sum_{i=0}^{c-1}\frac{\|\xi_{t,\Psi_i}\|^2}{b_{t,i}^2} + \alpha +
$$

$$
\sum_{t=1}^{T}\sum_{i=0}^{c-1}M_{T,\Psi_i}\left(\frac{\|\xi_{t,\Psi_i}\|^2}{2b_{t,i}^2} + \frac{\|\nabla_{t,\Psi_i}\|^2}{2a_{t,i}^2}\right) + (\eta L + 4\alpha)\sum_{i=0}^{c-1}\log\frac{b_{T,i}}{b_{0,i}} \tag{18}
$$

$$
\leq \frac{\Delta_1}{\eta} + 2\alpha\sum_{t=1}^{T}\sum_{i=0}^{c-1}\frac{\|\xi_{t,\Psi_i}\|^2}{b_{t,i}^2} + \alpha + \tag{19}
$$

$$
\sum_{t=1}^{T}\sum_{i=0}^{c-1}M_{T,\Psi_i}\frac{\|\xi_{t,\Psi_i}\|^2}{2b_{t,i}^2} + \sum_{t=1}^{T}\sum_{i=0}^{c-1}M_{T,\Psi_i}\frac{\|\nabla_{t,\Psi_i}\|^2}{2a_{t,i}^2} + (\eta L + 4\alpha)\sum_{i=0}^{c-1}\log\frac{b_{T,i}}{b_{0,i}}. \tag{20}
$$

Let us turn our attention to $M_{T,\Psi_i} := \max_{t\leq T}\|\xi_{t,\Psi_i}\|$. Note that

$$
\Pr\left[\max_{t\in[T]}\|\xi_{t,\Psi_i}\|^2 \geq A\right] = \Pr\left[\exp\left(\frac{\max_{t\in[T]}\|\xi_{t,\Psi_i}\|^2}{w}\right) \geq \exp\left(\frac{A}{w}\right)\right] \qquad (\text{for } w > 0)
$$

$$
\leq \exp\left(-\frac{A}{w}\right)\mathbb{E}\left[\exp\left(\frac{\max_{t\in[T]}\|\xi_{t,\Psi_i}\|^2}{w}\right)\right] \qquad (\text{Markov})
$$

$$
= \exp\left(-\frac{A}{w}\right)\mathbb{E}\left[\max_{t\in[T]}\exp\left(\frac{\|\xi_{t,\Psi_i}\|^2}{w}\right)\right]
$$

$$
\leq \exp\left(-\frac{A}{w}\right)\sum_{t\in[T]}\mathbb{E}\left[\exp\left(\frac{\|\xi_{t,\Psi_i}\|^2}{w}\right)\right].
$$

We have

$$\mathbb{E}\left[\exp\left(\frac{\|\xi_{t,\Psi_i}\|^2}{w}\right)\right] = \mathbb{E}\left[\exp\left(\frac{\sum_{j\in\Psi_i}\xi_{t,j}^2}{w}\right)\right]$$

$$= \mathbb{E}\left[\exp\left(\frac{\sum_{j\in\Psi_i}\xi_{t,j}^2}{w}\right)\right]$$

$$= \mathbb{E}\left[\prod_{j\in\Psi_i}\exp\left(\frac{\xi_{t,j}^2}{w}\right)\right]$$

$$= \prod_{j\in\Psi_i}\mathbb{E}\left[\exp\left(\frac{\xi_{t,j}^2}{w}\right)\right]. \qquad\text{(independence)}$$

Since sub-gaussianity give us

$$\mathbb{E}\left[\exp\left(\lambda^2\xi_{t,i}^2\right)\right] \le \exp\left(\lambda^2\sigma_i^2\right), \forall\,|\lambda| \le \frac{1}{\sigma_i}, \forall i\in[d]\,,$$

we have $\mathbb{E}\left[\exp\left(\frac{\xi_{t,j}^2}{w}\right)\right] \le \exp\left(\frac{\sigma_j^2}{w}\right)$ if $\sqrt{\frac{1}{w}} \le \frac{1}{\sigma_j}$. We pick $w := \|\sigma_{\Psi_i}\|^2 = \sum_{j\in\Psi_i}\sigma_j^2 \ge \sigma_j^2,\ \forall j\in\Psi_i$ . Hence, we have

$$\mathbb{E}\left[\exp\left(\frac{\|\xi_{t,\Psi_i}\|^2}{\|\sigma_{\Psi_i}\|^2}\right)\right] \le \prod_{j\in\Psi_i}\exp\left(\frac{\sigma_j^2}{\|\sigma_{\Psi_i}\|^2}\right)$$

$$= \exp\left(\frac{\|\sigma_{\Psi_i}\|^2}{\|\sigma_{\Psi_i}\|^2}\right) = 1. \qquad (21)$$

We have actually shown that $\xi_{t,\Psi_i}$ is a $\|\sigma_{\Psi_i}\|^2$-subgaussian random variable in $\mathbb{R}^k$ (see Proposition 2.5.2 in (Vershynin, 2018)). This fact will come in handy later. Now, we have

$$\Pr\left[\max_{t\in[T]}\|\xi_{t,\Psi_i}\|^2 \ge A\right] \le \exp\left(-\frac{A}{\|\sigma_{\Psi_i}\|^2}\right)\sum_{t\in[T]}\mathbb{E}\left[\exp\left(\frac{\|\xi_{t,\Psi_i}\|^2}{\|\sigma_{\Psi_i}\|^2}\right)\right]$$

$$= \exp\left(-\frac{A}{\|\sigma_{\Psi_i}\|^2}\right)T.$$

Setting $\exp\left(-\frac{A}{\|\sigma_{\Psi_i}\|^2}\right)T = \delta$ gives $A = \|\sigma_{\Psi_i}\|^2\ln T/\delta$. Hence, we have with probability at least $1-\delta$,

$$M_{T,\Psi_i} = \max_{t\in[T]}\|\xi_{t,\Psi_i}\|^2 \le \|\sigma_{\Psi_i}\|^2\ln T/\delta. \qquad (22)$$

Union bounding across all $i = 0, 1, \ldots, c-1$, we have that with probability at least $1 - c\delta$,

$$M_{T,\Psi_i} \le \|\sigma_{\Psi_i}\|^2\ln T/\delta,\ \forall i = 0, 1, \ldots, c-1. \qquad (23)$$

Let us denote the event in (23) by $E_2$. Combining it with event $E_1$ and starting from 19, we have that with probability

$1 - c\delta$:

$$\sum_{t=1}^{T}\sum_{i=0}^{c-1}\frac{\|\nabla_{t,\Psi_i}\|^2}{b_{t,i}} \leq \frac{\Delta_1}{\eta} + 2\alpha\sum_{t=1}^{T}\sum_{i=0}^{c-1}\frac{\|\xi_{t,\Psi_i}\|^2}{b_{t,i}^2} + \alpha + \sum_{t=1}^{T}\sum_{i=0}^{c-1}M_{T,\Psi_i}\frac{\|\xi_{t,\Psi_i}\|^2}{2b_{t,i}^2} +$$

$$\sum_{t=1}^{T}\sum_{i=0}^{c-1}M_{T,\Psi_i}\frac{\|\nabla_{t,\Psi_i}\|^2}{2a_{t,i}^2} + (\eta L + 4\alpha)\sum_{i=0}^{c-1}\log\frac{b_{T,i}}{b_{0,i}}$$

$$\leq \frac{\Delta_1}{\eta} + 2\alpha\sum_{t=1}^{T}\sum_{i=0}^{c-1}\frac{\|\xi_{t,\Psi_i}\|^2}{b_{t,i}^2} + \ln T/\delta\sum_{t=1}^{T}\sum_{i=0}^{c-1}\|\sigma_{\Psi_i}\|^2\frac{\|\xi_{t,\Psi_i}\|^2}{2b_{t,i}^2} + \alpha +$$

$$\ln T/\delta\sum_{t=1}^{T}\sum_{i=0}^{c-1}\|\sigma_{\Psi_i}\|^2\frac{\|\nabla_{t,\Psi_i}\|^2}{2a_{t,i}^2} + (\eta L + 4\alpha)\sum_{i=0}^{c-1}\log\frac{b_{T,i}}{b_{0,i}}$$

$$= \frac{\Delta_1}{\eta} + \sum_{i=0}^{c-1}\left(\ln T/\delta\frac{\|\sigma_{\Psi_i}\|^2}{2} + 2\alpha\right)\sum_{t=1}^{T}\frac{\|\xi_{t,\Psi_i}\|^2}{b_{t,i}^2} + \alpha +$$

$$\ln T/\delta\sum_{i=0}^{c-1}\frac{\|\sigma_{\Psi_i}\|^2}{2}\sum_{t=1}^{T}\frac{\|\nabla_{t,\Psi_i}\|^2}{a_{t,i}^2} + (\eta L + 4\alpha)\sum_{i=0}^{c-1}\log\frac{b_{T,i}}{b_{0,i}}.$$

Recall that $\frac{\|\nabla_{t,\Psi_i}\|^2}{a_{t,i}^2} \leq 2\frac{\|\widehat{\nabla}_{t,\Psi_i}\|^2}{b_{t,i}^2} + 2\frac{\|\xi_{t,\Psi_i}\|^2}{b_{t,i}^2}$, we then have

$$\ln T/\delta\sum_{i=0}^{c-1}\frac{\|\sigma_{\Psi_i}\|^2}{2}\sum_{t=1}^{T}\frac{\|\nabla_{t,\Psi_i}\|^2}{a_{t,i}^2} \leq \ln T/\delta\sum_{i=0}^{c-1}\frac{\|\sigma_{\Psi_i}\|^2}{2}\sum_{t=1}^{T}\left(2\frac{\|\widehat{\nabla}_{t,\Psi_i}\|^2}{b_{t,i}^2} + 2\frac{\|\xi_{t,\Psi_i}\|^2}{b_{t,i}^2}\right)$$

$$= \ln T/\delta\sum_{i=0}^{c-1}\|\sigma_{\Psi_i}\|^2\sum_{t=1}^{T}\frac{\|\widehat{\nabla}_{t,\Psi_i}\|^2}{b_{t,i}^2} + \ln T/\delta\sum_{i=0}^{c-1}\|\sigma_{\Psi_i}\|^2\sum_{t=1}^{T}\frac{\|\xi_{t,\Psi_i}\|^2}{b_{t,i}^2}$$

$$\leq \ln T/\delta\sum_{i=0}^{c-1}\|\sigma_{\Psi_i}\|^2\log\frac{b_{T,i}}{b_{0,i}} + \ln T/\delta\sum_{i=0}^{c-1}\|\sigma_{\Psi_i}\|^2\sum_{t=1}^{T}\frac{\|\xi_{t,\Psi_i}\|^2}{b_{t,i}^2}. \qquad \text{(from 14)}$$

Hence, we have with probability at least $1 - 2c\delta$:

$$\sum_{t=1}^{T}\sum_{i=0}^{c-1}\frac{\|\nabla_{t,\Psi_i}\|^2}{b_{t,i}} \leq \frac{\Delta_1}{\eta} + \sum_{i=0}^{c-1}\left(\ln T/\delta\,\|\sigma_{\Psi_i}\|^2 + 2\alpha\right)\sum_{t=1}^{T}\frac{\|\xi_{t,\Psi_i}\|^2}{b_{t,i}^2} \tag{24}$$

$$+ \alpha + \sum_{i=0}^{c-1}\ln T/\delta\,\|\sigma_{\Psi_i}\|^2\log\frac{b_{T,i}}{b_{0,i}} + \sum_{i=0}^{c-1}(\eta L + 4\alpha)\log\frac{b_{T,i}}{b_{0,i}}$$

$$= \frac{\Delta_1}{\eta} + \sum_{i=0}^{c-1}\left(\ln T/\delta\,\|\sigma_{\Psi_i}\|^2 + 2\alpha\right)\sum_{t=1}^{T}\frac{\|\xi_{t,\Psi_i}\|^2}{b_{t,i}^2} \tag{25}$$

$$+ \alpha + \sum_{i=0}^{c-1}\left(\ln T/\delta\,\|\sigma_{\Psi_i}\|^2 + \eta L + 4\alpha\right)\log\frac{b_{T,i}}{b_{0,i}}. \tag{26}$$

Now, we bound $\sum_{t=1}^{T}\frac{\|\xi_{t,\Psi_i}\|^2}{b_{t,i}^2}$ and $\log\frac{b_{T,i}}{b_{0,i}}$. We need to first lower bound $\sum_{s=1}^{t}\left\|\widehat{\nabla}_{t,\Psi_i}\right\|^2$. We proceed by noting that

$$\|\widehat{\nabla}_{t,\Psi_i}\|^2 = \|\nabla_{t,\Psi_i} + \xi_{t,\Psi_i}\|^2$$

$$= \|\nabla_{t,\Psi_i}\|^2 + 2\langle\xi_{t,\Psi_i}, \nabla_{t,\Psi_i}\rangle + \|\xi_{t,\Psi_i}\|^2$$

$$\Rightarrow \|\nabla_{t,\Psi_i}\| - \|\widehat{\nabla}_{t,\Psi_i}\|^2 + \|\xi_{t,\Psi_i}\|^2 = 2\langle\xi_{t,\Psi_i}, \nabla_{t,\Psi_i}\rangle.$$

Define for $t \in \{0, 1, \cdots, T\}$ and some constant $v_s$ to be specified later:

$$
\begin{aligned}
U_{t+1} &= \exp\left(\sum_{s=1}^{t} w_s \left(\|\nabla_{s,\Psi_i}\| - \|\widehat{\nabla}_{s,\Psi_i}\|^2 + \|\xi_{s,\Psi_i}\|^2\right) - v_s\|\nabla_{s,\Psi_i}\|^2\right) \\
&= U_t \cdot \exp\left(w_t\left(\|\nabla_{t,\Psi_i}\| - \|\widehat{\nabla}_{t,\Psi_i}\|^2 + \|\xi_{t,\Psi_i}\|^2\right) - v_t\|\nabla_{t,\Psi_i}\|^2\right) \\
&= U_t \cdot \exp\left(w_t\left(2\langle\xi_{t,\Psi_i}, \nabla_{t,\Psi_i}\rangle\right) - v_t\|\nabla_{t,\Psi_i}\|^2\right).
\end{aligned}
$$

First, note that $U_t \in \mathcal{F}_t$. We show that $U_t$ is a supermartingale

$$
\begin{aligned}
\mathbb{E}\left[U_{t+1} \mid \mathcal{F}_t\right] &= \mathbb{E}\left[U_t \cdot \exp\left(w_t\left(2\langle\xi_{t,\Psi_i}, \nabla_{t,\Psi_i}\rangle\right) - v_t\|\nabla_{t,\Psi_i}\|^2\right) \mid \mathcal{F}_t\right] \\
&= U_t \exp\left(-v_t\|\nabla_{t,\Psi_i}\|^2\right)\mathbb{E}\left[\exp\left(2w_t\langle\xi_{t,\Psi_i}, \nabla_{t,\Psi_i}\rangle\right) \mid \mathcal{F}_t\right] \\
&\overset{(*)}{\leq} U_t \exp\left(-v_t\|\nabla_{t,\Psi_i}\|^2\right)\mathbb{E}\left[\exp\left(4w_t^2\|\sigma_{\Psi_i}\|^2\|\nabla_{t,\Psi_i}\|^2\right) \mid \mathcal{F}_t\right] \\
&= U_t, && (v_t = 4w_t^2\|\sigma_{\Psi_i}\|^2)
\end{aligned}
$$

where $(*)$ is due to Lemma 2.2 of (Liu et al., 2023c) and the fact that $\xi_{t,\Psi_i}$ is $\|\sigma_{\Psi_i}\|^2$-subgaussian from (21). Hence, by Ville's supermartingale inequality, we have

$$
\Pr\left[\max_{t\in[T+1]} U_t \geq \delta^{-1}\right] \leq \delta\mathbb{E}\left[U_1\right] = \delta.
$$

This implies w.p. $\geq 1 - \delta$, $\forall 0 \leq t \leq T$:

$$
\begin{aligned}
&\sum_{s=1}^{t} w_s\left(\|\nabla_{s,\Psi_i}\| - \|\widehat{\nabla}_{s,\Psi_i}\|^2 + \|\xi_{s,\Psi_i}\|^2\right) - v_s\|\nabla_{s,\Psi_i}\|^2 \leq \log\frac{1}{\delta} \\
\implies &\sum_{s=1}^{t}\left(w_s - 4w_s^2\|\sigma_{\Psi_i}\|^2\right)\|\nabla_{s,\Psi_i}\|^2 + \sum_{s=1}^{t} w_s\|\xi_{s,\Psi_i}\|^2 \leq \sum_{s=1}^{t} w_s\|\widehat{\nabla}_{s,\Psi_i}\|^2 + \log\frac{1}{\delta} \\
\iff &\sum_{s=1}^{t}\left(1 - 4w_s\|\sigma_{\Psi_i}\|^2\right)\|\nabla_{s,\Psi_i}\|^2 + \sum_{s=1}^{t}\|\xi_{s,\Psi_i}\|^2 \leq \sum_{s=1}^{t}\|\widehat{\nabla}_{s,\Psi_i}\|^2 + \frac{1}{w_s}\log\frac{1}{\delta}.
\end{aligned}
$$

Set $w_s = \frac{1}{4\|\sigma_{\Psi_i}\|^2}$ to get

$$
\sum_{s=1}^{t}\|\xi_{s,\Psi_i}\|^2 \leq \sum_{s=1}^{t}\|\widehat{\nabla}_{s,\Psi_i}\|^2 + 4\|\sigma_{\Psi_i}\|^2\log\frac{1}{\delta}, \ \forall t \leq T. \tag{27}
$$

We are now ready to bound $\sum_{t=1}^{T} \frac{\|\xi_{t,\Psi_i}\|^2}{b_{t,i}^2}$. Starting by applying (27), we have that with probability at least $1 - \delta$

$$
\begin{aligned}
\sum_{t=1}^{T} \frac{\|\xi_{t,\Psi_i}\|^2}{b_{t,i}^2} &= \sum_{t=1}^{T} \frac{\|\xi_{t,\Psi_i}\|^2}{b_{0,i}^2 + \sum_{s=1}^{t}\left\|\widehat{\nabla}_{t,\Psi_i}\right\|^2} \\
&\leq \sum_{t=1}^{T} \frac{\|\xi_{t,\Psi_i}\|^2}{b_{0,i}^2 + \left(\sum_{s=1}^{t}\|\xi_{s,\Psi_i}\|^2 - 4\|\sigma_{\Psi_i}\|^2\log\frac{1}{\delta}\right)^+}
\end{aligned}
$$

where $(x)^+ = \max\{x, 0\}$. Let $\tau = \max\left(\{0\} \cup \left\{t \in \mathbb{N}_{\leq T} \mid \sum_{s=1}^{t} \|\xi_{s,\Psi_i}\|^2 \leq 2C\right\}\right)$ for some $C \geq 0$. We have

$$\sum_{t=1}^{T} \frac{\|\xi_{t,\Psi_i}\|^2}{b_{t,i}^2} = \sum_{t=1}^{\tau} \frac{\|\xi_{t,\Psi_i}\|^2}{b_{t,i}^2} + \sum_{t=\tau+1}^{T} \frac{\|\xi_{t,\Psi_i}\|^2}{b_{0,i}^2 + \sum_{s=1}^{t} \left\|\widehat{\nabla}_{t,\Psi_i}\right\|^2}$$

$$\leq \frac{1}{b_{0,i}^2} \sum_{t=1}^{\tau} \|\xi_{t,\Psi_i}\|^2 + \sum_{t=\tau+1}^{T} \frac{\|\xi_{t,\Psi_i}\|^2}{b_{0,i}^2 + \sum_{s=1}^{t} \|\xi_{s,\Psi_i}\|^2 - 4\|\sigma_{\Psi_i}\|^2 \log\frac{1}{\delta}}$$

$$\leq \frac{2C}{b_{0,i}^2} + \sum_{t=\tau+1}^{T} \frac{\|\xi_{t,\Psi_i}\|^2}{b_{0,i}^2 + \sum_{s=1}^{t} \|\xi_{s,\Psi_i}\|^2 - 4\|\sigma_{\Psi_i}\|^2 \log\frac{1}{\delta}}.$$

Now, since $\frac{\sum_{s=1}^{t} \|\xi_{s,\Psi_i}\|^2}{2} \geq C$ for $t > \tau$, we have $b_{0,i}^2 + \sum_{s=1}^{t} \|\xi_{s,\Psi_i}\|^2 - 4\|\sigma_{\Psi_i}\|^2 \log\frac{1}{\delta} \geq b_{0,i}^2 - 4\|\sigma_{\Psi_i}\|^2 \log\frac{1}{\delta} + C + \frac{1}{2}\sum_{s=1}^{t} \|\xi_{s,\Psi_i}\|^2$. If $b_{0,i}^2 - 4\|\sigma_{\Psi_i}\|^2 \log\frac{1}{\delta} \geq 0$, then we pick $C = 0$ and $b_{0,i}^2 - 4\|\sigma_{\Psi_i}\|^2 \log\frac{1}{\delta} + C + \frac{1}{2}\sum_{s=1}^{t} \|\xi_{s,\Psi_i}\|^2 \geq \frac{1}{2}\sum_{s=1}^{t} \|\xi_{s,\Psi_i}\|^2$. If $b_{0,i}^2 - 4\|\sigma_{\Psi_i}\|^2 \log\frac{1}{\delta} < 0$, we pick $C = 4\|\sigma_{\Psi_i}\|^2 \log\frac{1}{\delta} - b_{0,i}^2 > 0$, which gives $b_{0,i}^2 - 4\|\sigma_{\Psi_i}\|^2 \log\frac{1}{\delta} + C + \frac{1}{2}\sum_{s=1}^{t} \|\xi_{s,\Psi_i}\|^2 \geq \frac{1}{2}\sum_{s=1}^{t} \|\xi_{s,\Psi_i}\|^2$. In either case, we have $b_{0,i}^2 - 4\|\sigma_{\Psi_i}\|^2 \log\frac{1}{\delta} + C + \frac{1}{2}\sum_{s=1}^{t} \|\xi_{s,\Psi_i}\|^2 \geq \frac{1}{2}\sum_{s=1}^{t} \|\xi_{s,\Psi_i}\|^2$. Hence, letting $C = \max\left(0, 4\|\sigma_{\Psi_i}\|^2 \log\frac{1}{\delta} - b_{0,i}^2\right) \leq 4\|\sigma_{\Psi_i}\|^2 \log\frac{1}{\delta}$, we have w.p. at least $1 - \delta$:

$$\sum_{t=1}^{T} \frac{\|\xi_{t,\Psi_i}\|^2}{b_{t,i}^2} \leq \frac{2C}{b_{0,i}^2} + 2\sum_{t=\tau+1}^{T} \frac{\|\xi_{t,\Psi_i}\|^2}{\sum_{s=1}^{t} \|\xi_{s,\Psi_i}\|^2}$$

$$\leq \frac{2C}{b_{0,i}^2} + 2\sum_{t=1}^{T} \frac{\|\xi_{t,\Psi_i}\|^2}{\sum_{s=1}^{t} \|\xi_{s,\Psi_i}\|^2}$$

$$\leq \frac{8\|\sigma_{\Psi_i}\|^2 \log\frac{1}{\delta}}{b_{0,i}^2} + 2\sum_{t=1}^{T} \frac{\|\xi_{t,\Psi_i}\|^2}{\sum_{s=1}^{t} \|\xi_{s,\Psi_i}\|^2}.$$

Let $X_t = 1 + \sum_{s=1}^{t} \|\xi_{s,\Psi_i}\|^2 = X_{t-1} + \|\xi_{t,\Psi_i}\|^2$, where $X_0 = 1$. Then,

$$\sum_{t=1}^{T} \frac{\|\xi_{t,\Psi_i}\|^2}{\sum_{s=1}^{t} \|\xi_{s,\Psi_i}\|^2} = \sum_{t=1}^{T} \frac{X_t - X_{t-1}}{X_t} = \sum_{t=1}^{T} 1 - \frac{X_{t-1}}{X_t}$$

$$\leq \sum_{t=1}^{T} \log\left(\frac{X_t}{X_{t-1}}\right)$$

$$= \log\left(\prod_{t=1}^{T} \frac{X_t}{X_{t-1}}\right)$$

$$= \log\left(\frac{X_T}{X_0}\right) = \log\left(1 + \sum_{t=1}^{T} \|\xi_{s,\Psi_i}\|^2\right).$$

Hence, with probability at least $1 - \delta$:

$$\sum_{t=1}^{T} \frac{\|\xi_{t,\Psi_i}\|^2}{b_{t,i}^2} \leq \frac{8\|\sigma_{\Psi_i}\|^2 \log\frac{1}{\delta}}{b_{0,i}^2} + 2\log\left(1 + \sum_{t=1}^{T} \|\xi_{s,\Psi_i}\|^2\right). \tag{28}$$

It remains to bound $\sum_{t=1}^{T} \|\xi_{s,\Psi_i}\|^2$. Note that

$$
\begin{aligned}
\Pr\left[\sum_{t=1}^{T} \|\xi_{s,\Psi_i}\|^2 \geq u\right] &= \Pr\left[\exp\left(\sum_{t=1}^{T} \frac{\|\xi_{s,\Psi_i}\|^2}{\|\sigma_{\Psi_i}\|^2}\right) \geq \exp\left(\frac{u}{\|\sigma_{\Psi_i}\|^2}\right)\right] \\
&\leq \frac{\mathbb{E}\left[\exp\left(\sum_{t=1}^{T} \frac{\|\xi_{s,\Psi_i}\|^2}{\|\sigma_{\Psi_i}\|^2}\right)\right]}{\exp\left(\frac{u}{\|\sigma_{\Psi_i}\|^2}\right)} \\
&\leq \frac{\exp(T)}{\exp\left(\frac{u}{\|\sigma_{\Psi_i}\|^2}\right)} \qquad (\xi_{s,\Psi_i} \text{ is } \|\sigma_{\Psi_i}\|^2\text{-subgaussian})
\end{aligned}
$$

Choosing $u = \|\sigma_{\Psi_i}\|^2 T + \|\sigma_{\Psi_i}\|^2 \log\frac{1}{\delta}$ gives that with probability at least $1 - \delta$, we have

$$
\sum_{t=1}^{T} \|\xi_{s,\Psi_i}\|^2 \leq \|\sigma_{\Psi_i}\|^2 T + \|\sigma_{\Psi_i}\|^2 \log\frac{1}{\delta}. \tag{29}
$$

Having a high probability bound on the sum of the stochastic error of the subset-norm, we can combine both events from (28) and (29) to get that with probability at least $1 - 2\delta$:

$$
\sum_{t=1}^{T} \frac{\|\xi_{t,\Psi_i}\|^2}{b_{t,i}^2} \leq \frac{8\|\sigma_{\Psi_i}\|^2 \log\frac{1}{\delta}}{b_{0,i}^2} + 2\log\left(1 + \|\sigma_{\Psi_i}\|^2 T + \|\sigma_{\Psi_i}\|^2 \log\frac{1}{\delta}\right). \tag{30}
$$

Then we can also condition on the event that (30) happens and combine it with the event in (26) to get that with probability at least $1 - 2c\delta$ (assuming $c \geq 2$), we have

$$
\sum_{t=1}^{T}\sum_{i=0}^{c-1} \frac{\|\nabla_{t,\Psi_i}\|_2^2}{b_{t,i}} \leq \frac{\Delta_1}{\eta} + \sum_{i=0}^{c-1}\left(\ln T/\delta \|\sigma_{\Psi_i}\|^2 + 2\alpha\right)\sum_{t=1}^{T} \frac{\|\xi_{t,\Psi_i}\|^2}{b_{t,i}^2} \tag{31}
$$

$$
+ \alpha + \sum_{i=0}^{c-1}\left(\ln T/\delta \|\sigma_{\Psi_i}\|^2 + \eta L + 4\alpha\right)\log\frac{b_{T,i}}{b_{0,i}} \tag{32}
$$

$$
\leq \frac{\Delta_1}{\eta} + \underbrace{\sum_{i=0}^{c-1}\left(\ln T/\delta \|\sigma_{\Psi_i}\|^2 + 2\alpha\right)\left(\frac{8\|\sigma_{\Psi_i}\|^2 \log\frac{1}{\delta}}{b_{0,i}^2} + 2\log\left(1 + \|\sigma_{\Psi_i}\|^2 T + \|\sigma_{\Psi_i}\|^2 \log\frac{1}{\delta}\right)\right)}_{=:H(\delta)}
$$

$$
\tag{33}
$$

$$
+ \alpha + \sum_{i=0}^{c-1}\left(\ln T/\delta \|\sigma_{\Psi_i}\|^2 + \eta L + 4\alpha\right)\log\frac{b_{T,i}}{b_{0,i}}
$$

$$
= \frac{\Delta_1}{\eta} + H(\delta) + \alpha + \sum_{i=0}^{c-1}\left(\ln T/\delta \|\sigma_{\Psi_i}\|^2 + \eta L + 4\alpha\right)\log\frac{b_{T,i}}{b_{0,i}}. \tag{34}
$$

First, note that $b_{T,i} \leq \|b_T\|_1 = \sum_{i=0}^{c-1} b_{T,i}$. Letting $b_{0,\min} := \min_i b_{0,i}$, we then have

$$
\begin{aligned}
\sum_{i=0}^{c-1}\left(\ln T/\delta \|\sigma_{\Psi_i}\|^2 + \eta L + 4\alpha\right)\log\frac{b_{T,i}}{b_{0,i}} &\leq \log\frac{\|b_T\|_1}{b_{0,\min}}\sum_{i=0}^{c-1}\left(\ln T/\delta \|\sigma_{\Psi_i}\|^2 + \eta L + 4\alpha\right) \\
&= \log\frac{\|b_T\|_1}{b_{0,\min}}\left(\ln T/\delta \|\sigma\|_2^2 + c\eta L + 4c\alpha\right).
\end{aligned}
$$

Now, note the LHS term $\sum_{t=1}^{T} \sum_{i=0}^{c-1} \frac{\|\nabla_{t,\Psi_i}\|_2^2}{b_{t,i}}$ of (32):

$$\left(\sum_{i=0}^{c-1} \frac{\|\nabla_{t,\Psi_i}\|_2^2}{b_{t,i}}\right)\left(\sum_{i=0}^{c-1} b_{t,i}\right) \geq \left(\sum_{i=0}^{c-1} \|\nabla_{t,\Psi_i}\|_2\right)^2 \geq \sum_{i=0}^{c-1} \|\nabla_{t,\Psi_i}\|_2^2 = \|\nabla_t\|_2^2$$

$$\implies \frac{\|\nabla_t\|_2^2}{\left(\sum_{i=0}^{c-1} b_{t,i}\right)} \leq \sum_{i=0}^{c-1} \frac{\|\nabla_{t,\Psi_i}\|_2^2}{b_{t,i}}.$$

Now, $\sum_{i=0}^{c-1} b_{t,i} = \sum_{i=0}^{c-1} |b_{t,i}| = \|b_t\|_1$, so with probability $1 - 2c\delta$:

$$\sum_{t=1}^{T} \frac{\|\nabla_t\|_2^2}{\|b_T\|_1} \leq \sum_{t=1}^{T} \frac{\|\nabla_t\|_2^2}{\|b_t\|_1} \leq \sum_{t=1}^{T} \sum_{i=0}^{c-1} \frac{\|\nabla_{t,\Psi_i}\|_2^2}{b_{t,i}}$$

$$\implies \sum_{t=1}^{T} \|\nabla_t\|_2^2 \leq \|b_T\|_1 \sum_{t=1}^{T} \sum_{i=0}^{c-1} \frac{\|\nabla_{t,\Psi_i}\|_2^2}{b_{t,i}}$$

$$\leq \|b_T\|_1 \left(\frac{\Delta_1}{\eta} + cH(\delta) + \left(\ln T/\delta \|\sigma\|_2^2 + c\eta L + 4c\alpha\right) \log \frac{\|b_T\|_1}{b_{0,\min}}\right) \tag{35}$$

$$\leq \|b_T\|_1 \left(\frac{\Delta_1}{\eta} + cH(\delta) + \left(\ln T/\delta \|\sigma\|_2^2 + c\eta L + 4c\alpha\right) \log \frac{\|b_T\|_1}{b_{0,\min}}\right). \tag{36}$$

It remains to bound $\|b_T\|_1$. We start again from smoothness of $f$:

$$\Delta_{t+1} - \Delta_t \leq \langle \nabla_t, x_{t+1} - x_t \rangle + \frac{L}{2} \|x_{t+1} - x_t\|^2$$

$$= -\eta \left\langle \nabla_t, \frac{\widehat{\nabla}_t}{b_t} \right\rangle + \frac{\eta^2 L}{2} \left\|\frac{\widehat{\nabla}_t}{b_t}\right\|^2$$

$$= -\eta \left\langle \widehat{\nabla}_t - \xi_t, \frac{\widehat{\nabla}_t}{b_t} \right\rangle + \frac{\eta^2 L}{2} \sum_{i=0}^{c-1} \sum_{j \in \Psi_i} \frac{\widehat{\nabla}_{t,\Psi_j}^2}{b_{t,i}^2}$$

$$= -\eta \left\langle \widehat{\nabla}_t, \frac{\widehat{\nabla}_t}{b_t} \right\rangle + \eta \left\langle \xi_t, \frac{\widehat{\nabla}_t}{b_t} \right\rangle + \frac{\eta^2 L}{2} \sum_{i=0}^{c-1} \frac{\left\|\widehat{\nabla}_{t,\Psi_i}\right\|^2}{b_{t,i}^2}$$

$$= -\eta \sum_{i=0}^{c-1} \sum_{j \in \Psi_i} \frac{\widehat{\nabla}_{t,j}^2}{b_{t,i}} + \eta \sum_{i=0}^{c-1} \sum_{j \in \Psi_i} \frac{\xi_{t,j} \widehat{\nabla}_{t,j}}{b_{t,i}} + \frac{\eta^2 L}{2} \sum_{i=0}^{c-1} \frac{\left\|\widehat{\nabla}_{t,\Psi_i}\right\|^2}{b_{t,i}^2}$$

$$= -\eta \sum_{i=0}^{c-1} \frac{\left\|\widehat{\nabla}_{t,\Psi_i}\right\|^2}{b_{t,i}} + \frac{\eta^2 L}{2} \sum_{i=0}^{c-1} \frac{\left\|\widehat{\nabla}_{t,\Psi_i}\right\|^2}{b_{t,i}^2} + \eta \sum_{i=0}^{c-1} \sum_{j \in \Psi_i} \frac{\xi_{t,j} \widehat{\nabla}_{t,j}}{b_{t,i}}. \tag{37}$$

Note that

$$\sum_{i=0}^{c-1} \sum_{j \in \Psi_i} \frac{\xi_{t,j} \widehat{\nabla}_{t,j}}{b_{t,i}} \leq \frac{1}{2} \sum_{i=0}^{c-1} \sum_{j \in \Psi_i} \frac{\xi_{t,j}^2}{b_{t,i}} + \frac{1}{2} \sum_{i=0}^{c-1} \sum_{j \in \Psi_i} \frac{\widehat{\nabla}_{t,j}^2}{b_{t,i}}$$

$$= \frac{1}{2} \sum_{i=0}^{c-1} \sum_{j \in \Psi_i} \frac{\xi_{t,j}^2}{b_{t,i}} + \frac{1}{2} \sum_{i=0}^{c-1} \frac{\left\|\widehat{\nabla}_{t,\Psi_i}\right\|^2}{b_{t,i}}.$$

Plugging back in, we have

$$\Delta_{t+1} - \Delta_t \le -\frac{\eta}{2} \sum_{i=0}^{c-1} \frac{\left\|\widehat{\nabla}_{t,\Psi_i}\right\|^2}{b_{t,i}} + \eta^2 L \sum_{i=0}^{c-1} \frac{\left\|\widehat{\nabla}_{t,\Psi_i}\right\|^2}{b_{t,i}^2} + \frac{\eta}{2} \sum_{i=0}^{c-1} \frac{\|\xi_{t,\Psi_i}\|^2}{b_{t,i}}.$$

Summing over $T$ and rearranging, we get

$$\sum_{t=1}^{T} \sum_{i=0}^{c-1} \frac{\left\|\widehat{\nabla}_{t,\Psi_i}\right\|^2}{b_{t,i}} \le \frac{2\Delta_1}{\eta} + \sum_{t=1}^{T} \sum_{i=0}^{c-1} \frac{\|\xi_{t,\Psi_i}\|^2}{b_{t,i}} + 2\eta L \sum_{t=1}^{T} \sum_{i=0}^{c-1} \frac{\left\|\widehat{\nabla}_{t,\Psi_i}\right\|^2}{b_{t,i}^2}$$

$$\implies \sum_{t=1}^{T} \sum_{i=0}^{c-1} \frac{\left\|\widehat{\nabla}_{t,\Psi_i}\right\|^2}{b_{t,i}} \le \frac{4\Delta_1}{\eta} + 2 \sum_{t=1}^{T} \sum_{i=0}^{c-1} \frac{\|\xi_{t,\Psi_i}\|^2}{b_{t,i}} + \sum_{t=1}^{T} \sum_{i=0}^{c-1} \left( \frac{4\eta L}{b_{t,i}^2} - \frac{1}{b_{t,i}} \right) \left\|\widehat{\nabla}_{t,\Psi_i}\right\|^2.$$

We can bound $\sum_{t=1}^{T} \sum_{i=0}^{c-1} \left( \frac{4\eta L}{b_{t,i}^2} - \frac{1}{b_{t,i}} \right) \left\|\widehat{\nabla}_{t,\Psi_i}\right\|^2$ as follows. Consider $i \in [c]$. Let $\tau_i = \max\{t \le T \mid b_{t,i} \le 4\eta L\}$ so that $t \ge \tau_i$ implies $b_{t,i} > 4\eta L \iff \frac{4\eta L}{b_{t,i}^2} < \frac{1}{b_{t,i}}$:

$$\sum_{t=1}^{T} \left( \frac{4\eta L}{b_{t,i}^2} - \frac{1}{b_{t,i}} \right) \left\|\widehat{\nabla}_{t,\Psi_i}\right\|^2 = \sum_{t=1}^{\tau_i} \left( \frac{4\eta L}{b_{t,i}^2} - \frac{1}{b_{t,i}} \right) \left\|\widehat{\nabla}_{t,\Psi_i}\right\|^2 + \sum_{t=\tau_i+1}^{T} \left( \underbrace{\frac{4\eta L}{b_{t,i}^2} - \frac{1}{b_{t,i}}}_{<0} \right) \left\|\widehat{\nabla}_{t,\Psi_i}\right\|^2$$

$$\le \sum_{t=1}^{\tau_i} \left( \frac{4\eta L}{b_{t,i}^2} - \frac{1}{b_{t,i}} \right) \left\|\widehat{\nabla}_{t,\Psi_i}\right\|^2$$

$$\le 4\eta L \sum_{t=1}^{\tau_i} \frac{\left\|\widehat{\nabla}_{t,\Psi_i}\right\|^2}{b_{t,i}^2}$$

$$\le 8\eta L \log \frac{b_{\tau_i,i}}{b_{0,i}} \le 8\eta L \log \frac{4\eta L}{b_{0,i}}.$$

Hence, we have

$$\sum_{t=1}^{T} \sum_{i=0}^{c-1} \frac{\left\|\widehat{\nabla}_{t,\Psi_i}\right\|^2}{b_{t,i}} \le \frac{4\Delta_1}{\eta} + 2 \sum_{t=1}^{T} \sum_{i=0}^{c-1} \frac{\|\xi_{t,\Psi_i}\|^2}{b_{t,i}} + 8\eta L \sum_{i=0}^{c-1} \log \frac{4\eta L}{b_{0,i}}.$$

Consider the LHS

$$\sum_{t=1}^{T} \sum_{i=0}^{c-1} \frac{\left\|\widehat{\nabla}_{t,\Psi_i}\right\|^2}{b_{t,i}} = \sum_{t=1}^{T} \sum_{i=0}^{c-1} \frac{b_{t,i}^2 - b_{t-1,i}^2}{b_{t,i}} = \sum_{t=1}^{T} \sum_{i=0}^{c-1} b_{t,i} - \frac{b_{t-1,i}^2}{b_{t,i}}$$

$$\ge \sum_{t=1}^{T} \sum_{i=0}^{c-1} b_{t,i} - \frac{b_{t-1,i}^2}{b_{t-1,i}} = \sum_{t=1}^{T} \sum_{i=0}^{c-1} b_{t,i} - b_{t-1,i}$$

$$= \sum_{i=0}^{c-1} \sum_{t=1}^{T} b_{t,i} - b_{t-1,i} = \sum_{i=0}^{c-1} b_{T,i} - b_{0,i}$$

$$= \|b_T\|_1 - \|b_0\|_1.$$

Hence, we have

$$\|b_T\|_1 \le \|b_0\|_1 + \frac{2\Delta_1}{\eta} + \sum_{i=0}^{c-1} \sum_{t=1}^{T} \frac{\|\xi_{t,\Psi_i}\|^2}{b_{t,i}} + 8\eta L c \log \frac{4\eta L}{b_{0,\min}}.$$

It remains to bound $\sum_{t=1}^{T} \frac{\|\xi_{t,\Psi_i}\|^2}{b_{t,i}}$ for each $i \in [c]$. Recall from (30), with probability at least $1 - \delta$

$$\sum_{s=1}^{t} \|\xi_{t,\Psi_i}\|^2 \leq \sum_{s=1}^{t} \|\widehat{\nabla}_{t,\Psi_i}\|^2 + 4 \|\sigma_{\Psi_i}\|^2 \log \frac{1}{\delta}, \ \forall t \leq T.$$

We have with probability at least $1 - 2c\delta$,

$$
\begin{aligned}
\sum_{t=1}^{T} \frac{\|\xi_{t,\Psi_i}\|^2}{b_{t,i}} &= \sum_{t=1}^{T} \frac{\|\xi_{t,\Psi_i}\|^2}{\sqrt{b_{0,i}^2 + \sum_{s=1}^{t} \|\widehat{\nabla}_{s,\Psi_i}\|^2}} \\
&\overset{(1)}{\leq} \sum_{t=1}^{T} \frac{\xi_{t,i}^2}{\sqrt{b_{0,i}^2 + \left( \sum_{s=1}^{t} \|\xi_{s,\Psi_i}\|^2 - 4 \|\sigma_{\Psi_i}\|^2 \log \frac{1}{\delta} \right)^+}} \\
&\leq \frac{8 \|\sigma_{\Psi_i}\|^2 \log \frac{1}{\delta}}{b_{0,i}} + 2\sqrt{2} \sqrt{\sum_{s=1}^{T} \|\xi_{s,\Psi_i}\|^2} \\
&\overset{(2)}{\leq} \frac{8 \|\sigma_{\Psi_i}\|^2 \log \frac{1}{\delta}}{b_{0,i}} + 4\sqrt{\|\sigma_{\Psi_i}\|^2 T + \|\sigma_{\Psi_i}\|^2 \log \frac{1}{\delta}},
\end{aligned}
$$

where (1) is due to (27) and (2) is due to Lemma (29). Hence, we have that with probability at least $1 - 2c\delta$,

$$
\begin{aligned}
\|b_T\|_1 &\leq \|b_0\|_1 + \frac{2\Delta_1}{\eta} + \sum_{i=0}^{c-1} \frac{8 \|\sigma_{\Psi_i}\|^2 \log \frac{1}{\delta}}{b_{0,i}} + \sum_{i=0}^{c-1} 4\sqrt{\|\sigma_{\Psi_i}\|^2 T + \|\sigma_{\Psi_i}\|^2 \log \frac{1}{\delta}} + 8\eta L c \log \frac{4\eta L}{b_{0,\min}} \\
&\leq \|b_0\|_1 + \frac{2\Delta_1}{\eta} + \frac{8 \log \frac{1}{\delta}}{b_{0,\min}} \sum_{i=0}^{c-1} \|\sigma_{\Psi_i}\|^2 + 4\sqrt{T} \sum_{i=0}^{c-1} \|\sigma_{\Psi_i}\| + \sqrt{\log \frac{1}{\delta}} \sum_{i=0}^{c-1} \|\sigma_{\Psi_i}\| + 8\eta L c \log \frac{4\eta L}{b_{0,\min}} \\
&= 4\sqrt{T} \sum_{i=0}^{c-1} \|\sigma_{\Psi_i}\| + \underbrace{\|b_0\|_1 + \frac{2\Delta_1}{\eta} + \frac{8 \log \frac{1}{\delta}}{b_{0,\min}} \|\sigma\|_2^2 + \sqrt{\log \frac{1}{\delta}} \sum_{i=0}^{c-1} \|\sigma_{\Psi_i}\| + 8\eta L c \log \frac{4\eta L}{b_{0,\min}}}_{=: I(\delta)}.
\end{aligned}
$$

Hence, we can combine (36) with the bound for $\|b_T\|_1$ to get that with probability $1 - 6c\delta$:

$$
\begin{aligned}
\sum_{t=1}^{T} \|\nabla_t\|_2^2 &\leq \|b_T\|_1 \left( \frac{\Delta_1}{\eta} + H(\delta) + \left( \ln T/\delta \|\sigma\|_2^2 + c\eta L + 4c\sigma_{\max} \sqrt{c \log \frac{1}{\delta}} \right) \log \frac{\|b_T\|_1}{b_{0,\min}} \right) \\
&\leq \left( 4\sqrt{T} \sum_{i=0}^{c-1} \|\sigma_{\Psi_i}\| + I(\delta) \right) \cdot \\
&\quad \left( \frac{\Delta_1}{\eta} + H(\delta) + \left( \ln T/\delta \|\sigma\|_2^2 + c\eta L + 4c^{3/2}\sigma_{\max} \sqrt{\log \frac{1}{\delta}} \right) \log \left( \frac{4\sqrt{T} \sum_{i=0}^{c-1} \|\sigma_{\Psi_i}\| + I(\delta)}{b_{0,\min}} \right) \right).
\end{aligned}
$$

Dividing both sides by $T$, we get the theorem that with probability $1 - 6c\delta$:

$$
\frac{1}{T} \sum_{t=1}^{T} \|\nabla_t\|_2^2 \leq G(\delta) \cdot \left( \frac{4 \sum_{i=0}^{c-1} \|\sigma_{\Psi_i}\|}{\sqrt{T}} + \frac{I(\delta)}{T} \right), \text{ where } G(\delta) \text{ and } I(\delta) \text{ are polylog terms:}
$$

$$
G(\delta) := \frac{\Delta_1}{\eta} + H(\delta) + \left( \ln T/\delta \|\sigma\|_2^2 + c\eta L + 4c^{3/2}\sigma_{\max} \sqrt{\log \frac{1}{\delta}} \right) \log \left( \frac{4\sqrt{T} \sum_{i=0}^{c-1} \|\sigma_{\Psi_i}\| + I(\delta)}{b_{0,\min}} \right)
$$

$$
I(\delta) := \|b_0\|_1 + \frac{2\Delta_1}{\eta} + \frac{8 \log \frac{1}{\delta}}{b_{0,\min}} \|\sigma\|_2^2 + \sqrt{\log \frac{1}{\delta}} \sum_{i=0}^{c-1} \|\sigma_{\Psi_i}\| + 8\eta L c \log \frac{4\eta L}{b_{0,\min}}
$$

$$
H(\delta) := \sum_{i=0}^{c-1} \left( \ln (T/\delta) \|\sigma_{\Psi_i}\|^2 + 2\alpha \right) \left( \frac{8 \|\sigma_{\Psi_i}\|^2 \log \frac{1}{\delta}}{b_{0,i}^2} + 2 \log \left( 1 + \|\sigma_{\Psi_i}\|^2 T + \|\sigma_{\Psi_i}\|^2 \log \frac{1}{\delta} \right) \right). \qquad \square
$$

