# OpenReview forum: "Lean and Mean Adaptive Optimization via Subset-Norm and Subspace-Momentum with Convergence Guarantees"
_ICML.cc/2025/Conference — ICML 2025 poster_

### Official Review · Reviewer_oW9K · 2025-02-25

**Overall Recommendation:** 2

**Summary:**

This paper introduces two complementary adaptive optimization techniques that reduce the memory footprint of optimizer states while accelerating LLM large-scale neural network training.

- Subset-Norm (SN): A generalization of AdaGrad-Norm and AdaGrad-Coordinate that shares step sizes across subsets of parameters. SN reduces AdaGrad’s memory footprint from $O(d)$ to $O(\sqrt{d})$ while maintaining strong theoretical convergence guarantees under sub-Gaussian noise assumptions.

- Subspace-Momentum (SM): A technique that applies momentum in a low-dimensional subspace while using SGD in the orthogonal complement. The authors prove high-probability convergence guarantees for SM under standard assumptions.

Empirical results demonstrate that combining SN and SM achieves Adam’s validation perplexity on LLaMA 1B with half the training tokens (6.8B vs. 13.1B) while reducing Adam’s optimizer state memory footprint by more than 80%, with minimal additional hyperparameter tuning.

**Claims And Evidence:**

The claims in the paper are generally well-supported by:

- Theoretical analysis: The authors provide strong mathematical guarantees for both SN and SM. The convergence proofs are well-presented and adhere to standard assumptions in stochastic optimization.
- Empirical validation: The experiments on pre-training and fine-tuning large-scale LLMs demonstrate clear improvements in memory efficiency while maintaining or improving convergence speed.
- Comparison to baselines: SN and SM are compared against AdaGrad, Adam, and GaLore, showing consistent improvements in both perplexity and memory efficiency.

However, some additional validations would strengthen the claims:

- CV benchmarks : The proposed methods should be tested on ImageNet/CIFAR-10 with ViTs or ResNets to evaluate performance in a broader range of architectures and compare with well-established optimizers in vision tasks.
- Robustness: The paper lacks hyperparameter robustness analysis, which would help practitioners understand how sensitive SN and SM are to tuning.
- Comparison with recent memory-efficient optimizers: The paper does not compare against newer low-memory optimizers such as Adam-mini or MicroAdam, which have been proposed to reduce the memory footprint of optimizers.

**Essential References Not Discussed:**

Missing citations to recent memory-efficient optimizers such as Adam-mini.

No discussion of quantization-based approaches like 8-bit Adam, which also aim to reduce optimizer memory.

**Experimental Designs Or Analyses:**

- The experimental design is strong, with multiple scales of LLaMA models tested.
- The results clearly demonstrate the memory efficiency of SN and SM, particularly on LLM pre-training tasks.
- However, there are no details on the numerical precision used during training (e.g., FP32, BF16, FP16). Clarifying this would be important since precision affects memory consumption.
- Hyperparameter tuning details are unclear:
	- How were the “red LR” values tuned? Do you adjust them based on instability?
	- How should practitioners tune SN and SM hyperparameters? A guideline for selecting subset sizes and subspaces would be helpful.

**Methods And Evaluation Criteria:**

The proposed SN and SM methods are well-formulated and grounded in existing literature on AdaGrad-style optimizers and gradient compression techniques.

The authors evaluate SN and SM across multiple model sizes (from 60M to 7B parameters) and compare optimizer memory footprints, training perplexity, and convergence rates. The evaluation is relevant, but additional benchmarks (e.g., ImageNet, CIFAR-10) would improve generalization.

**Other Comments Or Suggestions:**

The notations in Table 3 and Table 4 are inconsistent (Adam vs. AdamW). This should be clarified.

**Other Strengths And Weaknesses:**

**Strengths:**
- Well-motivated problem with significant practical relevance for training LLMs.
- Strong theoretical backing with convergence guarantees.
- Empirical results demonstrate state-of-the-art memory savings.

**Weaknesses**:
- No evaluation on computer vision benchmarks (ViT, ResNet).
- No comparison with MicroAdam or Adam-mini.
- Lack of clear recommendations for tuning SN and SM hyperparameters.
- Inconsistent notation between tables (Adam vs. AdamW).
- No discussion of numerical precision used in training.

**Questions For Authors:**

(1)	How does SN and SM compare to recent memory-efficient optimizers like MicroAdam and Adam-mini?

(2) Can SN and SM be effectively applied to ViTs and ResNets for computer vision tasks?

(3) What numerical precision was used during training (BF16, FP16, FP32)?

(4) How do you tune hyperparameters for SNSM?

(5) How was the “red LR instability” tuned?

(6)	What heuristics can be used to select the optimal subset size in SN?

**Relation To Broader Scientific Literature:**

The paper is well-positioned within the literature on adaptive optimizers (Adam, AdaGrad) and memory-efficient optimizers (GaLore, FLORA, GRASS). However, no comparisons are made to newer memory-efficient optimizers like MicroAdam or Adam-mini, which should be included.

**Theoretical Claims:**

The convergence proofs for SN and SM seem correct.

---

> ### Author Rebuttal · Authors · 2025-03-31
>
> We appreciate the reviewer’s careful consideration and constructive feedback. Below, we clarify points raised and address specific concerns.
> ___
> > No discussion of quantization-based approaches like 8-bit Adam
>
> Due to space limit, we cite and discuss quantization approaches (and more) in the related works section on page 13 of the Appendix. This is an important orthogonal direction that can be combined with our method for further future improvement.
>
> > Question 1: MicroAdam and Adam-mini comparisons
>
> We provide brief comparisons to MicroAdam and Adam-mini in the related works section on page 13, but will provide more detailed comparison for Adam-mini due to its relevance as also pointed out by other reviewers in our revision.
> - For MicroAdam, while it delivers good performance for fine-tuning, it has a limitation for the pre-training LLM setting (see page 10 of their paper) which is a setting we want to focus on due to the relative lack of effective memory efficient optimizers.
> - For Adam-mini, we want to note that it is concurrent work to ours, to be published in ICLR this year, but we agree with the reviewer that Adam-mini is highly relevant and including a detailed comparison in the main body improves the quality of our paper. We will include a more detailed comparison in the main body of our revision with a draft as in our response to reviewer 6d8c.
>
> > Question 2: vision tasks
>
> - While our main focus is on large models that are more typical to language models where memory is often a bottleneck, to address the reviewer’s request for additional experiments, we conducted further evaluations using the DiT-L/2 model (458M) from [facebookresearch/DiT](https://github.com/facebookresearch/DiT) on a setup with batch size 2048, image size 64, and 8×A6000 GPUs.
> We compared our method (SNSM) with Adam using the same training configuration as in the paper. As shown in the table below, SNSM outperforms Adam in FID similarly to LLM tasks:
>
> | FID/Iter | 200k  | 300k  | 400k  | 500k  | 700k  |
> |-----------|-------|-------|-------|-------|-------|
> | Adam | 56.69 | 56.63 | 40.69 | 41.15 | 39.61 |
> | AdamSNSM | 66.76 | 66.31 | 34.05 | 32.31 | **32.26** |
>
> - We further evaluate Adam, AdamSN, and AdamSNSM (rank 64 and no update gap) by training “vit_base_patch16_224” (~85M params) from the `timm` library on the `CIFAR100` dataset for 10 epochs with bs64 and wd0 on a 2x4090 machine. We tune the lr across {1e-3, 5e-3, 1e-4, 5e-4, 5e-5, 1e-5} grid for all methods.
>
> | | Adam  | AdamSN  | AdamSNSM  |
> |-----------|-------|-------|-------|
> |Best Val Acc| 43.3%| 45.2% |  **45.6%**|
>
> - Please see the loss curve here: https://imgur.com/a/4rtgxOx. These preliminary experiments show promising results for the application of our methods to vision tasks.
>
> > Question 3: numerical precision
>
> - We use BF16 as mentioned in Appendix B.2 and ensure that all our experiments across different optimizers use consistent numerical precision for fair comparisons. Due to space constraint, a lot of the experimental setup and implementation details are moved to the Appendix. We will make this clearer in the main body of our revision.
>
> > Question 4: tune hyperparameters
>
> - Tuning effort is provided in Appendix B.2. We use a similar setup (rank, projection update gap, etc.) to Zhao et al 2024 and try to minimize tuning (in the spirit of this resource constraint setting) and only tune for the learning rate. However, we perform extensive ablation studies (Section 5.4 and Appendix C) to show that performance can further be improved with additional tuning. We are grateful that reviewers t4t5 and 6d8c looked at these ablations and considered them "very solid" and "quite extensive and contains important details."
>
> > Question 5: red LR
>
> - The red LR simply highlights the results where the best LR found for the larger model differs from the smaller model and hence tuned similarly to the other LR. It is desirable for hyperparameters to transfer from smaller models to larger models. However, this is a complex topic in general (e.g. https://arxiv.org/abs/2203.03466).
>
> > Question 6: SN heuristics
>
> - On line 213 of the paper, we provide a simple heuristic of grouping the latent dimension together.  In Figure 5, we examine different subset sizes where one can tune to get a better result. However, our heuristic seems to be able to capture much of the performance. In general, we find that choosing a subset size around sqrt(mn) for each block seems to be a good heuristic and balance that is supported by our analysis.
>
> ___
> We sincerely appreciate your detailed reviews and constructive suggestions. We believe the discussion, analysis, explanations and additional experiments clarify your concerns and improve the quality of our submission. We hope this provides sufficient reasons to raise the score.

---

> > ### Comment · Reviewer_oW9K · 2025-04-03
> >
> > Thank you for your response.
> >
> > Q2. Thank you for the additional experiments. However, I was referring more to benchmarks like CIFAR-10—a small but widely adopted dataset that enables easier and more consistent comparisons—or ImageNet, using architectures such as ResNets or ViTs, which are considered standard in the field.
> >
> > Q4. I still find the description of the tuning process insufficiently detailed to ensure reproducibility.
> >
> > Overall, I still struggle to position this new method relative to Adam-mini, particularly in terms of computational complexity and memory footprint.

---

> > > ### Author Response · Authors · 2025-04-03
> > >
> > > > Q2: CIFAR-10
> > >
> > > The model that we used, [vit_base_patch16_224](https://huggingface.co/timm/vit_base_patch16_224.augreg2_in21k_ft_in1k), is the vision transformer model introduced in the paper “An Image is Worth 16x16 Words: Transformers for Image Recognition at Scale” (https://arxiv.org/abs/2010.11929v2). We use the randomly initialized version of the ViT above to train [CIFAR-100](https://www.cs.toronto.edu/~kriz/cifar.html) which is the 100 classes version of CIFAR-10, where it provides a good balance between CIFAR-10 and ImageNet. However, we further provide our results on CIFAR-10 on the ViT below:
> > >
> > > | Best val accuracy 10 epochs   | Adam   | AdamSN | AdamSNSM (r=64, g=1000) |
> > > |-----------|--------|--------|--------------------------|
> > > | CIFAR100  | 43.30% | 45.20% | **45.60%**               	|
> > > | CIFAR10   | 69.02% | 69.18% | **71.21%**               	|
> > > | Peak Mem (bs 64) | 9.288GB | 8.886GB | **8.878GB** |
> > >
> > > The loss curves for CIFAR10 is shown at: (1) train loss https://imgur.com/a/7ATZ4o7; and (2) val loss https://imgur.com/a/zRuBZgu. This is similar to CIFAR100.
> > >
> > > We want to emphasize again that our paper focuses primarily on language models and transformers due to the use of much larger models in language tasks (billions of parameters) than vision tasks (millions of parameters), where memory is a major bottleneck for large models.
> > >
> > > > Q4. I still find the description of the tuning process insufficiently detailed to ensure reproducibility.
> > >
> > > We already provided all the codes in the supplementary material and all the parameter choices for the tuning in our paper along with extensive ablation studies. We would appreciate it if the reviewer can describe more concretely the missing information. We will be happy to include that in our revision to make reproducing our results as easy as possible.
> > >
> > > To reiterate Appendix B here, $\beta_1$ and $\beta_2$ are not tuned and set according to the default of Adam in PyTorch ($\beta_1=0.9, \beta_2 =0.999$). Subspace rank and update gap are not tuned and set according to the default of GaLore (rank as in Table 3, update gap=200). We run all main experiments on BF16 format, gradient clipping of 1.0 (standard in training LLMs like LLaMA/DeepSeek and not tuned), and batch sizes 512. We use the same cosine learning rate decay and warm-up steps schedule as GaLore (Zhao et al., 2024) and Llama (Touvron et al., 2023; Dubey et al., 2024). We **only tune** for the learning rate within the set {0.1, 0.05, 0.01, 0.005, 0.001} (except for AdaGrad algorithms that need larger lrs). There are ablations on the effect of changing some of the parameters like rank and update gap, but they are not tuned for the experiments reported in Table 3. This information can be found in Appendix B1 and B2 of our paper. We will rewrite those sections to make the information more streamlined and clearer.
> > >
> > > > Comparison with Adam-mini
> > >
> > > As in our response to reviewer 6d8c, again while emphasizing that **Adam-mini is concurrent work to ours**, we will include a more detailed comparison to Adam-mini in our revision.
> > >
> > > In particular, Adam-mini is most comparable to our AdamSN in both computational complexity and memory footprint. The state of the Adam algorithm consists of two vectors of the same length: m and v. Both AdamSN and Adam-mini keep m the same and compress v to a negligible size (<1% of the original). Thus, in terms of memory for the optimizer state, both AdamSN and Adam-mini use about 50% that of Adam. In terms of computational complexity, they both compute m in the same way as Adam and spend negligible extra time per iteration to maintain v.
> > >
> > > Our AdamSNSM goes beyond this and also compresses the m vector. Our size for m is less than 50% for rank=1/4 (which is the choice that we make in our experiments -- see also Tables 3 and 4) of the original. Combining both SN and SM, the memory for the optimizer state of AdamSNSM is less than half that of Adam-mini. In computational complexity per iteration, the SM technique introduces a small extra overhead of less than 2.5% per iteration but due to a better preconditioning as suggested by reviewer t4t5, it converges faster than Adam on pre-training tasks by up to 50% fewer iterations.
> > >
> > > ___
> > > We thank the reviewer again for your time and effort. We believe that the discussion and additional experiments improve the quality of our submission and hope this provides sufficient reasons to raise the score.

---

### Official Review · Reviewer_t4t5 · 2025-03-04

**Overall Recommendation:** 4

**Summary:**

This paper proposes two modifications to AdaGrad: 1) instead of coordinate wise adaptive learning rate, subset norm adaptative learning rate can provide adaptive learning rate scaling for different subsets of parameters 2) similar to GaLore, it keeps momentum in low rank and recovers momentum by up-projection, whose projection matrix is periodically updated via SVD.

**Claims And Evidence:**

The experiments and ablations are very solid. I have nothing to complain about.

**Essential References Not Discussed:**

Besides Galore, a few of its follow-ups are highly relevant to the discussion too. I believe it will benefit the readers if the authors can discuss the difference and commonality compared to those.

**Experimental Designs Or Analyses:**

The experiment designs follow the common practice and ablation studies are abundant, nothing much can be complaint about.

**Methods And Evaluation Criteria:**

The method makes sense and evaluation all follows current common practices. I don't see anything suspicious in particular.

**Other Comments Or Suggestions:**

N/A

**Other Strengths And Weaknesses:**

The proposed SN and S momentum are simple and straightforward, though they are not really new or surprising. especially subspace momentum, this is an incremental on top of GaLore.

For this method to be truly convincing, I would like to see wall clock speed up on pretraining task as well as peak memory for different implementations. Despite of momentum being saved in low rank, it might still have to materialize larger tensor due to intermediate computation of residual r, which cancels out the benefit of lower memory consumption.

Another setting that is important but not really discussed in the paper is how friendly the proposed method is to distributed training.

This method also introduce extra hyperparameter tuning, default hyperparameters of AdaGrad can't be reused and non-trivial effort to find the optimal block size and etc. It also inherits all the problems of GaLore in practice: SVD can become a real bottleneck when weight matrices become large, randomized/approximated SVD has no guarantee that it wouldn't degrade the perplexity as shown in appendix.

Nonetheless, I am still leaning towards acceptance due to the comprehensive ablations and detailed guidelines in hyperparameter choices and etc. Theoretical propositions also seem solid and interesting, though I am not completely familiar with the mathematical tools/framework used in this paper.

**Questions For Authors:**

N/A

**Relation To Broader Scientific Literature:**

this work is related to a series of optimizers proposed to pretrain large language models faster and more efficiently, the subspace momentum can be seen as a form of preconditioning, which fits in the broader optimizer literatures, such Muon, MARS and Lion.

**Theoretical Claims:**

I checked Theorem 3.1 and 4,1, they seem correct. Though I have not carefully checked the proofs in appendix, the conclusion fits my expectation to the convergence rate, which is similar to that of the SGD's convergence under smooth conditions.

One complaint here is Theorem 3.1 is highly dependent on the variance of the stochastic noise, which might render the bound vacuous.

Theorem 4.1 has no dependency on k the rank of the subspace, I would imagine it should because when k is full rank it should exactly recover the SGD baseline case.

There is also no analysis on how the projection matrix would impact the convergence.

---

> ### Author Rebuttal · Authors · 2025-03-31
>
> We appreciate the reviewer’s careful consideration and constructive feedback. Below, we clarify points raised and address specific concerns.
> ____
> > Theorem 3.1 is highly dependent on the variance of the stochastic noise
>
> - The reviewer is right that the bounds depend on the noise and this is standard for any bounds on stochastic gradient descent. The novelty is that we show grouping the coordinates based on noise magnitudes can lead to a better performance. Our experiments show that our heuristic successfully isolates the noise into a small number of groups and perhaps as a result, delivers better training results.
>
> > Theorem 4.1 has no dependency on k the rank of the subspace
>
> - This is a limitation that we admit in the discussion section of the paper: the worst case convergence of SM is simply the SGD with momentum bound. We believe that a better analysis (that probably depends on a better subspace choice rather than the worst case one) could potentially show improved rates over naive SGDm. Our experiments show promising results for the top-k singular vector subspace so we believe that this is an interesting point for future works to show theoretical improvement.
> - And similarly to our response to Reviewer 6d8c, this is a limitation that is general to the class of algorithms that utilize momentum.
> This is an important open challenge for the community at large because the theoretical bound for momentum and without momentum are the same. We believe that any advance for SGD with momentum will lead to a better understanding of SM in our setting.
>
> > Besides Galore, a few of its follow-ups are highly relevant to the discussion too.
>
> - Please refer to our response to Reviewer 6d8c. We will add additional comparisons to Table 3 of our revision and a more detailed comparison with our Adam-mini in our main body.
>
> > I would like to see wall clock speed up on pretraining task as well as peak memory for different implementations.
>
> We provide the per iteration time, peak memory (via nvidia-smi), and time to Adam’s val perplexity after 100K steps for the 1B model for each method on a 2x4090 machine with the same setup as the paper (seq length 256, total batch size 512, micro batchsize 16) below:
> |                     	| Adam    	| AdamSNSM  (Gap=5000) | AdamSNSM  (Gap=200)| AdamSN                 	| GaLore   (Gap=200)|
> |-------------------------|-------------|----------------------------|----------------------------|----------------------------|----------------------------|
> | Time for 1K iters  	| 7426s   | 7465 s            |  7624 s	| **7399s**              	| 7827s              	|
> | Time per iteration  	| 7.43 s/it   | 7.47 s/it     |         7.62 s/it	| **7.39 s/it**              	| 7.83 s/it              	|
> | Time to perplexity < 16 | ~206.4 hrs  (100K iters)| ~136.9 hrs (<66K iters) | **~101.6 hrs (<48K iters)**  | ≈118.9 hrs (<58K iters)   | >217 hrs (>100K iters)	|
> |Peak mem	| 21.554 GB/GPU | **16.642 GB/GPU**    | **16.642 GB/GPU**     	| 19.193 GB/GPU          	| 18.187 GB/GPU          	|
>
>
> - The results above show that the dimensionality reduction can achieve some speedup and the improved performance can offset some overhead of SVD. Furthermore, as demonstrated in Table 6 on page 8, our methods do not require the SVD update gap to be as frequent as GaLore.
> - Peak memory for batch size 1 is also shown in Figure 6 on page 16.
>
> > Another setting that is important but not really discussed in the paper is how friendly the proposed method is to distributed training.
>
> - Thank you for bringing up this great point. Our non-heuristic version of the SN algorithm (Algorithm 5) is FSDP-friendly due to being shape agnostic and can group coordinates locally. Our supplementary material contains the code for that version (`adamw_sng.py`). Figure 5 shows that local grouping with group size of around $\sqrt(d)$ gives good results.
> - A brief discussion regarding SM’s distributed training is on line 932 where it is more subtle and depends on the subspace choice where subspace that aligns with the standard bases would work on distributed setup but there are other tradeoffs.
> - We will include additional discussion regarding this point in our revision.
>
> > This method also introduce extra hyperparameter tuning...
>
> - We do not tune for the block size as it is simply the row/column of each matrix. While we show that tuning can bring additional benefits as shown in Figure 5, we can already attain a large part of that gain with no tuning via our heuristics on line 213. Finally, due to the modularity of our approaches, we can even abandon the use of momentum and still achieve strong performance relative to vanilla Adam with RMSPropSN as shown in Table 3 and 8 (versus vanilla RMSProp).
> ___
> We sincerely appreciate your detailed reviews and constructive suggestions. We believe that the discussion and additional experiments improve the quality of our submission and hope this provides sufficient reasons to raise the score.

---

### Official Review · Reviewer_KhrK · 2025-03-12

**Overall Recommendation:** 2

**Summary:**

This work studies two modifications to widespread adaptive optimization meta-algorithm procedures, with the goal of reducing the memory footprint of adaptive optimization while simultaneously improving performance.

The first modification is referred to as adaptive subset-norm (SN) stepsizes, which is similar to existing methods such as Adam-Mini. Rather than maintaining an adaptive learning rate for each parameter (as in traditional Adagrad) or maintaining only a single adaptive learning rate for all parameters (Adagrad-Norm), they propose partitioning parameters into groups and maintaining an adaptive coordinate for each group. They state that their analytical results (e.g. Thm. 3.1) show that by choosing an intermediate number of groups, they can obtain better convergence guarantees. This reduces the memory footprint of maintaining adaptive stepsizes from $O(d)$ for $d$ the number of parameters to $O(c)$ for $c$ the number of groups.

Their second proposed method, subspace momentum, computes the momentum update in some $k$-dimensional subspace instead of the full-dimensional space, similar to existing methods like GaLore. However the authors propose simultaneously performing a step of (stochastic) gradient descent in the orthogonal complement of the selected subspace on each iteration. This allows them to obtain convergence guarantees (Thm. 4.1) in contrast with other methods which restrict momentum to some low-dimensional subspace, which cannot guarantee convergence without stronger assumptions about the span of this low-dimensional subspace relative to the objective function. This reduces the memory cost of maintaining historical momentum terms from $O(d)$ to $O(k)$. They suggest using the leading-$k$ singular subspace of a stochastic gradient sample to set the subspace, which is an intuitive rule and closely related to GaLore.

**Claims And Evidence:**

Claim 1: Subset-norm (SN) can improve performance over either Adagrad or Adagrad-Norm.

Evidence:
- Thm. 3.1, which the authors describe as capturing a tradeoff between the number of subsets into which parameters are partitioned and the coordinate-wise stochastic gradient noise.
- Fig. 1, which shows that Adam+SN (e.g. Adagrad+SN with full-dim. momentum) minimizes validation perplexity during model pre-training compared with Adam and also has a lower memory footprint (2.6 Gb vs 5 Gb).

Claim 2: Subspace-momentum (SM) can reduce the memory footprint of momentum while still maintaining convergence guarantees.

Evidence:
- Thm. 4.1, which shows that SM is still guaranteed to converge in appropriate settings.
- Fig. 1, which shows that Adam+SNSM minimizes validation perplexity in fewest iterations and has the lowest memory footprint out of Adam, GaLore, and Adam+SN.

**Essential References Not Discussed:**

Adam-mini only mentioned briefly in the appendix, but I think the main idea behind parameter partitioning is as closely-related to SN as GaLore is to SM. I think given the strength of the connection to Adam-mini and the level of discussion that the authors devote to GaLore in the main body, it would be appropriate to discuss Adam-mini in the main body so that readers are aware of the relevant work.

**Experimental Designs Or Analyses:**

I did not check the soundness beyond what is presented in the main body.

**Methods And Evaluation Criteria:**

They consider pre-training of LLaMa models on the C4 dataset. This seems reasonable given their motivating interest in reducing memory footprints for large-scale models. It would be interesting to see if their modifications also improve performance on smaller models/other architectures, as their claims suggest that such improvements should hold in generality, but this may be out-of-scope.

**Other Comments Or Suggestions:**

Parameter $k$ is referenced in “Our Contributions” before it is defined, even informally (line 90, left column). A reasonable reader can infer that $k$ must be related to the dimension of the momentum subspace, but a few words to clarify what is meant by $k$ would be an appropriate addition.

The note about norm notation (norm without subscript defaults to ell-2) is currently located at 126 left-hand column. I would suggest moving it up to the first paragraph on notation: when I was reading the statement of Thm. 3.1, I was very confused about whether the fact that some norms had subscripts and others did not meant that I should assume these norms are different. I could not find the note about norm notation because I looked in the notation paragraph, rather in the sub-Gaussian noise assumption paragraph where it is currently located. I might suggest switching to consistent norm notation within the same equation; in the statement of Thm. 3.1, all global-vector quantities have subscript-norms, while all subset quantities do not have subscripts, potentially adding to readers’ confusion.

**Other Strengths And Weaknesses:**

Strengths:
- The problem studied (“How can adaptive optimization be improved and made more memory-efficient for large models?”) is very well-motivated.
- The authors focus on adding theoretical analysis to heuristics that have begun to be explored in practice.
- The numerical results show that Adam+SMSN yields good training performance while also having a drastically smaller memory footprint.

Weaknesses:

My overall impression of the paper is that its main contribution is to combine two pre-existing heuristics that already existed in literature. The authors argue that their analysis makes their approach more theoretically-grounded, but the most novel analysis about stochastic noise (Thm. 3.1) is not utilized at all in their implementation, which instead chooses a simple heuristic partitioning of parameters based on matrix rows/columns. If I am mis-understanding the implications of their theory, or not seeing how it is informing their choice of parameter grouping, then I would be happy to increase my score (see Questions).

- Both SN and SM are closely related to existing methods, and the authors argue that one of the central contributions of this work is new theory that better explains these methods. Given the fact that the authors hold up the analysis as a significant portion of the contribution, I think the intended take-aways from the analysis need to be better-explained in text. For example, I found the explanation of the tradeoff between parameter partitioning after Thm. 3.1 to be confusing.
- Moreover, the practical implementations used in the paper seem very divorced from the theory; it is my understanding that when choosing parameter partitions, the authors merely group matrix-valued parameters into either rows or columns (depending on which is larger), whereas the guarantees in Thm. 3.1. The fact that this is the proposed partition also weakens the argument made in the appendix that the SN partitioning scheme is supposedly more theoretically-motivated than the scheme in Adam-mini.

**Questions For Authors:**

As discussed in Strengths/Weaknesses, if the authors can clarify the insights provided by their theory and whether these insights are leveraged in their implementation, then I am open to increasing my score.

1. Can you please explain how the result in Thm. 3.1 expresses a trade-off between partition strategies and stochastic gradient noise? It almost seems to me as if apart from the 4-th order terms in $G(\delta)$, every other part of the guarantee gets worse as c increases: as $c\rightarrow d$, $\sum\_i ||\sigma\_{\Psi\_i} ||\_2 \rightarrow ||\sigma||\_1$ whereas for $c\rightarrow 1$, $\sum_i ||\sigma_{\Psi_i} ||_2 \rightarrow ||\sigma||_2$ where $||\sigma||_2 \leq ||\sigma||_1$. On top of that, for $\delta$ fixed, the probability of success $1-O(c\delta)$ gets smaller as $c$ grows, and of course all terms with explicit dependence on $c$ grow. Is my understanding correct?

2. If this understanding is correct, then it seems like for in the small-noise regime (e.g. $\sigma_i \leq 1$), the 4-th order terms are not dominant terms in the expression, and thus all dominant terms get worse as $c$ increases, suggesting that Adagrad-Norm has the best guarantee. Is this correct?

3. As I've written elsewhere, it seems to me that the parameter-partitioning strategy used in practice (group by rows/columns) is not particularly informed by the theoretical results, other than the fact that the theoretical results suggest that in some regimes an intermediate number of groups may outperform both Adagrad and Adagrad-Norm. Is this correct?

4. In Table 3, it is my understanding that the column HP counts the number of hyperparameters per method. Based on this, it seems to me that the authors treat SN as not introducing any additional hyperparameters; this seems slightly misleading; the number of groups in the parameter partition/choice of parameter partitioning creates new hyperparameters. It is true that using a particular fixed strategy (i.e. the row/column grouping, applied only to 2D linear modules) fixes the hyperparameters associated with this procedure, but this is true of any algorithm. For example, the number of hyperparameters associated with GaLore can be reduced if one commits ahead of time to using a particular procedure for picking subspace dimension.

**Relation To Broader Scientific Literature:**

Both of the proposed modifications, SN and SM, are closely related to existing methods. SN is closely related to Adam-mini, which also partitions parameters into blocks and maintains adaptive learning rates per-block. However the partitioning strategy in Adam-mini is related to the block-diagonal structure of the Hessian, while the theoretically-optimal partitioning strategy for SN is related to coordinate-wise stochastic gradient noise levels. However, the practical procedures for parameter partitioning proposed in the appendix are much more simplistic and do not analyze coordinate-wise noise; instead they merely group matrix-valued parameters into blocks of rows and/or columns depending on which dimension is larger.

SM is closely related to low-rank learning strategies, particularly GaLore, which also restricts updates to a low-dimensional subspace chosen based on estimating leading singular subspaces. Unlike GaLore, SM involves also performing an update in the orthogonal complement of the identified subspace, allowing the authors to still derive convergence guarantees.

The authors claim that related methods (e.g. Adam-mini, factorization like Shampoo/SOAP, low-rank parameterization like LoRA and GaLore) either lack theoretical guarantees, sacrifice too much in performance, or greatly increase the cost of parameter tuning.

Zhang, Yushun, et al. "Adam-mini: Use fewer learning rates to gain more." arXiv preprint arXiv:2406.16793 (2024).

Zhao, Jiawei, et al. "Galore: Memory-efficient llm training by gradient low-rank projection." arXiv preprint arXiv:2403.03507 (2024).

Vyas, Nikhil, et al. "Soap: Improving and stabilizing shampoo using adam." arXiv preprint arXiv:2409.11321 (2024).

Gupta, Vineet, Tomer Koren, and Yoram Singer. "Shampoo: Preconditioned stochastic tensor optimization." International Conference on Machine Learning. PMLR, 2018.

**Theoretical Claims:**

I looked at the proof of Thm. 3.1. The lines I checked seemed correct, and the techniques seem to be reasonable extensions of traditional convergence bounds for Adagrad under L-smoothness assumptions.

---

> ### Author Rebuttal · Authors · 2025-03-31
>
> We appreciate the reviewer’s careful consideration and constructive feedback. Below, we clarify points raised and address specific concerns.
> ___
> > Question 1 and 2
>
> We answer question 1 and 2 of the reviewer together as they are related:
> - The reviewer is correct in that arbitrarily increasing the number of subsets $c$ towards $d$ (moving from AdaGrad-Norm towards AdaGrad-Coordinate) would yield a suboptimal dependency and each n-th order term of the noise requires a more careful of balancing the subset size. Our derivation in Appendix B.2 shows the effect of the number of subsets c on all n-th order noise terms that appear in the bound.
> In particular, Table 2 shows that the grouping of $c=d^{2/5 (1+\beta)}$ groups of size $d^{3/5-2\beta/5}$ each (which depends on the fraction of noisy coordinates) gives better dependency than $c=1$ (AdaGradNorm) or $c=d$ (AdaGrad-Coordinate) on most noise-density $\beta$ setting.
> - And as the reviewer points out in question 2, indeed that in the low noise regime, AdaGrad-Norm would attain the best guarantee (second row of Table 2) and our grouping strategy of $c=d^{2/5 (1+\beta)}$ groups would be suboptimal. There, a more careful case-work analysis to handle edge cases could help attain a more optimal strategy.
> - Regarding the reviewer’s point on $\delta$, the probability grows with $c$ but since our guarantee is high-probability, the worst case is $log(d/ \delta)$ so it is not the dominant term in the final bound.
> - Finally, we thank the reviewer for the attention to details and constructive feedback. We will incorporate the discussion above and revise the writing in the section after Theorem 3.1 to improve clarity and intuition.
>
> > Question 3
>
> - Our theory provides an optimal grouping strategy that depends on the noise density. However, in practice, we must trade off the cost to figure out a good grouping and the performance gain from it.
> - The key from the theory improvement is to group the coordinates with similar noise magnitudes together. However, any expensive method to figure out these groups (e.g. the Hessian in Adam-mini) would have detrimental effects on the wall clock time and memory (which are pointed out as crucial concerns by reviewer t4t5). Instead, our heuristic is meant to be a simple method to capture most of these groups.
> - Intuitively, coordinates in the same row/column either act on the same input or are used to compute the same output. The noise and normalization on each input and output would affect coordinates in the same row/columns in a correlated way.
> To provide some evidence, we perform the experiments in Section D.1 again but with the noise grouped by the corresponding dimension according to the heuristics. We show it in the plot at https://imgur.com/a/uKfHJ1B.
> - There we see most groups have very low noise (very close to 0, namely less than 10^-12) while a small number of groups (top 1 percentile in the annotation) have much larger noises.
> - Overall, our heuristics aim to capture the similar noise coming from the same inputs and outputs. Our experiments suggest that this is a major part of the gain. There might be other simple sources of correlation in the noise magnitudes, which we leave for future work.
>
> > Question 4
> - We agree with the reviewer. If there is an effective adaptive strategy to picking the subspace dimension, then one should not count that as a hyperparameter. Unfortunately, at this time, we are not aware of an effective strategy.
> - We do not tune for the number of groups. With tuning, as in Figure 5, the performance can be improved. We will rename that column to more precisely reflect the number of tunable parameters as opposed to hyperparameters. The number of tuning runs is a more accurate measure of tuning cost but is not easily obtainable.
> ___
> Other points
> > modifications also improve performance on smaller models/other architectures
> - Please see our response to Reviewer oW9K for additional experiments on vision tasks.
>
> > discuss Adam-mini in the main body so that readers are aware of the relevant work
> - We want to note that Adam-mini is concurrent work to ours, to be published in ICLR this year, but we agree with the reviewer that Adam-mini is highly relevant and including a detailed comparison in the main body improves the quality of our paper. We will include a more detailed comparison in the main body of our revision with a draft as in our response to reviewer 6d8c.
>
> > clarify what is meant by k would be an appropriate addition.
> > move norm notation to first paragraph.
>
> We thank the reviewer for the suggestions and attention to details. We will clarify the meaning of $k$ in our revision and include all the norm subscripts in theorems for clarity in our revision.
> ___
> We sincerely appreciate your detailed reviews and constructive suggestions. We believe that the discussion and additional experiments improve the quality of our submission and hope this provides sufficient reasons to raise the score.

---

> > ### Comment · Reviewer_KhrK · 2025-04-02
> >
> > >in the low noise regime, AdaGrad-Norm would attain the best guarantee (second row of Table 2)
> >
> > I am still a little confused about this point. It seems like by re-scaling the loss-function and the gradient estimates by the same scalar constant, one could arbitrarily change the values of $\sigma_i$. Note that in practice, gradient estimates are derived from mini-batch gradients, so re-scaling the loss function by a scalar would re-scale these estimates by the same scalar. This seems to suggest that a re-scaling of the loss function can move one into and out-of the regime where Thm. 3.1 implies a benefit from parameter grouping. This is very counter-intuitive to me; re-scaling the loss function preserves the geometry of the optimization problem and does not usually drastically change guarantees. Am I over-looking something?

---

> > > ### Author Response · Authors · 2025-04-02
> > >
> > > In the context of Table 2, the gain of grouping comes not from the magnitude of the noise but rather the *density* of the noise. In Table 2, we mean low-noise as in low-noise density i.e. $\beta=0$ means that only 1 coordinate contains noise. While scaling the loss would indeed scale the noise magnitude, it would not impact the noise density which is the primary contribution to the dimensional dependency that our analysis aims to target. Our Theorem 3.1 is also largely scale invariant: if we, say, double all the loss functions, the LHS of Theorem 3.1 goes up by a factor 4, but $G(\delta)$ (containing $f(x_1)-f(x^*)$ as in appendix F) and the sum of $|| \sigma_{\Psi_i} ||$ also go up by a factor of 2 each so it is still the same statement.

---

### Official Review · Reviewer_6d8c · 2025-03-14

**Overall Recommendation:** 3

**Summary:**

This paper proposes two techniques, Subset-Norm (SN) and Subspace-Momentum (SM), to reduce the memory footprint of adaptive optimization methods (like Adam and AdaGrad) when training large deep learning models, particularly large language models (LLMs). Subset-Norm reduces the memory of the adaptive step-size term by sharing it across subsets of parameters. Subspace-Momentum reduces momentum memory by restricting momentum updates to a low-dimensional subspace and using SGD in the orthogonal complement. The paper provides theoretical convergence guarantees for both methods and demonstrates empirically that combining them (SNSM) improves both memory efficiency and performance (lower perplexity) compared to standard adaptive optimizers on LLM pre-training and fine-tuning tasks.

**Claims And Evidence:**

Most claims are supported by evidence, but some require further strengthening. For example, the main claim is that SN and SM improve performance and memory efficiency. While the empirical evidence is generally strong, the gains are sometimes modest, and the "why" behind the performance improvement of SM is not fully explained.

**Essential References Not Discussed:**

The paper cites many relevant works, but some important ones are not included in the comparison such as Adam-mini and FLORA.

**Experimental Designs Or Analyses:**

The experimental setup is generally well-described, following standard practices for LLM pre-training and fine-tuning. The ablation studies on subset size and subspace selection are useful. However, in addition to Adam and GaLore, a wider range of baselines would strengthen the empirical evaluation.

**Methods And Evaluation Criteria:**

The proposed methods and evaluation criteria make sense for the problem.

**Other Comments Or Suggestions:**

None.

**Other Strengths And Weaknesses:**

Strengths: The problem of optimizer memory is clearly motivated, and the proposed solutions are intuitive.
Weaknesses: Compared to standard adaptive optimizers, the proposed methods are fairly complex, and the reliance on SVD for subspace selection in SM is a computational bottleneck.

**Questions For Authors:**

See above.

**Relation To Broader Scientific Literature:**

The paper's contributions are related to memory-efficient adaptive optimization and LLM training.

**Theoretical Claims:**

I didn't thoroughly check the correctness of the proofs.

---

> ### Author Rebuttal · Authors · 2025-03-31
>
> We appreciate the reviewer’s careful consideration and constructive feedback. Below, we clarify points raised and address specific concerns.
> ___
>
> > the "why" behind the performance improvement of SM is not fully explained.
>
> This is a limitation that is general to the class of algorithms that utilize momentum and it is an important open challenge for the community at large because the theoretical bound for momentum and without momentum are the same. We believe that any advance for SGD with momentum will lead to a better understanding of SM in our setting.
>
> > in addition to Adam and GaLore, a wider range of baselines would strengthen the empirical evaluation.
>
> We will add comparison with the following methods to Table 3 of our revision for the pretraining task. Some comparisons are shown below:
> | Method          | LLaMA 60M (1.3B tokens) | LLaMA 130M (2.62B tokens) | LLaMA 350M (7.86B tokens) | LLaMA 1B (13.1B tokens) |
> |-----------------|-------------------------|----------------------------|----------------------------|--------------------------|
> | AdamSNSM (ours) | 29.74                  | 22.43                      | 16.91                      | 13.96                   |
> | AdamW           | 30.46                  | 24.60                      | 18.67                      | 16.00                   |
> | GaLore          | 34.73                  | 25.31                      | 20.51                      | 16.76                   |
> | FLORA           | 32.52                  | N/A                        | 23.69                      | N/A                     |
> | LoRA            | 34.99                  | 33.92                      | 25.58                      | 19.21                   |
> | ReLoRA          | 37.04                  | 29.37                      | 29.08                      | 18.33                   |
>
> >  some important ones are not included in the comparison such as Adam-mini and FLORA.
>
> We currently cite and compare with these works in the related works in the related works, Section A of the Appendix. However, we will include a more detailed comparison to Adam-mini in our revision, where we include a draft below:
>
> "While Adam-mini also employs a grouping strategy for the adaptive step size, it is primarily motivated empirically and lacks a general grouping strategy for general parameters. Adam-mini gives a heuristic approach to form the group for transformers’ parameters without showing theoretically what this grouping is trying to achieve and if this goal is achieved, why the convergence is improved. In contrast, our theory results show that grouping by noise magnitude leads to improvement. Our heuristic is an efficient method toward this goal and the experiment validates this by showing that our groups successfully isolate the noise into a small number of groups and perhaps as a consequence, have improved performance (also see our additional experiments below).  We further demonstrate in Table 2 that there are scenarios where subset-norm attains better convergence than existing methods.
> In experiments, our AdamSNSM uses less memory than Adam-mini, due to the fact that Adam-mini uses full momentum while we use momentum only in a subspace (which outperforms full momentum in many cases given a good choice of subspace). Furthermore, in terms of perplexity, Adam-mini performs very closely to AdamW while our methods outperform Adam (which performs similarly to AdamW) on a range of language tasks and model sizes."
>
> > Compared to standard adaptive optimizers, the proposed methods are fairly complex
>
> The theories we developed for SN and SM are general and allow for many options, including future works. However, as reviewer t4t5 and KhrK pointed out, we also developed specific implementations that are simple and effective.
>
> > The reliance on SVD for subspace selection in SM is a computational bottleneck.
>
> As demonstrated in Table 6 on page 8, our methods do not depend on the SVD update gap to be as frequent as GaLore. In contrast to GaLore, our algorithms (e.g. AdaGradSNSM) could even benefit from a larger update gap as in Table 6, further reducing the cost of SVD, while achieving the best performance out of all baselines. Section C.4 of the Appendix contains additional ablation studies on the projection gap and rank which control the additional overhead cost and benefit from the projection computation.
> Finally, please refer to our response to reviewer t4t5 which contains additional results showing the wallclock time and peak memory for different optimizers.
> ____
> We sincerely appreciate your detailed reviews and constructive suggestions. We believe that the discussion and additional experiments improve the quality of our submission and hope this provides sufficient reasons to raise the score.

---

### Decision · Program_Chairs · 2025-05-01

**Decision:**

Accept (poster)

**Comment:**

This submission considers two modifications to adaptive algorithms for training neural networks, e.g., the adagrad/adam family of algorithms, which can reduce the memory footprints of these algorithms. There is both theoretical analysis in the form of convergence guarantees for their modified algorithms and there is also experimental validation in the form of pre-traing/fine-tuning language models and CIFAR100 classification; in both settings comparisons are made with baselines like adam, GaLore, or adagrad. The empirical benefits in terms of memory are clear and the performance as an optimization algorithm seems reasonable as well, with many ablation experiments in different settings (language, vision).

In the rebuttal period, the authors addressed the majority of the concerns of the reviewers. The main sticking point seemed to be comparison with adam-mini, recently proposed optimizer for the same tasks that also claims a reduced memory footprint. To this, the authors agreed to cite and discuss adam-mini in their paper.

The other sticking point was the fact that the analysis presented is not completely connected to what is done in practice; indeed the analysis relies on certain knowledge about the stochastic noise in the gradient computations that is typically not available nor easy to estimate and a different heuristic is used in the actual implementation of these methods. Although I agree that it would be a much stronger contribution if the theoretical results could characterize what is done in practice, I do not see this as a strong negative for the paper because of the accompanying experiments, which are extensive.

For these reasons, I recommend to accept this paper.